# Turbulent and Boundary Layer Characteristics during VOCALS-REx

Dillon S. Dodson[1] and Jennifer D. Small Griswold[1]

[1]Department of Atmospheric Sciences, University of Hawaii, Manoa, Honolulu, HI, USA

**Correspondence:** Jennifer D. Small Griswold (smalljen@hawaii.edu)

**Abstract.**

Boundary layer and turbulent characteristics (surface fluxes, turbulent kinetic energy (TKE), turbulent kinetic energy dissipation rate ($\epsilon$), etc.), along with synoptic scale changes in these properties over time, are examined using data collected from 14 research flights made with the CIRPAS Twin Otter Aircraft. Data was collected during the VOMOS Ocean-Cloud-Atmosphere-Land Study-Regional Experiment (VOCALS-REx) at Point Alpha (20°S, 72°W) in October and November of 2008 off the coast of South America. The average boundary layer depth is found to be 1175-m, with 28% of the boundary layer profiles analyzed displaying decoupling. Analysis of correlation coefficients indicate that as atmospheric pressure decreases, the boundary layer height ($z_i$) increases. The increase in $z_i$ is accompanied by a decrease in turbulence within the boundary layer. As $z_i$ deepens, cooling near cloud top cannot sustain mixing over the entire depth of the boundary layer, resulting in less turbulence. As the latent heat flux (LHF) and sensible heat flux (SHF) increases, $z_i$ increases, along with the cloud thickness decreasing with increasing LFH. This suggests that an enhanced LHF results in enhanced entrainment which acts to thin the cloud layer while deepening the boundary layer.

A maximum in TKE on Nov. $1^{st}$ (both overall average and largest single value measured) is due to precipitation acting to destabilize the sub-cloud layer (through evaporation away from the surface), while acting to stabilize the cloud layer. Enhanced moisture above cloud top from a passing synoptic system also acts to reduce cloud top cooling, reducing the potential for mixing of the cloud layer. This is observed in both the vertical profiles of the TKE and $\epsilon$ values, where it is found that the distributions of turbulence for the sub-cloud and in-cloud layer are completely offset from one another, with the TKE in the sub-cloud layer maximizing for the analysis period, while the TKE in the in-cloud layer is below the average in-cloud value for the analysis period. Measures of TKE, $\epsilon$, and the buoyancy flux averaged over all 14 flights display a maximum near cloud middle (between normalized in-cloud values of 0.25-0.75). Seven of the fourteen flights display two peaks in TKE within the cloud layer, one near cloud base and another near cloud top, signifying evaporative and radiational cooling near cloud top and latent heating near cloud base. Overall, it appears that turbulence measured at Point Alpha is weaker than that measured over the open ocean to the west of Point Alpha, and that measured during other scientific campaigns.

## 1 Introduction

Stratocumulus (Sc) clouds have a significant impact on climate due to their large spatial extent, covering approximately 20% of Earth's surface (23% over the ocean and 12% over the land) in the annual mean (Randall et al., 1984). According to Wood (2012), the subtropical eastern oceans in particular are marked by extensive regions of Sc sheets (often referred to as semipermanent subtropical marine stratocumulus sheets). Of those, the largest and most persistent Sc deck in the world, the

30 Peruvian Sc deck, lies off the west coast of South America (Bretherton et al., 2004), making its role in climate an essential building block to improved modeling of the overall earth system. A better understanding of Sc decks is therefore necessary to improve our physical understands of mechanisms controlling Sc clouds, and to improve confidence in climate model sensitivity (Zhang et al., 2013), especially considering climate models suffer from order-one uncertainties in Sc cloud representation (Noda and Satoh, 2014; Gesso et al., 2015).

It is a challenge for models to successfully simulate the Peruvian Sc deck due to the importance of subgrid scales and physical processes which are poorly represented (Wood et al., 2011). Most models continue to struggle with the boundary layer vertical structure (Wyant et al., 2010) which is important for determining Sc cloud properties. One example, as discussed in Akinlabi et al. (2019), is that a robust estimation of the turbulent kinetic energy dissipation rate ($\epsilon$) is needed when creating subgrid models for Lagrangian trajectory analysis of passive scalars (Poggi and Katul, 2006) or large-eddy simulation. Other

vertical profiles of turbulent fluxes (liquid water, water vapor, energy) determine the mean state of the boundary layer and the resulting properties of the Sc deck (Schubert et al., 1979; Bretherton and Wyant, 1997; Ghate and Cadeddu, 2019).

Although turbulence is critical to atmospheric boundary layer, microphysical, and large scale cloud dynamics, it is difficult to measure, with literature on describing cloud-related turbulence based on in situ data being scarce (Devenish et al., 2012; Shaw, 2003). This study therefore aims to characterize turbulence throughout the vertical profile of the Stratocumulus topped

marine boundary layer (STBL) over a three-week observation period in October and November of 2008 during the Variability of the American Monsoon Systems (VAMOS) Ocean-Cloud-Atmosphere-Land Study-Regional Experiment (VOCALS-REx). A large in situ dataset was collected throughout the boundary layer, allowing for analysis (on a variety of spatial and temporal scales) in the aims of improving predictions of the Southeast Pacific coupled ocean-atmosphere-land system (Wood et al., 2011). This dataset allows for a classification of turbulent properties not only through vertical profiles, but provides an

opportunity to analyze how turbulence changes within the boundary layer with varying synoptic conditions.

The main objectives of this paper include a quantification of the amount of turbulence occurring within the boundary layer through the evaluation of turbulent kinetic energy (TKE), $\epsilon$, and other turbulent flux measurements. In particular, the main goals include: (1) Analyze day to day variability in turbulent measurements and boundary layer characteristics, relating them to synoptic changes in meteorological conditions; (2) Determine average turbulent values throughout the vertical structure of

55 the STBL, classifying the STBL based on different turbulent profiles analyzed.

There has been a plethora of publications stemming from the VOCALS-REx campaign over the last ten years. Papers range from focusing on climatic and synoptic conditions for the VOCALS region (Toniazzo et al., 2011; Rahn and Garreaud, 2010a, b; Rutllant et al., 2013), analyzing cloud-aerosol interactions (Jia et al., 2019; Blot et al., 2013; Painemal and Zuidema,

2013; Twohy et al., 2013), and analyzing precipitation, boundary layer decoupling, and other boundary layer characteristics
(Jones et al., 2011; Bretherton et al., 2010; Terai et al., 2013; Petters et al., 2013; Zheng et al., 2011), to name a few. A total
of five aircraft platforms and two ship based platforms were utilized during VOCALS-REx (Wood et al., 2011), with most
publications from VOCALS-REx relying and/or focusing on aircraft observations and other data sources outside of those used
here (all but Zheng et al. (2011) and Jia et al. (2019) mentioned above). Results found and presented here therefore provide
not only a collection of in-situ turbulent measurements, but provide for the opportunity to relate results to other findings at
additional measurement locations within the VOCALS domain. An extensive look at turbulent characteristics of the boundary
layer during VOCALS-REx does not exist (note that although Zheng et al. (2011) does give a broad analysis of boundary layer
characteristics, their focus on turbulence was minimal), which is puzzling given that the Twin Otter aircraft (the data used here,
see Section 2.1) was instrumented with an objective to make turbulence measurements.

Section 1.1 introduces typical boundary layer vertical structure and scientific background. Section 2 provides an overview
of the data and methods, followed by synoptic and boundary layer characteristics during VOCALS-REx in Section 3. Section
4 will evaluate and discuss the results. Section 5 will provide concluding remarks.

## 1.1 Boundary Layer Vertical Structure

The vertical structure of the boundary layer is strongly tied to the horizontal and vertical structure of Sc clouds (Lilly, 1968;
Bretherton et al., 2010). The STBL is characterized by Sc cloud tops located at the base of an inversion, with subsiding air
aloft (as part of the descending branch of the Hadley cell circulation) and well mixed conditions and near-constant conserved
variables with height throughout the boundary layer (Wood, 2012). Multiple papers have analyzed typical well mixed STBL
vertical structures (i.e., Albrecht et al. (1988); Nicholls (1984)), showing constant potential temperature and mixing ratio with
height up until the inversion, when the mixing ratio (potential temperature) sharply decreases (increases). Horizontal winds
(both direction and velocity) are typically constant with height throughout the well mixed boundary layer, with changes in both
direction and strength typically present at the top of the STBL, influencing cloud-top entrainment (Mellado et al., 2014; Kopec
et al., 2016; Schulz and Mellado, 2018).

Convection in the STBL is limited. Unlike updrafts through convective heating over the ground, updrafts within the STBL
do not penetrate the inversion. This is because convection within the STBL is primarily driven by cooling near cloud top and
not heating at the ocean surface, where cloud top cooling is primarily from a combination of (1) longwave radiational cooling
and (2) evaporational cooling from entrainment. The cloud top cooling leads to instability and the convection of warmer, moist
air at the surface (Lilly 1968). The cloud cover is greatest when the STBL is shallow [$0.5 < z_i < 1$ km], where $z_i$ is the inversion
layer (i.e., boundary layer) height (Wood and Hartmann, 2006).

It is known that clouds are areas of enhanced turbulence (Pinsky and Khain, 1996). Therefore, Sc sheets are turbulent but
in contact with an almost non-turbulent upper atmospheric environment. The boundary layer top is characterized by several
strong gradients, including the cloud boundary (gradient in liquid water content), the entrainment zone (gradient in vorticity,
where the entrainment zone separates regions of weak and strong mixing between laminar (warmer and dryer) flow above
and turbulent (cooler and more moist) flow below), and the capping inversion (gradient in potential temperature). The cloud

boundary typically lies in the entrainment zone (Albrecht et al., 1985; Kurowski et al., 2009; Malinowski et al., 2013), which in turn lies in the capping inversion, although these layers do not necessarily coincide (Mellado, 2017). Turbulent analysis of these layers in Jen-La Plante et al. (2016) found that turbulence (both TKE and $\epsilon$) decreases moving from cloud top into the free atmosphere above. Through cloud top entrainment, the STBL deepens beyond 1-km and can become decoupled. According to Bretherton and Wyant (1997), due to longwave cooling at the cloud top being unable to maintain mixing of the positively buoyant entrained air over the entire depth of the STBL, the upper (cloud containing) layer (turbulence driven primarily by cloud top cooling) becomes decoupled from the surface moisture supply (turbulence driven by surface-fluxes and shear).

The vertical profile of various turbulent fluxes, particularly that of buoyancy (which is dependent on moisture and heat fluxes which drive buoyancy differences), can tell one a lot about the state of the STBL. For a boundary layer to remain well mixed, the vertical energy and moisture fluxes must be linear functions of height. According to Bretherton and Wyant (1997), the buoyancy flux is not a linear function of height however (unlike that of a dry boundary layer). An increase in the buoyancy flux above cloud base is typically proportional to the upward transport of liquid water that is required to sustain the cloud against entrainment drying (i.e., continued mixing of the cloud layer is sustained by surface fluxes). Decoupling of the boundary layer (and the subsequent decrease in cloud cover) can occur when the sub-cloud buoyancy fluxes become negative, capping convection below cloud base (Albrecht et al., 1988; Ackerman et al., 2009). According to Shaw (2003), one of the main sources of TKE in clouds is evaporative cooling (due to the entrainment of dry air) and condensational heating (due to droplet condensational growth), implying the buoyancy flux is the primary generator of TKE in the STBL (Schubert et al., 1979; Heinze et al., 2015). Given this, the buoyancy flux nearly always has a maximum in the cloud layer (Nicholls and Leighton, 1986; Bretherton and Wyant, 1997), with TKE being generated due to longwave and evaporational cooling at cloud top, and condensational heating at cloud base (Moeng et al., 1992). Nicholls (1989) observed through aircraft observations that the largest buoyancy fluxes are close to cloud top, with further observations (Caughey et al., 1982; Nicholls, 1989) suggesting that the descending regions of air originating near cloud top are more a result of radiative cooling rather than evaporative cooling.

The main source of moisture for the STBL is supplied by the surface latent heat flux (LHF), making it an important source of buoyant TKE production (Bretherton and Wyant, 1997), with the surface sensible heat flux (SHF) typically being a much weaker source of turbulence. The SHF and LHF can be compared using the Bowen ratio (the ratio of the sensible to the latent heat flux). A larger LHF (or smaller Bowen ratio) leads to decoupling of the boundary layer due to the LHF concentrating convective energy generation (through condensational and evaporational heating/cooling) within the cloud layer. To state this another way, an enhanced LHF leads to increased moisture transport to the cloud layer and a thicker cloud, producing more turbulence and enhanced entrainment cooling near cloud top. Enhanced entrainment results in a deepening of the boundary layer, which favors decoupling (Jones et al., 2011). It is argued in (Bretherton and Wyant, 1997) that the surface LHF is the most important determinant of decoupling within the STBL.

Vertical velocity variance typically displays the strongest updrafts and downdrafts in the upper half of the STBL (Hignett, 1991; Heinze et al., 2015; Mechem et al., 2012), consistent with the largest production of turbulence being contained within the cloud layer. A positive (negative) vertical velocity skewness indicates that strong narrow updrafts (downdrafts) are surrounded

by larger areas of weaker downdrafts (updrafts). It has been found that negative vertical velocity skewness is typically contained within most of the cloud layer and below (Nicholls and Leighton, 1986; Nicholls, 1989; Mechem et al., 2012) for well mixed boundary layers, whereas a decoupled boundary layer containing cumulus below stratocumulus may contain positive vertical velocity skewness (de Roode and Duynkerke, 1996) due to convection being driven in the surface layer (as compared to cooling near cloud top). The tendency of the vertical velocity skewness to be positive in a strongly precipitating STBL is also well known (Ackerman et al., 2009), with precipitation being a key contributor leading to boundary layer decoupling (Rapp, 2016; Yamaguchi et al., 2017; Feingold et al., 2015).

## 2  Data and Methods

### 2.1  Data

Data was collected during the Variability of the American Monsoons (VAMOS) Ocean Cloud-Atmosphere-Land Study-Regional Experiment (VOCALS-REx) from the Peruvian Stratocumulus deck off the west Coast of Chili and Peru during October and November of 2008. VOCALS-REx used various platforms, including five aircraft and two research vessels to accumulate an extensive dataset of the boundary layer, lower free troposphere, and cloud deck along 20°S from 70°W to 85°W. Although multiple sampling platforms, locations, and mission types were deployed during the campaign (see Wood et al. (2011)), data collected by the Center for Interdisciplinary Remotely-Piloted Aircraft Studies (CIRPAS) Twin Otter aircraft will be the focus of this paper, which collected data in the vicinity of 20°S, 72°W; from here on termed Point Alpha. The Twin Otter aircraft was operational for 19 flights from October $16^{th}$ to November $13^{th}$, 2008.

The Twin Otter platform is ideal for a turbulent analysis of the boundary layer due to the aircraft being instrumented to make turbulence and cloud microphysics measurements, with the same location being sampled for each flight. The Twin Otter is also a relatively slow-moving aircraft with a flight speed of roughly $60$ ms$^{-1}$, allowing for a higher resolution of spatial sampling as compared to a faster moving aircraft. Each of the Twin Otter flights was carried out using a stacked flight path (Wood et al., 2011), which involved using stacked legs of 50-100 km in length (horizontal flight paths) to sample various levels of the boundary layer and cloud layer, with at least one aircraft vertical sounding (vertical profile) performed for each flight where the aircraft sampled the free upper troposphere and boundary layer in a single ascent or descent. Each flight of five hours originated from Iquique Chile, allowing for roughly three hours of sampling at Point Alpha.

Of the 19 flights performed by the Twin Otter, only 14 are used here due to instrumentation failure on five of the flights (Phase Doppler Interferometer and the cloud/aerosol probe). Table 1 displays each of the Research Flights (RF) used in this paper. All flights occurred during the day, with all but two flights (RF 8 and RF 17) starting around 7:00 AM local time, with the first vertical profile flown around 8:00 AM local time at Point Alpha. Having each flight sample the same location at roughly the same time is critical, as turbulence typically displays diurnal patterns, with the strongest turbulent mixing occurring during the night when longwave radiational cooling dominates due to the absence of the stabilizing effect of shortwave absorption, which is largest near cloud top due to the scattering of solar radiation limiting absorption lower in the cloud layer (Hignett, 1991).

Meteorological variables were collected at 40-Hz (including $u$, $v$, and $w$ wind velocity, water vapor mixing ratio ($q$) and potential temperature ($\theta$), to name a few) while most cloud and aerosol data were collected at 1-Hz. A 5-port Radome wind gust probe was used with plumbing that effectively trapped liquid water, preventing any liquid water from obstructing the pressure transducer lines. There were zero failures during the campaign, with an accuracy of $\pm\,0.4$ ms$^{-1}$ for horizontal wind components and $\pm\,0.2$ ms$^{-1}$ for vertical velocity. The LI-COR 7500 H$_2$O/CO$_2$ gas analyzer was used for all measurements of absolute humidity and $q$, with an ambient air intake setup that resulted in the LI-COR source and detector window to be liquid free, even during prolonged cloud penetrations. The LI-COR accuracy is reported to be within 1% of the actual reading. Further instrumentation information can be found in Zheng et al. (2010) and Wood et al. (2011).

To analyze the synoptic conditions over the study period, data from the National Centers for Environmental Prediction (NCEP) / National Center for Atmospheric Research (NCAR) Reanalysis Project (NNRP, Kistler et al. (2001)) will be used. The data resolution of the NCEP/NCAR reanalysis data is 2.5° x 2.5° x 17 pressure levels, available at six hour intervals. The resolution of this data is suitable for analyzing synoptic scale patterns, but is not ideal for depicting mesoscale variability that may be present on any given day. Boundary layer height is also derived from relative humidity data from the European Centre for Medium-Range Weather Forecasts (ECMWF) Re-Analysis (ERA5), which has a resolution is 0.25° x 0.25° x 37 pressure levels, and is available at an hourly interval (Hersbach et al., 2020).

## 2.2 Turbulent Calculations

The randomness of turbulence makes deterministic description difficult, limiting description to statistics and average values of turbulence, in particular that of Reynolds decomposition (or averaging). Reynolds decomposition uses a mean value (over some time period, determined by low pass filtering or applying a linear trend) and subtracts it from the actual instantaneous velocity to obtain the turbulent component (or perturbation value). Reynolds decomposition is based on the underlying assumption that the turbulence is isotropic and stationary, conditions that are hardly fulfilled for atmospheric boundary layer flows however, especially when working with data spanning larger timeframes. The problem is defining how to average collected data to best represent the mean and turbulent components for the fluid flow (with shorter subsets of data having more stationary properties in general than that of longer subsets of data). Using the 40-Hz data, a 320-point averaging window is used here for all turbulent analysis, following the methods outlined in Jen-La Plante et al. (2016). A 320-point averaging window corresponds to 8 second subsets of data, or a roughly 440-m subset of data in the horizontal spatial scale (assuming average aircraft speed of 55 ms$^{-1}$). Linear regression is then applied to each 320-point averaging window to calculate the mean and determine the perturbation values.

Applying the averaging method discussed above leads to the calculation of the fluctuations of the $u$, $v$, and $w$ components of the velocity, along with other parameters used to measure various turbulent fluxes. Variables to be obtained include turbulent kinetic energy, which is given by:

$$TKE = \frac{1}{2}\left(\overline{u'^2} + \overline{v'^2} + \overline{w'^2}\right) \tag{1}$$

where $u'$, $v'$, and $w'$ are the fluctuations of the velocity components. The turbulent sensible heat, latent heat, and buoyancy fluxes will also be obtained, given by:

$$F_\theta = C_p\bar{\rho}\overline{w'\theta'} \tag{2}$$

$$F_q = L_v\bar{\rho}\overline{w'q'} \tag{3}$$

$$F_{\theta_v} = C_p\bar{\rho}\overline{w'\theta_v'} \tag{4}$$

respectively. Where $C_p$ is the specific heat of air (1005 J kg$^{-1}$K$^{-1}$), $L_v$ is the latent heat of vaporization at 20°C (2.45 · 10$^6$ J kg$^{-1}$), $\rho$ is the mean air density, and $\theta'$, $q'$, and $\theta_v'$ are the potential temperature, mixing ratio, and virtual potential temperature perturbations, respectively. Note that $\theta_v$ (given by $\theta_v = \theta(1+0.61q-q_l)$) is commonly used as a proxy for density when calculating the buoyancy. Humid air has a warmer $\theta_v$ because water vapor is less dense than dry air, while liquid water drops (if falling at terminal velocity) make the air heavier and therefore associates with a colder $\theta_v$, where $q_l$ is the liquid water mixing ratio.

Just like that of Reynolds decomposition, the calculation of $\epsilon$ is based on conditions that the flow is isotropic (i.e., uniformity in all directions), making the measurement of $\epsilon$ challenging. In particular, classical turbulence theory in the inertial subrange from Kolmogorov (1941) is based on assumptions of local isotropy. With that said, there are multiple methods to measure $\epsilon$, including the inertial dissipation method, structure functions, and the direct method. Siebert et al. (2006) found that both the inertial dissipation and structure function methods are useful, but the inertial dissipation method sometimes underestimates $\epsilon$ at low values due to no clear inertial subrange behavior being observed in the power spectral density, which is not the case for the structure function. The structure function method is therefore considered more robust for cases with small values of $\epsilon$, and will be used here. Due to questions of isotropy, $\epsilon$ will be evaluated on the $u$, $v$, and $w$ components of the wind, and an average dissipation rate will be calculated from the three components.

The calculation of $\epsilon$ comes from the analysis of the velocity perturbations through the n$^{th}$ order structure function (i.e., a statistic to analyze common variation in a time series). The perturbations, as for other turbulent parameters, are determined with respect to an averaging window of 320-points. Each subset of perturbations is then appended to the end of the previous subset to create a single time series of velocity perturbations. The structure function is given by:

$$S_n(l) = \left(\overline{|u(x+l) - u(x)|}\right)^n \tag{5}$$

where $l$ is the distance (or in the case of a temporal series, $l$ is equivalent to $t$ assuming constant flight speed). From Frisch (1995), $\epsilon$ using the n$^{th}$ order structure function can be obtained by using:

$$S_n(l) = C_n|l\epsilon|^{\frac{n}{3}} \tag{6}$$

where $C_n$ is a constant of the order 1. The second order structure function will be used here ($n=2$), where $C_2 = 2$ for transverse velocity fluctuations and $C_2 = 2.6$ for longitudinal velocity fluctuations (Chamecki and Dias, 2004), where vertical fluctuations are considered transversal and horizontal fluctuations are considered longitudinal. The structure function follows a 2/3 power law within the inertial subrange, and will only be used to calculate $\epsilon$ between frequencies of 0.3-5-Hz, neglecting

the higher frequency features attributed to interactions with the plane (i.e., vibrations due to the aircraft) and other instrumental artifacts.

    Figure 1 Panel (a) provides the power spectral density of vertical velocity and $q$ for three horizontal flight legs within RF3, one in-cloud, one sub-cloud, and one near surface. Note that the power spectral density follows a -5/3 power law fit (red) within the inertial subrange (as opposed to the 2/3 power law fit of the structure function). A spike in energy can be seen at $\sim$10-Hz,

which represents the aircraft interactions discussed previously. The power spectral density overlaid in black represents a single calculation using a 320-point averaging window. The data follows the -5/3 fit well, and the inertial subrange is well resolved for the averaging window used (with the light gray envelope representing the 0.3 to 5-Hz range). A lack of significant flattening within the power spectra at higher frequencies suggests that the random noise level is low (this is more evident in the vertical velocity spectra than that of the $q$ spectra).

Analysis of the turbulence as presented here introduces two types of error, including sampling and noise error. This must be analyzed to determine the statistical significance when analyzing vertical profiles, especially since error propagation into higher order moments can be significant (McNicholas and Turner, 2014). Sampling errors were estimated using approaches derived and discussed in Lenschow et al. (1994, 2000) and will not be repeated here. Noise error must be considered, as noise within the instrumentation may be significant enough that the atmospheric component of the variance is small compared to the

overall measured variance. Noise is measured using the extrapolations of the measured autocovariance functions to lag 0 by the structure function. This technique was introduced in Lenschow et al. (2000) to estimate the noise contribution from the second to fourth order moments. Although this technique was traditionally used to estimate lidar noise (Wulfmeyer, 1999; Wulfmeyer et al., 2010, 2016), it has also been extended to in-situ observations (Turner et al., 2014).

    Figure 1, Panel (b) provides the autocovariance function of vertical velocity and $q$ for a sub-cloud fight leg in RF3 (black).

The fit using the structure function is provided in red (vertical velocity) and green ($q$). The structure function at lag zero provides the mean variance, while the difference between the autocovariance and structure function at lag zero provides the system noise variance at the corresponding temporal resolution. It is clear that the atmospheric variance and noise can be separated. For example, from Panel (b), looking at the vertical velocity data, $\overline{w'w'} = 0.20$ m$^2$s$^{-2}$ and the noise variance $\overline{\delta_w^2} = 0.014$ m$^2$s$^{-2}$. This results in a noise standard deviation of $\delta_w = 0.12$ ms$^{-1}$.

Extending this analysis to determine the error propagation within higher order moments, error bars for vertical velocity variance ($w'w'$) and $q$-variance ($q'q'$), vertical velocity skewness ($w'w'w'$), and the kinematic moisture flux ($w'q'$) can be found in Panels (d) through (f), respectively, with noise error bars in red and sampling error bars in black. The noise error is negligible compared to the sampling error, in agreement with results from Turner et al. (2014). Note that some data points do not have noise error bars associated with them. This is due to the fact that the noise was so small, the error bars would not plot.

The various vertical profiles displayed show that the sampling errors result in a lack of statistical significance between flight legs of different altitudes.

    Equations used to determine the noise in the higher order moments from Wulfmeyer et al. (2016) are:

$$\sigma_{w'^2} \cong 2\sqrt{\overline{w'^2}}\sqrt{\frac{\delta^2}{N}} \qquad (7)$$

$$\sigma_{w'3} \cong 3\sqrt{3\overline{w'^2}}\sqrt{\frac{\delta^2}{N}} \qquad (8)$$

$$\sigma_{w'q'} \cong \sqrt{\overline{q'^2}\frac{\delta_w^2}{N} + \overline{w'^2}\frac{\delta_q^2}{N}} \qquad (9)$$

where N is the number of data points. Using this, the absolute error for the vertical velocity variance is found to be 0.00068 $m^2 s^{-2}$ and the relative error is 0.35% (the relative error for the $q$ variance is 1.9%). Both errors are very reasonable, and demonstrate the low noise of the instrumentation.

## 3 Synoptic and Boundary Layer Characteristics

### 3.1 Mean Synoptic Conditions

The Southeast Pacific Ocean is found on the eastern edge of the south-Pacific semipermanent subtropical anticyclone, characterized by large scale upper tropospheric subsidence leading to a strong temperature inversion with a well-mixed boundary layer below. The surface pressure therefore is controlled in part by the location of the south-Pacific subtropical anticyclone. This anticyclone is routinely interrupted (especially between fall and spring) by periods of relative low pressure which is associated with localized troughing or the passage of midlatitude cyclones to the south. Several papers (Toniazzo et al., 2011; Rahn and Garreaud, 2010a) have analyzed the synoptic characteristics during VOCALS-REx. These papers however tend to focus on the VOCALS-REx region as a whole and not specifically on Point Alpha, which is done in this section.

Figure 2 shows the mean of large-scale meteorological conditions, including sea level pressure, omega ($\omega$, representing vertical velocity in pressure coordinates) and 700-hPa geopotential height from NCEP reanalysis data over the study region between Oct. $19^{th}$ to Nov. $12^{th}$. The mean sea level pressure (panel (a)) displays the anticyclone near its climatological position of 30°S, 100°W (Toniazzo et al., 2011). The overlaid sea level pressure standard deviation (only displayed up to 20 hPa) shows variability increasing southward, indicating enhanced midlatitude storm tracks. Enhanced variability that is in line with midlatitude troughing from panel (b) also decreases toward the coast, suggesting more variation in the synoptic pattern over the open ocean as compared to the near-coastal region. This is as expected, as Barret et al. (2009) found that synoptic systems tend to weaken as they move towards the coast of South America.

The mean 700-hPa geopotential height is displayed in panel (b), overlaid with $\omega$ data. Subsidence (green shading) dominated the VOCALS-REx region, with Point Alpha having an average value of 57 hPa day$^{-1}$ at the 700-hPa level. While enhanced storm tracks were primarily contained within the mid-latitudes, the 700-hPa geopotential height displays midlatitude troughing extending between Point Alpha and the subtropical high (as was found in (Zheng et al., 2011)), suggesting that meteorological conditions at Point Alpha were influenced by both midlatitude synoptic systems and the subtropical anticyclone.

The sea-level pressure was also measured using both reanalysis data and aircraft 30-m level horizontal flight legs. Figure 3, panel (a) shows that the reanalysis data at Point Alpha tended to be on average 1.5-hPa greater than the aircraft measured

sea level pressure. The pressure decreased by roughly 3-hPa from Oct. $19^{th}$ to Nov. $12^{th}$, however, this decrease cannot be considered a seasonal signal because it is within synoptic scale variation. The sea surface temperature (SST) and atmospheric surface temperature (both measured during 30-m horizontal flight legs) increased steadily throughout the observation period, increasing by 2.79 and 2.28 °C, respectively.

## 3.2 Synoptic Variability at Point Alpha

Synoptic variability at Point Alpha is summarized by time series of geopotential height at various levels. Higher geopotential heights are associated with ridging aloft while decreases in geopotential heights are associated with synoptic disturbances or troughs. The 500-hPa geopotential height (see Figure 3) varied between 5840 and 5900-m, with an increase of 9-m between Oct. $19^{th}$ and Nov. $12^{th}$. Figure 3 also displays enhanced synoptic scale variation during October, with several disturbances effecting Point Alpha. The 500, 700 (panel (c)), 850, and 1000 (panel (d)) hPa geopotential heights alternate between areas of high and low height through Nov. $2^{nd}$. After Nov. $2^{nd}$, the 500-hPa geopotential height is more consistent, with height increasing over Point Alpha until Nov. $10^{th}$, at which point the height begins to decrease.

Besides minor disturbances in October, there are two main disturbances that stand out. The first disturbance occurs on Nov. $1^{st}$ and $2^{nd}$ (green shading in Figure 3), where both the 500 and 700-hPa heights have minimums (5842 and 3134 m, respectively) due to the influence of a synoptic system. The 850 and 1000-hPa heights also have secondary minimums. The second disturbance was the formation of a costal low, which can be seen by decreasing geopotential heights on Nov. $12^{th}$. Both the 850 and 1000-hPa geopotential heights reached minimums on Nov. $12^{th}$ (1498 and 104 m, respectively). This costal low reached a minimum (the coastal low was strongest) after the analysis period, on Nov. $15^{th}$ (Rahn and Garreaud, 2010a). The ridging which formed after Nov. $2^{nd}$ leads to the formation of the coastal low through the warming of the lower and middle troposphere (Garreaud and Rutllant, 2003).

The 700-hPa geopotential height map (not shown here) displayed a midlatitude trough developing and extending past Point Alpha from Oct. $29^{th}$ through Nov. $3^{rd}$. A deep midlatitude trough forms off the west coast of South America by Oct. $30^{th}$, extending past 15°S. The trough axis begins to move over Point Alpha by October $31^{st}$, with the main impacts of the trough on Point Alpha (in terms of lowest geopotential height) being observed on Nov. $1^{st}$ and $2^{nd}$. The 500-hPa geopotential height map (not shown here) shows the ridge axis directly over Point Alpha on Nov. $1^{st}$.

Figure 4 (panels (a) through (c)) show atmospheric wind direction and velocity using data collected from horizontal flight legs. Panel (d) and (e) displays wind direction and wind speed using data collected from aircraft vertical soundings. Atmospheric winds near the surface (measured during 30-m horizontal flight legs) at Point Alpha were mostly southerly (150 to 180°) with a mean of 176°. Strong wind shear was present near the inversion, with winds above the marine boundary layer (measured during horizontal flight legs above the inversion) having a mostly northwesterly component (mean of 273°) while having more variability in direction than that of the boundary layer (300 to 360°). Although on most flight days the wind speed and direction were mostly constant with height in the boundary layer (see panel (d) and (e)), on Nov. $1^{st}$ and $4^{th}$ (blue lines) the wind direction shifted sharply within the boundary layer from southerly to northeasterly, along with varying wind speed. On Nov. $2^{nd}$ (green line), the wind direction had its strongest westerly component (214°). Shear within the boundary layer is

not common. Zheng et al. (2011) suggest that this shear is linked to coastal processes such as the propagation of the upsidence
wave. It should also be noted however that the wind shear within the boundary layer is present on the same day (November
$1^{st}$) that the trough axis is located over point Alpha. On the proceeding day, the surface winds experience their most westerly
component. According to Rahn and Garreaud (2010a), as troughs approach the coast of South America, southeast winds are
typically replaced by southwest winds. Between Oct. $29^{th}$ and Nov. $2^{nd}$, wind direction within the boundary layer shows its
most variation, gradually shifting from 153° (most easterly component measured) to 213° (most westerly component mea-
sured), respectively. While the trough approaches the coast of Chile, southeast winds are replaced by southwest winds, as is
typical of synoptic scale disturbances in the region (Rahn and Garreaud, 2010a).

### 3.3 Boundary Layer Characteristics

Boundary layer height is perhaps the most important feature of the marine boundary layer (MBL), with $z_i$ being one of the
main dictators for boundary layer characteristics such as decoupling and cloud cover (Albrecht et al., 1995). Findings from
Rahn and Garreaud (2010a) at a separate observation point within the VOCALS-REx region suggests that $z_i$ tended to be
either low (600-m) or high (1500-m) with periods of high or low depth interrupted by rapid transitions between the two states
over 12 to 36 hour periods due to synoptic variability. Figure 5 shows the thickness of the Sc cloud layer, the thickness of
the inversion layer, and subsequently the MBL height for each flight. The expected lifted condensation level (LCL) for a well
mixed boundary layer is also provided, using $z_{LCL} = 123(T - T_d)$, where $T_d$ is dew point temperature. $z_i$ is also provided
from extrapolating relative humidity data from ECMWF reanalysis (Engeln and Teixeira, 2013). The cloud layer was identified
using a liquid water content (LWC) greater than or equal to 0.01 g m$^{-3}$, while the inversion layer was identified by the region
of greatest change in $q$ (absolute change $\geq 0.10$ g kg$^{-1}$ per measurement) and $\theta$ (absolute change $\geq 0.20$ K per measurement)
within the vertical profiles. This results in the bottom of the inversion layer characterized by the profiles beginning to lose the
boundary layer features, while the top of the inversion layer had lost all boundary layer features.

The average $z_i$ was 1175-m (see Table 2 for boundary layer characteristics), with the average cloud layer and inversion
thickness being 239 and 59-m, respectively. Figure 5 shows that $z_i$ varied between 996 and 1450-m, with mostly gradual
changes in height from flight day to flight day (note that the mean difference between $z_i$ and ECMWF-$z_i$ was $43 \pm 26$-m).
The average change in $z_i$ (in regards to the in-situ data) was 68 m day$^{-1}$ with four occurrences of a rate of change above 100
m day$^{-1}$. After Oct. $27^{th}$ is when the most significant changes took place to the cloud thickness and $z_i$. Between Oct. $27^{th}$
and $29^{th}$, $z_i$ increased from 995 to 1300-m (152 m day$^{-1}$, the second largest rate of change), where the ECMWF-$z_i$ shows
that the increase was mostly confined from Oct. $27^{th}$ to Oct. $28^{th}$. The next four flight days recorded the thickest cloud layers,
peaking on Nov. $1^{st}$ and $2^{nd}$ with thicknesses of 382 and 472-m, respectively. It should also be noted that between Oct. $29^{th}$
and $30^{th}$, $z_i$ decreased from 1300 to 1177-m (124 m day$^{-1}$, the third largest rate of change, although this is not conveyed in the
ECMWF-$z_i$ data). After Nov. $2^{nd}$, the cloud layer thinned and $z_i$ increased from 1136-m to 1450-m between November $4^{th}$
and Nov. $8^{th}$. Although this is a rate of 79 m day$^{-1}$, there is no in situ data in-between November $4^{th}$ and $8^{th}$. The ECMWF-$z_i$
provides a mean rate of change for this period of 93 m day$^{-1}$, with the largest change of 160 m day$^{-1}$ between November $7^{th}$
and $8^{th}$, suggesting a rapid rise in $z_i$, in concurrence from findings in Rahn and Garreaud (2010a). After $z_i$ peaks on November

$8^{th}$, $z_i$ falls rapidly over the next two days, showing decreases of 174 m day$^{-1}$ and 102 m day$^{-1}$ from Nov. $8^{th}$ to Nov. $10^{th}$, respectively.

Although the time series of cloud droplet number concentration is not shown here, it showed a notable dip to a minimum on Nov. $1^{st}$ of 81 cm$^{-3}$ (where the average is 292 cm$^{-3}$), corresponding with minimums in both boundary layer cloud condensation nuclei and aerosol number concentration. Above boundary layer aerosol number concentration had a maximum on Nov. $1^{st}$. However, this can most likely be attributed to enhanced moisture (see Figure 6) above the boundary layer due to the passing synoptic system, where enhanced moisture can increase the size of hygroscopic aerosols that would otherwise be too

small to be measured under dryer conditions.

Figure 6 shows vertical profiles (where the height ($z$) is normalized with the inversion height to give a non-dimensional vertical coordinate of $z/z_i$) of $\theta$, $q$, LWC, and the aerosol number concentration. Individual flight profiles are in gray, with the red profile representing the mean and the blue profiles representing the flights conducted on November $1^{st}$ (RF11) and Nov. $2^{nd}$ (RF12). Mean profiles show that on average the MBL is well mixed up to the inversion, which then prevents mixing into

the free atmosphere above (as evident by the decrease in aerosol number concentration between the boundary layer and free atmosphere).

The largest deviations from the mean in the profiles occur during the passage of the synoptic system on Nov. $1^{st}$ and $2^{nd}$. At this time, both RF11 and RF12 measured (1) The thickest Sc cloud layer, with Nov. $1^{st}$ having the largest average cloud droplet size (20.8 $\mu$m) and in-cloud drizzle rates, while November $2^{nd}$ had the lowest recorded cloud base and largest recorded

LWC; (2) A larger mixing ratio above the boundary layer. This suggests the presence of a moist layer aloft which may have helped in producing the thickest cloud layers observed; (3) The smallest differences in both $\theta$ and $q$ from the bottom to the top of the inversion layer. During the passage of strong events as described by Rahn and Garreaud (2010a), the inversion defining the MBL erodes, making it hard to define $z_i$. This process is partially displayed by the small differences in temperature and moisture across the inversion layer during the passage of the synoptic disturbance.

The differences in $q$ and $\theta$ can be better visualized in Figure 7, which shows the differences between below and above inversion values in panel (a). $z/z_i$ values between 0.85 and 0.95 were used for the averages below the inversion, while data between $z/z_i$ values of 1.10 and 1.20 were used for the averages above the inversion. Besides Nov. $1^{st}$, $2^{nd}$, and to a lesser degree Nov. $4^{th}$, the average difference in $\theta$ across the inversion was 17-K, while the average difference in $q$ was -6.2 g kg$^{-1}$. On Nov. $1^{st}$ when both reached a minimum difference, the difference between $q$ and $\theta$ across the inversion was 1.9 g kg$^{-1}$ and

14-K, respectively, where a weaker inversion allows for more entrainment mixing near cloud top (Galewsky, 2018).

There are multiple methods which can be used to analyze whether the boundary layer is well mixed or decoupled. Methods used here include (1) decoupling parameters and (2) analysis of the expected LCL for a well-mixed layer in relation to actual cloud base. Decoupling parameters $\alpha_\theta$ and $\alpha_q$ depend on the profiles of $\theta$ and $q$, respectively (Wood and Bretherton, 2004). The decoupling parameters measure the relative difference in $q$ and $\theta$ between the bottom (near the surface) and top (near the

inversion) portions of the boundary layer, and are given by:

$$\alpha_\theta = \frac{\theta(z_i^-) - \theta(0)}{\theta(z_i^+) - \theta(0)} \tag{10}$$

$$\alpha_q = \frac{(z_i^-) - q(0)}{q(z_i^+) - q(0)}, \tag{11}$$

where $z_i^+$ ($z_i^-$) is the level $\sim$25 m above (below) $z_i$, and $\theta(0)$ and $q(0)$ are the potential temperature and mixing ratio at the surface. Here, $z_i^+$ is calculated using data between $z/z_i$ values of 1.03 to 1.05, while $z_i^-$ is calculated using data between $z/z_i$ values of 0.95 to 0.97 (this is roughly 25 m above and below $z_i$, respectively). The closer to zero the decoupling parameters are, the more well-mixed the boundary layer is. Previous observations suggest that if the parameters exceed $\sim 0.30$, the boundary layer is decoupled (Albrecht et al., 1995).

Mixed layer cloud thickness represents the difference between $z_i$ and the LCL ($\Delta z_m$), and was found to be strongly corre-lated to decoupling in Jones et al. (2011). The difference between cloud base ($z_b$) and the LCL represents another decoupling index ($\Delta z_b$) related to the LCL presented in Jones et al. (2011). Decoupling of the boundary layer occurs when the boundary layer deepens, resulting in a larger difference between the inversion and the LCL as the LCL diverges from cloud base. A well-mixed boundary layer would have $z_b$ and LCL measurements which are in close agreement, while a decoupled boundary layer would have a divergence in the similarities between the two values. Previous observations within the VOCALS-REx domain from Jones et al. (2011) found that the boundary layer tended to be decoupled if $\Delta z_b > 150$-m and if $\Delta z_m > 500$-m.

Figure 7 shows the decoupling parameters in panel (b). The average value of $\alpha_\theta$ ($\alpha_q$) are 0.15 (0.07), both which are within the regime of well mixed. During RF11 and RF12, $q$ increases above the inversion leading to large values for $\alpha_q$, while $\Delta\theta$ is relatively small as compared to other flights, with $\alpha_\theta$ being above 0.30 during November $1^{st}$, where Zheng et al. (2011) suggest drizzle processes act to stabilize the boundary layer, leading to decoupling. Panel (c) provides values for $\Delta z_b$ and $\Delta z_m$ for each flight, with average values of 125 and 363-m, respectively. Again, both values are within the regime of well mixed. RF11,13, and 15 are shown to be decoupled, with both $\Delta z_b$ and $\Delta z_m$ at or above the 150 and 500-m threshold values, respectively. RF12 is decoupled according to $\Delta z_m$ only. Looking at raw profiles of $q$ and $\theta$ (not shown here), RF11, 12, 13, and 15 appear to be decoupled due to distinct humidity changes within the sub-cloud profiles, including the presence of a cumulus layer below the Sc deck that is visible from analyzing the LWC profiles (not displayed here) during RF11 (Nov. $8^{th}$). This results in 28% of profiles analyzed being decoupled.

The comparison between Panels (b) and (c) demonstrate that determining decoupling using $\Delta z_b$ and $\Delta z_m$ appears to be more accurate than the decoupling parameters when comparing the results to the raw vertical profiles. A more accurate value for determining decoupling using $\alpha_\theta$ and $\alpha_q$ for the data presented here is 0.20, as compared to the 0.30 stated in Albrecht et al. (1995). A value of 0.20 would lead to better agreement between the two methods. Note that the correlation between $\Delta z_b$ and $\Delta z_m$ is 0.79 (i.e., when the mixed layer cloud thickness increases, the difference between the LCL and cloud base increases). This suggests that when the boundary layer deepens, the cloud layer remains relatively consistent, in agreement with findings from Jones et al. (2011).

## 4 Results

Here, we will quantify the amount of turbulence occurring within the boundary layer. In particular, analysis includes: (1) Analyze day to day variability in turbulent measurements and boundary layer characteristics, relating them to synoptic changes in meteorological conditions; (2) Determine average turbulent values throughout the vertical structure of the STBL, classifying the STBL based on different turbulent profiles analyzed. For each flight analyzed here, the Sc deck lies directly below a strong inversion. This extreme vertical gradient can cause instrument response issues with the measurement of both the dry bulb and dew point temperature for some distance beneath cloud top (Nicholls and Leighton, 1986). Therefore, data collected during both vertical profiles and horizontal legs will be used and compared.

### 4.1 Synoptic Variability of Turbulence

Figure 8 shows the mean surface (30-m horizontal flight leg) LHF (panel (a)), SHF (panel (b)), and Bowen ratio (panel (c)) for each flight day with the standard deviation represented by the gray envelope. Note that for days with two or more mean values, there were two or more 30-m horizontal flight legs, with good agreement between mean leg values within the same flight. The LHF peaks on Oct. $26^{th}$ with a value of 53.3 W m$^{-2}$, and from that point decreases steadily to its minimum values of 19.7 and 18.5 W m$^{-2}$ just as and after the minimum in geopotential height on Nov. $2^{nd}$ and $4^{th}$, respectively. The SHF has a sharp increase to its maximum value of 17.1 W m$^{-2}$ on Nov. $1^{st}$ and decreases to its secondary minimum on Nov. $2^{nd}$ (note that mean values of surface fluxes can be found in Table 3). The Bowen ratio is typically small (less than 0.20), especially for the first half of the campaign. The Bowen ratio has a sharp increase on Nov. $1^{st}$ to match the increase in the SHF (and remains above 0.20 for the remainder of the analysis period), suggesting that the liquid water flux in the cloud layer should not be taken to be proportional to the upward LHF after Nov. $1^{st}$. Note that the average surface values of the LHF and SHF are generally in agreement with those found in Zheng et al. (2011), who found values of 48.5 and 7.1 W m$^{-2}$, respectively. The differences most likely arise due to different averaging techniques.

Figure 9 gives the surface friction velocity (vertical transport of horizontal momentum), vertical velocity variance, TKE, and $\epsilon$ in Panels (a) through (d), respectively. One commonality between each parameter is that the maximum value is reached on Nov. $1^{st}$ followed by the minimum value on Nov. $2^{nd}$ (see Table 3 for the mean and range of the values). For all four variables, there is very little variation between measurements, except for between Oct. $30^{th}$ and Nov. $2^{nd}$, where a large increase in turbulence is observed before a rapid decrease. Overall, there is good agreement between mean values for the same flight, with the exception of Nov. $12^{th}$, which contains the largest difference between mean values for each variable in discussion here. This large difference was not observed however for the surface LHF and SHF.

Shifting focus to the entire depth of the boundary layer, Figure 10 shows boxplots (made up of leg mean values) of sub-cloud (white) and in-cloud (blue) values of LHF (Panel (a)) and buoyancy flux (Panel ((c)). Panels (b) and (d) display histograms of the LHF and buoyancy flux data with normal distribution fits for reference, respectively. The overall LHF was $11.03 \pm 12.97$ Wm$^{-2}$, with the sub-cloud mean being $15.74 \pm 16.4$ Wm$^{-2}$ and the in-cloud mean being $6.01 \pm 3.75$ Wm$^{-2}$. The sub-cloud LHF is clearly offset to larger values, owing to surface evaporation and subsequent transport of moisture. The red dots in Panel

(a) represent the surface values, which are always the largest within the entirety of the vertical layer. The lowest mean values occurred on the same days as the minimum in geopotential height, Nov. $1^{st}$ and $2^{nd}$, with values of 5.51 and 4.67 $\mathrm{Wm}^{-2}$, respectively. Although these two data sets are visually different, statistically speaking they are similar, with a p-value of 0.22 (note that all statistical significance testing will be carried out using the Wilcoxon-Sum-Rank-Test).

The buoyancy flux in Panel (c) displays that the overall mean buoyancy flux was $4.89 \pm 4.86$ $\mathrm{Wm}^{-2}$, with the sub-cloud mean being $4.64 \pm 3.94$ $\mathrm{Wm}^{-2}$ and the in-cloud being $5.12 \pm 5.64$ $\mathrm{Wm}^{-2}$. From just analyzing the mean values of flight legs, there does not appear to be a large difference in the buoyancy flux between the sub-cloud and in-cloud sections of the boundary layer, which is not as expected. In-cloud buoyancy in general is enhanced due to latent heating and cooling effects. There is no statistical significance between the in-cloud and sub-cloud data, with a p-value of 0.39. While the medians in the data populations are similar, the buoyancy flux in-cloud has a much larger range, suggesting isolated occurrences of extremely large buoyancy fluxes within the cloud. Connecting back to concepts discussed in the introduction, the coefficient correlation between the surface LHF and the in-cloud buoyancy is 0.40, suggesting some evidence that a larger surface LHF leads to a larger in-cloud buoyancy flux, as suggested by Bretherton and Wyant (1997) and Lewellen et al. (1996).

Figure 11 provides the same format as that of Figure 10, except for TKE (Panel (a)) and $\epsilon$ (Panel(c)). The total mean TKE was $0.132 \pm 0.03$ $\mathrm{m}^2\mathrm{s}^{-2}$, with a sub-cloud mean of $0.133 \pm 0.05$ $\mathrm{m}^2\mathrm{s}^{-2}$ and an in-cloud mean of $0.132 \pm 0.04$ $\mathrm{m}^2\mathrm{s}^{-2}$. The total mean $\epsilon$ was $3.97 \pm 1.28$ $\mathrm{cm}^2\mathrm{s}^{-3}$, with a sub-cloud mean of $4.14 \pm 2.45$ $\mathrm{cm}^2\mathrm{s}^{-3}$ and an in-cloud mean of $3.80 \pm 1.81$ $\mathrm{cm}^2\mathrm{s}^{-3}$. Overall, very consistent values (when looking at the means) between sub-cloud and in-cloud exist, resulting in statistical similarity between the data populations for both TKE and $\epsilon$. However, in looking at the boxplots, one can see that there are several cases (including Nov. $1^{st}$ and Nov. $2^{nd}$) where the entire turbulent distribution of the sub-cloud data is shifted to larger values than those of in-cloud data, with minimal overlap. This implies that the two layers have limited mixing between them, perhaps due to a more turbulent decoupled lower boundary layer. This will be explored in further detail in Section 4.2. Along with having different turbulent distributions between in-cloud and sub-cloud, both the TKE and the $\epsilon$ had maximum values on Nov $1^{st}$ ($0.163$ $\mathrm{m}^2\mathrm{s}^{-2}$ and $6.13$ $\mathrm{cm}^2\mathrm{s}^{-3}$, respectively) and minimum values on Nov. $2^{nd}$ ($0.065$ $\mathrm{m}^2\mathrm{s}^{-2}$ and $1.30$ $\mathrm{cm}^2\mathrm{s}^{-3}$, respectively).

The analysis to this point clearly shows a maximum in turbulent properties on Nov. $1^{st}$ and a minimum on Nov. $2^{nd}$. This maximum is driven from turbulence below the cloud however, with the in-cloud TKE ($0.128$ $\mathrm{m}^2\mathrm{s}^{-2}$) and $\epsilon$ ($2.78$ $\mathrm{cm}^2\mathrm{s}^{-3}$) being below normal for in-cloud values, where the normal is $0.129$ $\mathrm{m}^2\mathrm{s}^{-2}$ and $3.68$ $\mathrm{cm}^2\mathrm{s}^{-3}$, respectively. Panel (d) shows the total $\epsilon$ distribution for in-cloud and sub-cloud. An increase in in-cloud frequency for $\epsilon$ is clear for the lowest values (first two histogram bars). Eight of the 15 measurements from the first two histogram bars came from RF11 and RF12 (Nov. $1^{st}$ and Nov. $2^{nd}$), which includes all in-cloud values for those flights. The other seven measurements were all sampled above a normalized boundary layer height of 0.90, suggesting entrainment mixing of more laminar flow near the top of the entrainment layer into the upper cloud layer, reducing the turbulent energy.

It is important to analyze turbulent fluxes of energy, momentum, and moisture as they act to determine boundary layer structure and characteristics, along with analyzing how these variables are related to synoptic scale properties such as geopotential height. The correlation coefficients between boundary layer characteristics and synoptic scale properties can be found in Table

4. The 700-hPa geopotential height (i.e., pressure) is fairly correlated with $z_i$, although this correlation is negative with a value of -0.37, suggesting that as the pressure increases, $z_i$ decreases. The rate of change in $z_i$ can be governed by the entrainment rate ($\omega_e$) and $\omega$. If the rate of subsidence increases to the point that it is larger than $\omega_e$ , then $z_i$ will decrease with time. $\omega$ depends primarily on synoptic scale patterns, in particular that of geopotential height. Pressure and $\omega$ have a correlation of -0.89, suggesting that as pressure increases, the subsidence increases (or at the very least, upward vertical motion is dimin-

ished). Entrainment on the other hand, can depend on multiple variables including the inversion layer thickness, wind shear, and surface fluxes. Increases in $\omega_e$ result in a higher LCL for the entrained air and a resulting increase in boundary layer height as a result. Given that $z_i$ acts to decrease as the pressure increases, this suggests that the subsidence becomes the dominating component that governs $z_i$ over that of entrainment.

The surface LHF provides the main source of moisture in the STBL, which in turn is an important source of buoyant TKE
production. An enhanced (reduced) LHF will generate thicker (thinner) clouds with larger (smaller) LWC values, resulting in enhanced (reduced) evaporative cooling near cloud top leading to enhanced (reduced) buoyancy driven entrainment, and a subsequent deepening (thinning) of the boundary layer. This process is demonstrated well when analyzing the correlation coefficients. Both the LHF and SHF are positively correlated with $z_i$ (correlation coefficients of 0.36 and 0.44, respectively) while the LHF is negatively correlated with the Sc cloud thickness (correlation coefficients of -0.50). Therefore, a larger LHF
tends to result in a thinner Sc cloud layer but a larger $z_i$, suggesting enhanced entrainment which acts to thin the cloud layer while deepening the boundary layer. It should also be noted that the correlation between the SHF and wind speed is significant, as anticipated since the SHF is expected to increases linearly with wind speed (Palm et al., 1999).

Both TKE and $\epsilon$ increase in-cloud with respect to pressure (correlation coefficient of 0.23 and 0.24, respectively) and decreases with respect to $z_i$ (-0.32 and -0.34, respectively). The observed decrease in boundary layer turbulence with increasing $z_i$
is due to decoupling and an inability for the entire boundary layer to be mixed (leading to a subsequent decrease in turbulence), while a shallow boundary layer can be easily mixed through cooling near cloud top.

As the cloud droplet number concentration and aerosol number concentration increase (accompanied by a decrease in average droplet size), the TKE and $\epsilon$ increase (with correlation coefficients of 0.35, 0.42, and -0.32 in relation to TKE, respectively). Physically this makes sense. As precipitation is suppressed due to larger number concentrations and smaller droplet sizes, a
reduced moisture loss from the STBL can result, leading to thicker clouds, a larger buoyancy flux, and a larger TKE. Smaller droplets will also evaporate more readily, leading to enhanced latent heating effects and a resultant increase in turbulence.

### 4.2    Vertical Profiles

It has been shown through the boundary layer vertical structure in Figure 6 that the boundary layer is, on average, well mixed when considering thermodynamic variables. Figure 12 represents vertical profiles of the buoyancy flux (Panel (a)), LHF (Panel
(b)), vertical velocity variance (Panel(c)), and TKE (Panel (d)), where each dot represents a leg mean value, with in-cloud values in red and values measured during Nov. $1^{st}$ and Nov. $2^{nd}$ in blue. The buoyancy flux in the sub-cloud layer (on average) varied between -2 and 20 $\mathrm{Wm}^{-2}$ and decreased with height until increasing within the cloud layer with values ranging between -5 and 43 $\mathrm{Wm}^{-2}$. The standard deviation (in orange) was produced using data from vertical flight profiles as opposed to the

horizontal legs due to data uniformity throughout the boundary layer depth. The buoyancy flux has a clear increase in variance
within the cloud layer. The LHF peaks near the surface, ranging between -1 and 55 Wm$^{-2}$ below the cloud layer and generally
decreases with height. The variance peak of 33 Wm$^{-2}$ occurs at $z/z_i = 0.99$, signifying the large gradient in $q$ near $z_i$ and the
variation in evaporative cooling due to entrainment mixing at cloud top between flight days.

Vertical velocity variance (from here on $w'w'$) ranged from 0.008 to 0.20 m$^2$s$^{-2}$. The observed in-cloud $w'w'$ at Point Alpha
was 0.105 m$^2$s$^{-2}$ with values fluctuating considerably more than those in the sub-cloud layer (in agreement with findings from
Bretherton et al. (2010), who measured a larger standard deviation in vertical velocity in-cloud vs. sub-cloud). The average
in-cloud value of $w'w'$ found here is significantly lower than what was found over more remote ocean areas (80°W - 85°W,
20°S) of 0.36 m$^2$s$^{-2}$ (Bretherton et al., 2010). Nocturnal measurements of the Californian Sc deck during DYCOMS-II also
revealed a stronger turbulent structure than that measured at Point Alpha, with observations showing in-cloud $w'w'$ larger than
0.4 m$^2$s$^{-2}$ with a maximum of 0.5 m$^2$s$^{-2}$ near the base of the Sc deck (Stevens et al., 2005). As discussed in Wood (2012),
$w'w'$ is typically more vigorous at night due to the buoyancy production being larger from the lack of shortwave radiation
absorption, which acts to stabilize the layer. As is found here, Hignett (1991); Nicholls (1984); Ghate et al. (2014) also found
that $w'w'$ peaked in the upper half of the STBL away from any boundaries such as cloud top. Note that the TKE mirrors that
of $w'w'$ in terms of vertical spatial tendencies.

Considering data collected during aircraft soundings (as opposed to mean values of horizontal flight legs), u-variance ($u'u'$),
v-variance ($v'v'$), $w'w'$, and the TKE are displayed in Figure 13 Panels (a) through (d), respectively, with the red line repre-
senting the mean profile and each gray line representing individual flight profiles. The blue lines represent flight profiles for
Nov. 1$^{st}$ and Nov. 2$^{nd}$. Panel (e) displays the mean values from each of Panels (a) through (d). The profile of each variable
in question shows a near constant value below cloud base, with an increase in-cloud before beginning to decrease near cloud
top. Both $w'w'$ and TKE reach their peak values at $z/z_i = 0.88$ (or a normalized in-cloud location of 0.40). Simulations and
observations from Pasquier and Jonas (1998) of in-cloud TKE showed that the maximum TKE occurred in two locations, near
cloud top and near cloud base, suggesting that turbulence is being generated through two processes: (1) Cooling at or near
cloud top (through evaporation or longwave cooling), resulting in cool, dry downdrafts; (2) Warming near cloud base from the
release of latent heat through condensation, resulting in positively buoyant updrafts. However, no conclusions can be made
here on whether or not there are two sources of TKE due to the low vertical resolution of the mean values (i.e., averaging over
14 flight profiles). TKE values plummet above the inversion due to the dominance of clear, stable, and subsiding air aloft. The
overall maximum in TKE measured (for all 14 flights) is found near $z/zi = 0.60$ (looking at the blue profile line in Figure 13,
Panel (d)) during RF11 (Nov. 1$^{st}$). This will be discussed in more detail in Section 4.3.

Looking at individual profiles of TKE, (not shown here), only six of the fourteen flights have a maximum TKE within the
cloud layer. Modeling and observations of boundary layer profiles of turbulence from Pasquier and Jonas (1998) showed that
mixing and overturning of the boundary layer profile due to buoyancy effects leads to a maximum in turbulence commonly
being reached in the sub-cloud layer. Seven of the fourteen flights display two peaks in TKE within the cloud layer, one near
cloud base and another near cloud top, signifying evaporative cooling near cloud top and latent heating near cloud base. Of the
six flights that have a maximum TKE within the cloud layer, all six display two peaks in the TKE within the cloud layer, one

near cloud base and one near cloud top. Having the maximum in TKE in the sub-cloud layer can signify decoupling (Durand and Bourcy, 2001). A slight decoupling can lead to less moisture transport into the Sc layer, resulting in less latent heat release due to condensation. This could be why only one flight has two peaks in TKE within the cloud when the turbulence maximum is reached below cloud, due to latent heat release at cloud base being suppressed.

Figure 14 provides the same format as Figure 13, except for values of buoyancy flux (Panel (a)), LHF (Panel (b)), vertical velocity skewness (Panel (c)), and the cloud droplet number flux (Panel (d)). Note that Figure 14 displays the range of data in the gray envelope, as opposed to showing each individual profile with a single gray line. Both the buoyancy flux and the droplet number concentration flux (from here on $w'N'$) have maximum values at $z/z_i = 0.93$ (normalized in-cloud height of 0.59). The peak near cloud middle is due to a combination of the warm/moist updrafts and cool/dry downdrafts meeting, formed by evaporative cooling at cloud top and latent heating near cloud base. The same concept can be extended to $w'N'$, where droplets are activating near cloud base while evaporating near cloud top, suggesting that in the lower cloud the cloud droplet number concentration increases with updrafts (condensation), while the cloud droplet number concentration decreases with downdrafts (evaporation) in the upper cloud region. According to Pasquier and Jonas (1998), the buoyancy flux should reach a minimum near cloud top from the entrainment of warm, dry air down into the cloud layer. Although the mean profile does not show a decrease at cloud top, the raw data (i.e., unsmoothed) does show a negative buoyancy flux at cloud top. For individual flights, only RF11 (Nov $1^{st}$) had a maximum in the buoyancy flux in the sub-cloud layer. The LHF peaks at the surface, but also sees a secondary maximum at $z/z_i = 0.99$. The maximum at cloud top can be attributed to the strong $q$ gradient and to entrainment of drier air down into the cloud (i.e., also a positive flux since both $w'$ and $q'$ are negative).

Well-mixed STBLs tend to show characteristics of downdrafts that are spatially smaller, but stronger, than updrafts. This results in a negative vertical velocity skewness (from here on $w'w'w'$) through most of the cloud and sub-cloud layer (Nicholls, 1989; Hogan et al., 2009; Ghate et al., 2014). Panel (c) displays that $w'w'w'$ on average is negative throughout the cloud layer and through most of the sub-cloud layer, having a maximum value near the surface. The minimum values in $w'w'w'$ occurs at cloud base (normalized in-cloud value of 0.04), suggesting that overall, the downdrafts are smallest, yet strongest at cloud base while updrafts are spatially larger, yet weaker.

Figure 15 shows the average buoyancy flux (Panel (a)), LHF (Panel (b)), TKE (Panel (c)), and $\epsilon$ (Panel (d)) averaged over all flights for sub-cloud and for different layers within the cloud. Each black dot represents the average value for individual flights using horizontal leg averages. The blue dots represent mean values using horizontal flight legs, while the red dots represent mean values using flight vertical profile data. Values for in-cloud are calculated for layers between normalized in-cloud height values of 0-0.25 (cloud base), 0.25-0.50 (bottom-middle), 0.50-0.75 (top-middle), and 0.75-1.0 (cloud top). Table 5 summarizes the mean values for each layer. A clear difference in values and trends can be seen between sampling methods. For example, looking at the LHF in Panel (b), we see that the horizontal leg sampling correctly captures the larger LHF at the surface due to evaporation from the ocean surface, whereas the profile samples do not capture this increase at the surface (the profile data is terminated at the start of the 30-m horizontal flight legs, meaning there is limited samples near the surface for the profile method). Conversely, the profile method observed a large increase in TKE at cloud top from evaporative cooling due to entrainment mixing, which is not observed in the horizontal leg method. Another example is the buoyancy flux, which is

seen to have a large increase in-cloud as compared to sub-cloud using the profile method. The horizontal leg method displays a maximum in the top-middle region of the cloud, but the overall buoyancy flux increase in-cloud vs. sub-cloud is compressed as compared to the profile method. In analyzing Table 5, it is clear that the average turbulence (both TKE and $\epsilon$) peaks either in the bottom-middle or top-middle of the cloud (i.e., between a normalized in-cloud height of 0.25-0.75). This is also the two layers in which the buoyancy flux is at a maximum. TKE production near cloud base from latent heat release moves up through the cloud layer, while TKE production near cloud top from evaporative cooling moves down through the cloud layer, resulting in a maximum within the middle of the cloud.

Panels (e) and (f) represent the $u$, $v$, and $w$ components of the TKE and $\epsilon$, respectively. The anisotropic conditions present within the turbulent boundary flow can clearly be seen due to the differing values in each component. Although the $u$ and $v$ components are similar for most layers, differences are evident in the $w$-component. If the flow was perfectly isotropic, one would expect the same values for each component of the TKE and $\epsilon$.

## 4.3  RF 11 (November 1st)

Turbulent and boundary layer characteristics have been shown to be abnormal on Nov. $1^{st}$, with a minimum in 500-hPa geopotential height, aerosol number concentration, and cloud droplet number concentration. November $1^{st}$ also had overall mean maximum values of TKE and $\epsilon$ within the sub-cloud layer, along with maximum values in the surface SHF and in-cloud drizzle rate.The average drizzle rate in-cloud on November $1^{st}$ was the largest recorded (a mean in-cloud drizzle water content of 0.025 gm$^{-3}$ measured by the CIP probe) and roughly 4.5 times that of the second largest in-cloud average recorded on November $2^{nd}$ (0.0055 gm$^{-3}$), where the average for all other flights was 0.0014 gm$^{-3}$. A moist layer is present above the boundary layer from looking at profiles of $q$ in Figure 6, leading to the secondary maximum in LWC and cloud thickness (November $2^{nd}$ had the largest cloud thickness and LWC). Also, visible in Figure 4 is the presence of wind shear near $z/z_i = 0.60$.

In order to explore this case further, Figure 16 shows profiles of multiple thermodynamic and turbulent variables as a function of $z/z_i$. Panel (a) shows profiles of $\theta$ (blue), LWC (black), and $q$ (red). The gray envelope represents the cloud layer, while the orange envelopes represent areas in the sub-cloud layer where the SHF is negative and TKE and $\epsilon$ are enhanced. The potential temperature at the base of the lowest orange envelope begins to deviate from its surface value, decreasing significantly. Normalizing $\theta$ from 0 to 1 (where the surface is 0 (the minimum temperature) and the top is 1 (the maximum temperature), we find that the value of $\theta$ is 0.32 at cloud top and 0.10 at cloud base, inferring significant entrainment of the warmer, less buoyant air aloft. However, $q$ within the boundary layer stays relatively constant. This is due to the fact that the entrainment of the warmer air aloft has a larger $q$ than that near the surface of the boundary layer. Significant decoupling is occurring in the sub-cloud layer, near $z/z_i = 0.60$ (where the largest TKE and $\epsilon$ are located) and 0.40 (secondary maximum in the TKE and $\epsilon$). It is suggested here that precipitation acts to decouple the boundary layer and enhance sub-cloud turbulence due to evaporative cooling of precipitation from the Sc deck above. Zheng et al. (2011) states that the cloud liquid water path reached a maximum on Nov. $1^{st}$ and Nov. $2^{nd}$ due to the total-water specific humidity above the inversion being larger than that within

the boundary layer. The inversion strength became significantly weaker on these two days (as evident from Figure 7) and the boundary layer was decoupled due to drizzle.

Several variables must be considered here. First, the moist layer above the Sc deck can have two effects, including (1) changing the radiative balance at cloud top through increased downwelling longwave radiation (Christensen et al., 2013) and (2) Entrainment of more moist air near cloud top, reducing evaporational cooling (Eastman et al., 2017). Both effects act to reduce cooling (both evaporational and radiational) near cloud top and slows the rate of boundary layer deepening through decreases in entrainment. Eastman and Wood (2018) found that high humidity above the Sc deck acts to slow boundary layer deepening while the entrainment of increased water vapor into the boundary layer results in enhanced cloud cover.

Second, drizzle can have multiple effects on boundary layer structure, including (1) precipitation removes liquid water from the Sc deck, resulting in cloud thinning if the LHF is not large enough to maintain the Sc deck (Austin et al., 1995); (2) Warming of the drizzle producing cloud layer occurs through latent heating, acting to stabilize the cloud layer; (3) Changing the stability of the sub-cloud layer depending on the rate of precipitation. Significant proportions of precipitation are known to evaporate before reaching the surface (Comstock et al., 2004; Wood et al., 2015; Zhou et al., 2015). The profile of sub-cloud evaporation determines whether the layer will become more or less unstable. When precipitation is heavier and in the form of large drops it tends to stabilize the boundary layer from evaporational cooling spread over the depth of the sub-cloud layer, with substantial evaporation near the surface stabilizing the boundary layer. When precipitation is lighter and in the form of small drops, cooling persists in the uppermost part of the sub-cloud region, resulting in destabilization of the sub-cloud layer (Feingold et al., 1996; Wood, 2005; Mechem et al., 2012; Rapp, 2016; Ghate and Cadeddu, 2019; Wood, 2012).

Here, precipitation promotes STBL decoupling by reducing the diabatic cooling in the cloud layer through in-cloud latent heating effects resulting in a stabilization of the cloud layer (where the average in-cloud turbulence is the $4^{th}$ lowest measured on Nov. $1^{st}$ and lowest measured on Nov. $2^{nd}$, see Figure 11). The sub-cloud evaporation leads to cooling below cloud and a resultant local minimum in the buoyancy flux is created (Bretherton and Wyant, 1997). It is known from Wood (2005) that evaporative cooling shows cooler and more moist characteristics than that of non-precipitating regions.The SHF is observed to be negative from $z/z_i \sim 0.4$ up to cloud base, with the minimum and local minimum outlined in the orange envelopes. The LHF is also shown to be slightly enhanced within these regions. This suggests that evaporational cooling is occurring away from the surface, resulting in the largest average turbulence being measured in the sub-cloud layer on this day due to sub-cloud destabilization. From earlier, it was mentioned that Zheng et al. (2011) suggested drizzle processes acted to stabilize the boundary layer, leading to decoupling. This is partially true, as the precipitation does lead to decoupling, however, the precipitation actually destabilized the sub-cloud layer while stabilizing the cloud layer.

Normally, this process will result in the cloud layer being decoupled form the surface moisture source, leading to a thinning cloud layer. However, the Sc deck is receiving moisture from the upper atmosphere (as seen in the negative LHF above cloud, where $w'$ is negative but $q'$ is positive). This process acts to moisten the boundary layer, which will lower the LCL, and assuming that $z_i$ does not change, this will thicken the cloud (Randall, 1984). Note that the cloud layer on Nov. $2^{nd}$ is thicker than that on Nov. $1^{st}$ by roughly 100 m, while $z_i$ is roughly 50-m lower.

Looking at $w'N'$, an increase occurs near cloud base up to the middle region of the cloud, before decreasing to negative values in the upper half of the cloud. The positive values near cloud base occur due to droplet activation through condensation, while the negative values occur in the upper half of the cloud from upward vertical velocity perturbations having less droplets than that of negative vertical velocity perturbations, suggesting that droplet activation may be occurring near cloud top as well. $w'w'w'$ also varies between positive and negative values within the sub-cloud layer, providing more evidence that decoupling is occurring.

To summarize, it appears that the sub-cloud layer is decoupled from the Sc deck due to the evaporative cooling of precipitation. This increases turbulence within the sub-cloud layer while reducing turbulence in the cloud layer. However, the cloud layer is still supplied with moisture through the entrainment of the more moist air aloft, driving cloud deepening and sustaining the Sc deck. The wind direction shifts from the south in the lower portion of the boundary layer to from the north near $z/z_i = 0.60$. The fact that the free atmosphere wind direction extends into the sub-cloud layer indicates that significant entrainment mixing has occurred, resulting in the upper $40\%$ of the boundary layer to share characteristics with the free atmosphere (whereas Zheng et al. (2011) attribute this to an upsidence wave). Note that the maximum value in TKE that is measured on Nov. $1^{st}$ at $z/z_i = 0.60$ (see the blue profile line in Figure 12) matches the location at which the wind shear is occurring. However, this spike in TKE cannot be attributed to the wind shear alone, as wind shear that occurs at the inversion for each flight day and within the boundary layer on Nov. $4^{th}$ do not result in large increases in turbulence. The increase in turbulence seen on Nov. $1^{st}$ is related to latent heating affects and the resulting changes in the buoyancy fluxes.

Although not displayed here, profiles for Nov. $2^{nd}$ (the day with the lowest average turbulence, both in-cloud and sub-cloud) shows a very consistent turbulent profile (no large spikes within or below the cloud layer). It is suggested here that between Nov. $1^{st}$ and $2^{nd}$ one of two things occurred, either (1) Precipitation stopped (i.e., the source of instability in the sub-cloud layer) and enhanced turbulent mixing of the sub-cloud layer ceased (while the cloud layer continued to deepen from the entrainment of more moist air reducing the LCL) or (2) Precipitation continued to occur, leading to evaporation near the surface and a stabilization of the entire boundary layer. Note that although in-cloud drizzle is occurring on Nov. $2^{nd}$, there is no evidence of sub-cloud evaporation. There are limited sources of turbulent production until dryer air moves in and enhanced entrainment cooling near cloud top can resume mixing of the boundary layer, or if precipitation restarts and acts to destabilize the sub-cloud layer.

Comparing RF11 to a well-mixed boundary layer, Figure 17 provides the same format as that of Figure 16, except for RF03 (Oct. $19^{th}$). Both $\theta$ and $q$ appear to be well-mixed throughout the boundary layer, with a slight decrease in $\theta$ throughout the cloud layer. TKE, $\epsilon$, the LHF, and the SHF all have two peaks near cloud base and cloud top, suggesting latent heating near cloud base and evaporative cooling near cloud top. The SHF also has a negative value above cloud top due to the entrainment of warm, dry air down into the cloud. The droplet number concentration flux increases near cloud base owing to droplet activation, and sees a sharp decrease near a normalized in-cloud height of 0.50, suggesting most of the activation is occurring in the bottom half of the cloud layer. The vertical velocity skewness has a maximum negative value near cloud base, and never has an increase to positive values. The negative TKE flux within the cloud layer suggest that upward moving air is transporting less TKE than that of downward moving air.

## 5 Conclusion

Variations in turbulent and meteorological properties within the boundary layer on a flight by flight basis (synoptic variation) have been examined. It has been shown that the influence of a synoptic system on Nov. $1^{st}$ and Nov. $2^{nd}$ leads to a deepening of the cloud layer during passage from a moist layer directly above the boundary layer. A large increase in $z_i$ is observed after passage. Although the pressure is increasing (and subsidence becomes stronger) after the passage of the synoptic system, it is proposed that the moist layer above the boundary layer limits $z_i$ deepening due to reduced evaporational and radiational cooling near cloud top, limiting entrainment (counteracting the fact that subsidence is weaker). As the synoptic system passes and the upper atmosphere dries, cloud top cooling is enhanced and entrainment acts to deepen $z_i$, counteracting the fact that subsidence is increasing. Turbulence is shown to be rather weak as compared to other observational studies of Sc decks. TKE is shown to vary around 0.13 m$^2$s$^{-2}$, except on the days leading up to and following the synoptic system passage, where the TKE increases rapidly to a maximum on Nov. $1^{st}$ due to precipitation leading to enhanced turbulence in the sub-cloud layer and then decreases significantly to a minimum on Nov. $2^{nd}$. Analysis over the observation period indicates:

- As the pressure decreases (increases), $z_i$ increases (decreases), accompanied by a decrease (increase) in turbulence within the boundary layer. As $z_i$ deepens, cooling near cloud top cannot sustain mixing over the entire depth of the boundary layer, resulting in less turbulence.

- As the LHF and SHF increases (decreases), $z_i$ increases (decreases). When the LHF increases however, the cloud thickness decreases (increases). A larger LHF tends to produce thinner Sc clouds but a larger $z_i$, suggesting enhanced entrainment which acts to thin the cloud layer while deepening the boundary layer.

- A maximum in TKE on Nov. $1^{st}$ (both overall average and largest single value measured) is due to precipitation acting to destabilize the sub-cloud layer (through evaporation away from the surface), while acting to stabilize the cloud layer. This is observed in both the vertical profiles of RF11 and the TKE and $\epsilon$ values in Figure 11, where it is shown that the distributions of turbulence for the sub-cloud and cloud layer are completely offset from one another, with the TKE in the sub-cloud layer maximizing for the analysis period, while the TKE in the cloud layer is below the average value for the analysis period. Nov. $2^{nd}$ has the lowest average turbulence measured (both in-cloud and sub-cloud), and is believed to be a result of (1) lack of cooling near cloud top due to the enhanced moist layer above and (2) Heavy precipitation from the previous day (or sometime prior to the measurements being made) leading to evaporation through the entire sub-cloud layer, stabilizing it.

- Six of the fourteen flights have a maximum TKE within the cloud layer. Seven of the fourteen flights display two peaks in TKE within the cloud layer, one near cloud base and another near cloud top, signifying evaporative cooling near cloud top and latent heating near cloud base. Of the six flights that have a maximum TKE within the cloud layer, all six display two peaks in the TKE within the cloud layer, one near cloud base and one near cloud top. This suggests that enhanced turbulence below the cloud can act to reduce latent heating and cooling effects within the cloud layer which generate turbulence near cloud top and bottom. Perhaps, enhanced sub-cloud turbulence (as compared to in-cloud) could be an

initial indicator that the process of boundary layer decoupling has begun, but has not developed to the point that classical measurement techniques of decoupling (like those discussed in Section 3.3) can measure decoupling yet.

– Analyzing different layers of turbulence over the 14 flights shows that TKE, $\epsilon$, and the buoyancy flux, on average, all reach maximum values near cloud middle (between normalized in-cloud values of 0.25- 0.75).

The results presented here represent a snapshot of data through 14 aircraft flights, with at least a day between any two flights. Therefore, the results presented represent boundary layer conditions that were present at the time of measurement, limiting any analysis of continuously evolving boundary layer and turbulent conditions. For example, being able to analyze the changing thermodynamic and dynamic conditions that resulted in large turbulent changes between Nov. $1^{st}$ and Nov. $2^{nd}$ would be ideal, especially since multiple papers have called for observational studies to assess the impact of drizzle evaporation induced cooling on boundary layer turbulence (Wood et al., 2016; Zheng et al., 2016, 2017). It has also been displayed that how turbulence is analyzed is important to understanding the true extent of how turbulence varies within the boundary layer. Taking large scale averages of turbulent parameters (such as over entire horizontal flight legs) may lead to important smaller resolution variations being averaged out. For example, the vertical profiles presented in Figures 16 and 17 show much more detail in the vertical trends as compared to the averaged results of horizontal leg means displayed in Figure 12.

*Data availability.* All cabin data from different aircraft platforms can be found on the VOCALS-Rex website at https://archive.eol.ucar.edu/projects/vocals/rex.html (last access: 02 February 2020). All NCEP/NCAR reanalysis data can be found from NOAA at https://www.esrl.noaa.gov/psd/data/gridded/data.ncep.reanalysis.html.

*Author contributions.* DSD and JDSG contributed equally to both the analysis and the writing of this paper.

*Competing interests.* There are no competing interests to declare

*Acknowledgements.* We thank the CIRPAS Twin Otter crew and personnel, including pilots Mike Hubble and Chris McGuire, for their effort and support during the field program, along with any individual who contributed to the planning and execution of VOCALS-Rex. This work was funded by NASA NESSF grant 80NSSC18K1406.

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

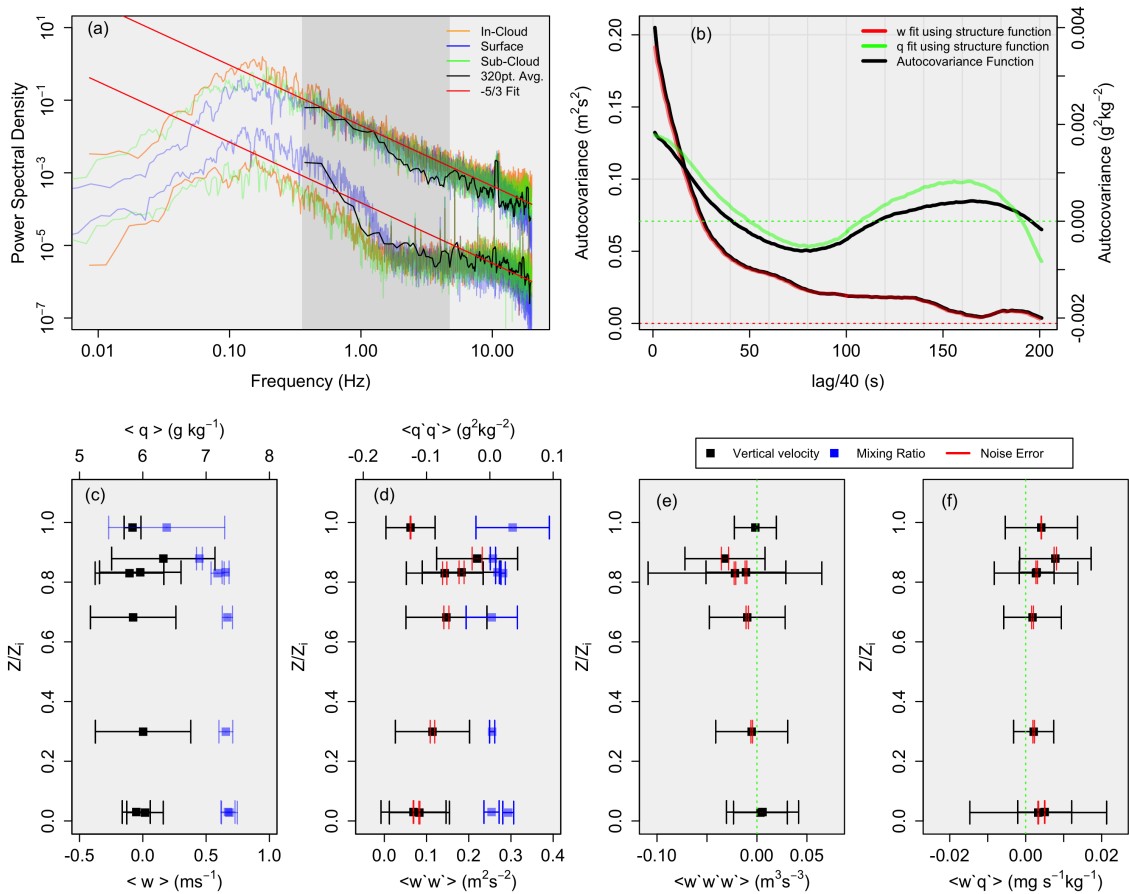

**Figure 1.** All data presented is from RF3. Panel (a): Power spectral density for three different horizontal flight legs, including in-cloud (orange), sub-cloud (green), and near surface (blue) fit with a -5/3 power law (red), where the upper fit is vertical velocity ($m^2s^{-1}$), and the lower fit is $q$ ($g^2kg^{-2}$). The black specra overlaid represents a single specra using a 320-point average. The light gray envelope represents the 0.3 to 5-Hz range; Panel (b): Autocovariacne functions of vertical velocity and $q$ (black) with the fit structure function (green for $q$ and red for vertical velocity); Panel (c): leg mean vertical velocity (black) and $q$ (blue), where the error bars represent the square root of the total variance; Panel (d): As in Panel (c), except for the variance. Note that red error bars represent the noise error, while the remaining error bars represent the sampling error; Panel (e): As in Panel (d), except for vertical velocity skewness; Panel (f); As in Panel (d), except for the flux $w' q'$.

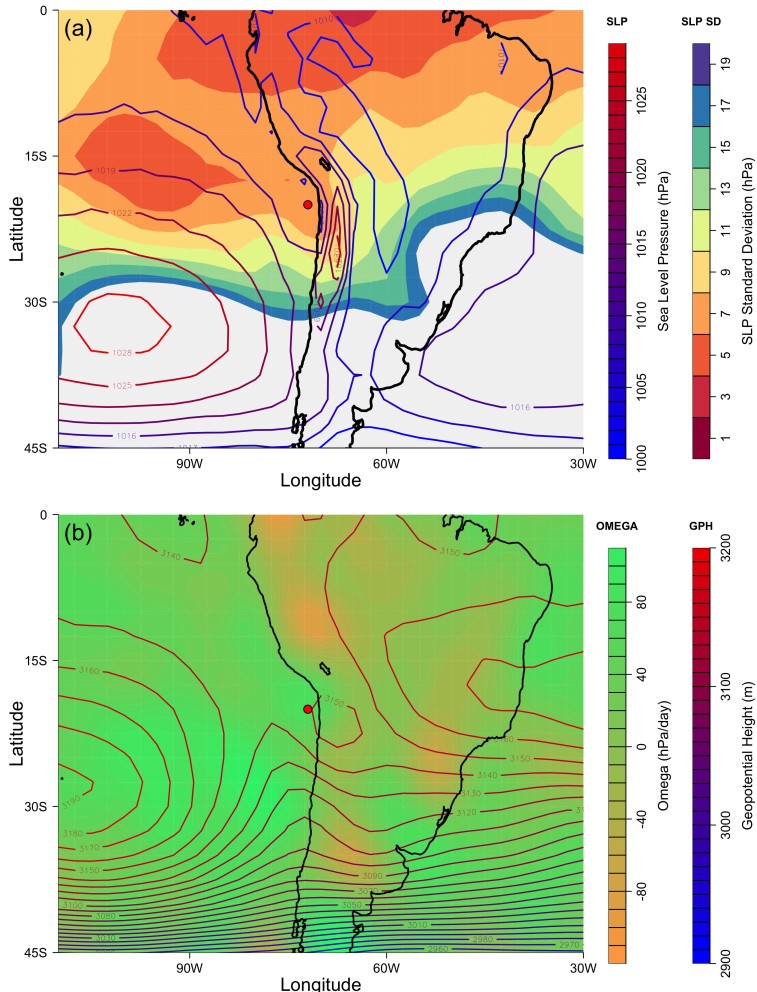

**Figure 2.** Panel (a): Mean sea level pressure (hPa) between Oct. $19^{th}$ and Nov. $12^{th}$ with the standard deviation overlaid. Note that the contours are every 3-hPa; Panel(b): Mean 700-hPa geopotential height with mean omega ($\omega$, hPa/day) overlaid at the same level. Contours are every 10-m. The red dot in both panels represents the location of Point Alpha.

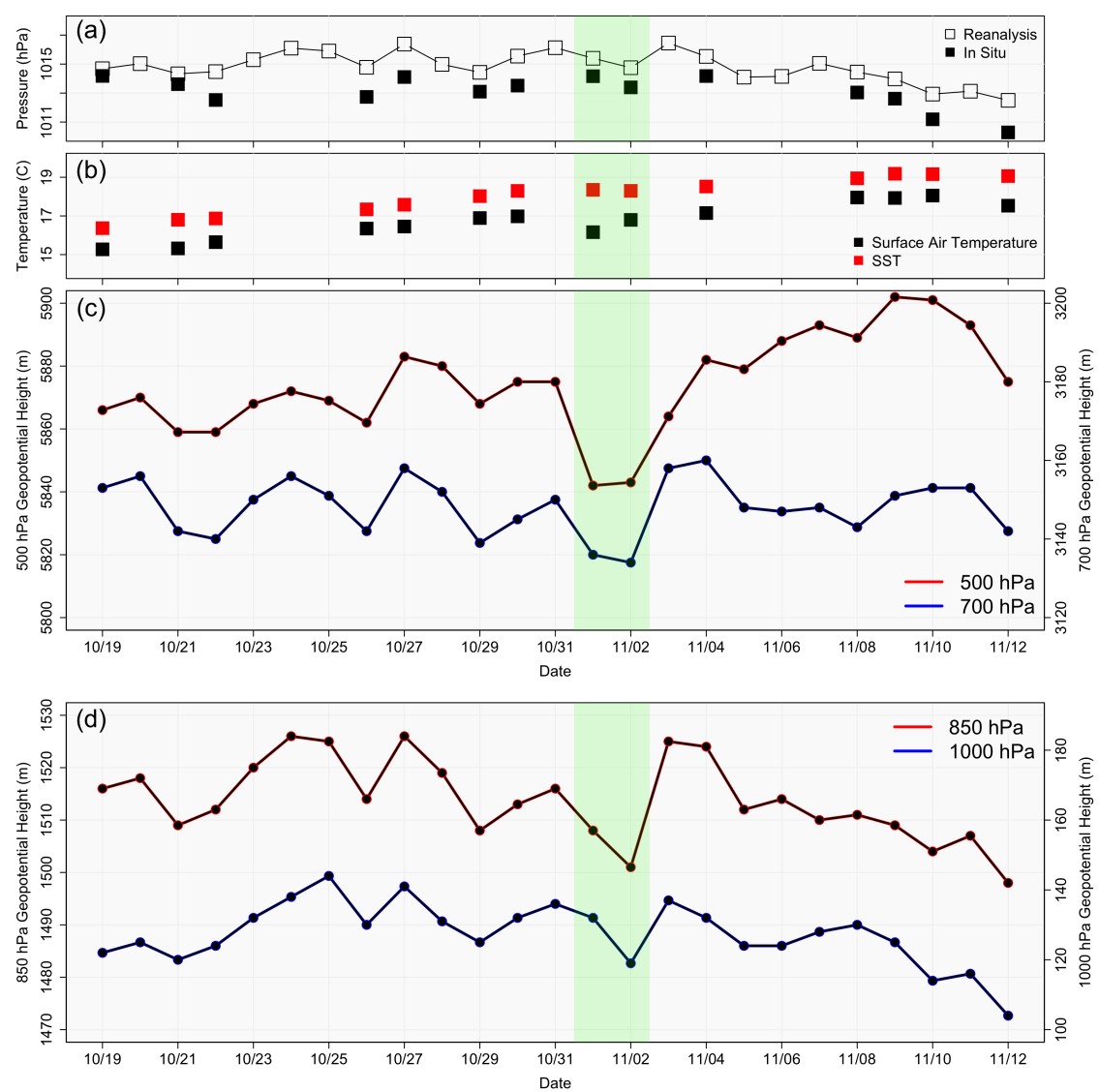

**Figure 3.** Panel (a): NCEP/NCAR reanalysis (open squares) and flight data (solid squares); Panel (b): Sea surface temperature and atmospheric surface temperature collected from flight data; Panel (c): 500-hPa (red) and 700-hPa (blue) geopotential height data from NCEP/NCAR reanalysis data; Panel (d): As in panel (c), except for 850-hPa (red) and 1000-hPa (blue).

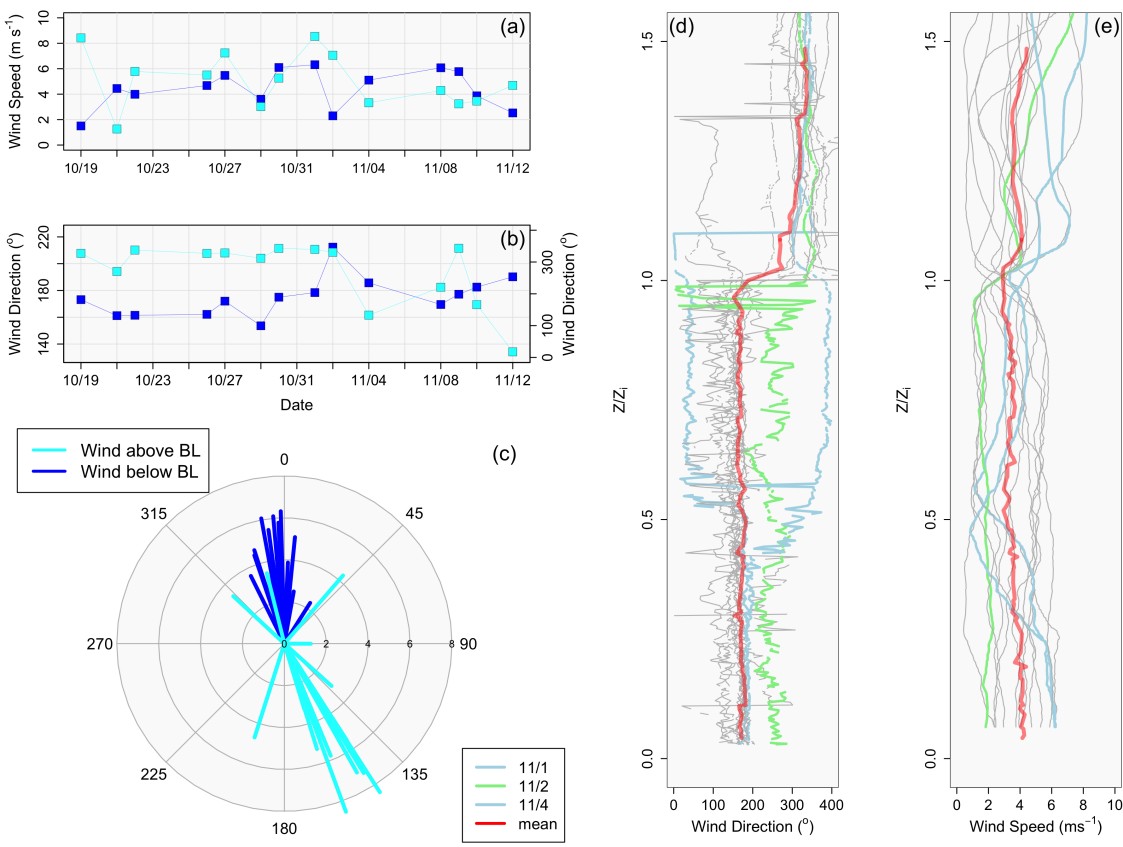

**Figure 4.** Panel(a): Wind speed ($ms^{-1}$) at the surface collected during 30-m horizontal flight legs (dark blue) and above the inversion collected during horizontal flight legs above the boundary layer (light blue); Panel (b): As in Panel (a), except for wind direction (degree); Panel (c): Vectors showing wind direction from panel (b); Panel (d): Vertical profiles (collected during aircraft soundings) of wind direction for each flight day plotted vs. normalized boundary layer height. Nov. $1^{st}$ and $4^{th}$ are displayed in light blue, Nov. $2^{nd}$ is green, and the mean wind direction is represented by red; Panel (e): As in Panel (d), except for wind speed ($ms^{-1}$).

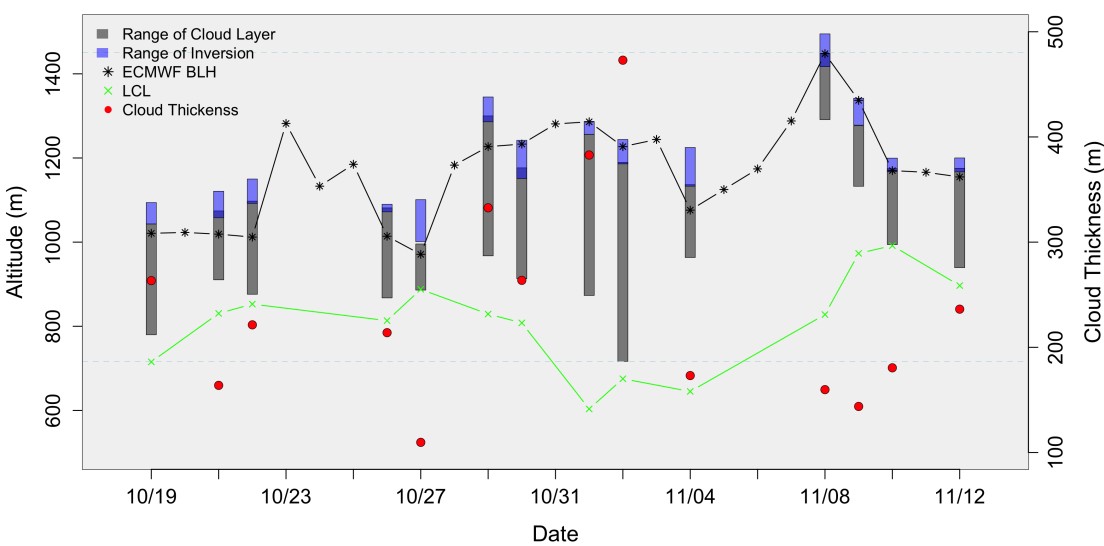

**Figure 5.** Shows the range of the cloud (gray) and inversion (blue) layer as a function of altitude for each RF. The top of each gray profile represents cloud top and $z_i$. The bottom of each gray profile represents cloud base. Cloud thickness (represented as a single value) is represented by each red dot (right y-axis). The LCL and ECMWF-$z_i$ are provided with the black star and green x, repsectively.

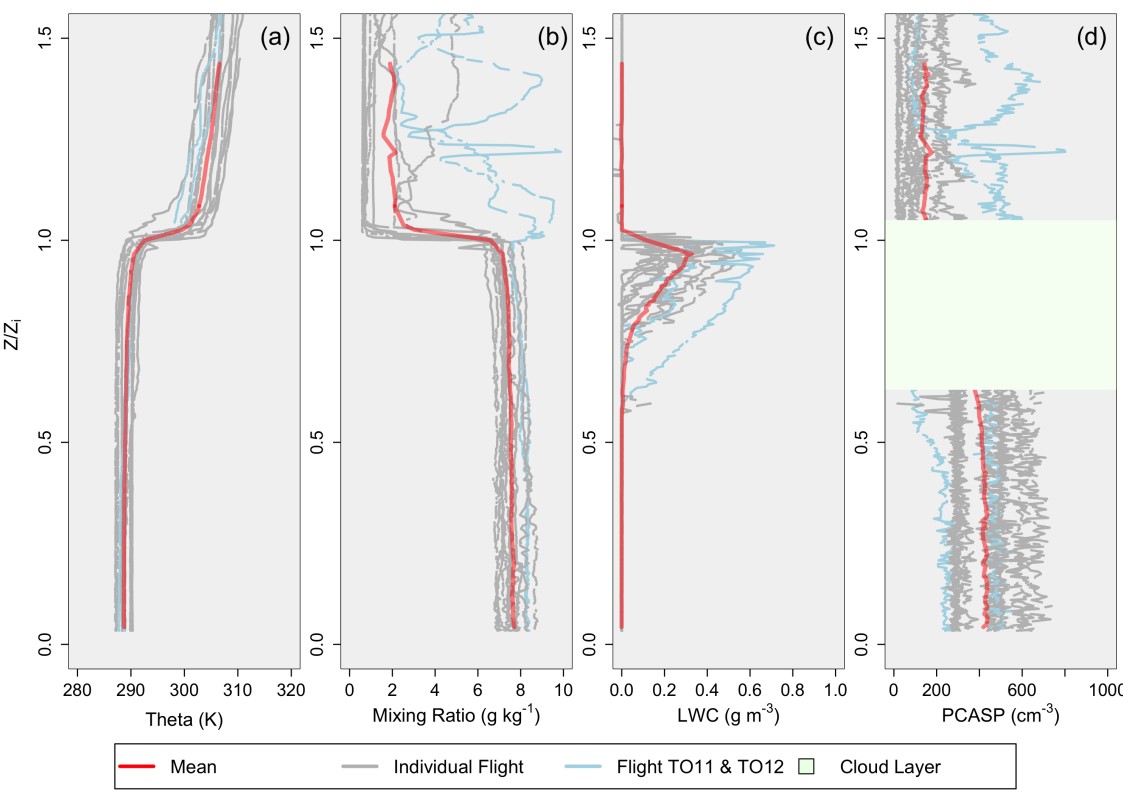

**Figure 6.** Profiles scaled by $z_i$. (a) $\theta$ (K); (b) $q$ (g kg$^{-1}$); (c) LWC (g m$^{-3}$); (d) Aerosol number concentration (cm$^{-3}$). The red profile represents the mean value, and the two blue profiles represent RF 11 and RF 12. The green layer represents the relative cloud layer for panel (d), as aerosol data cannot be collected in the cloud layer.

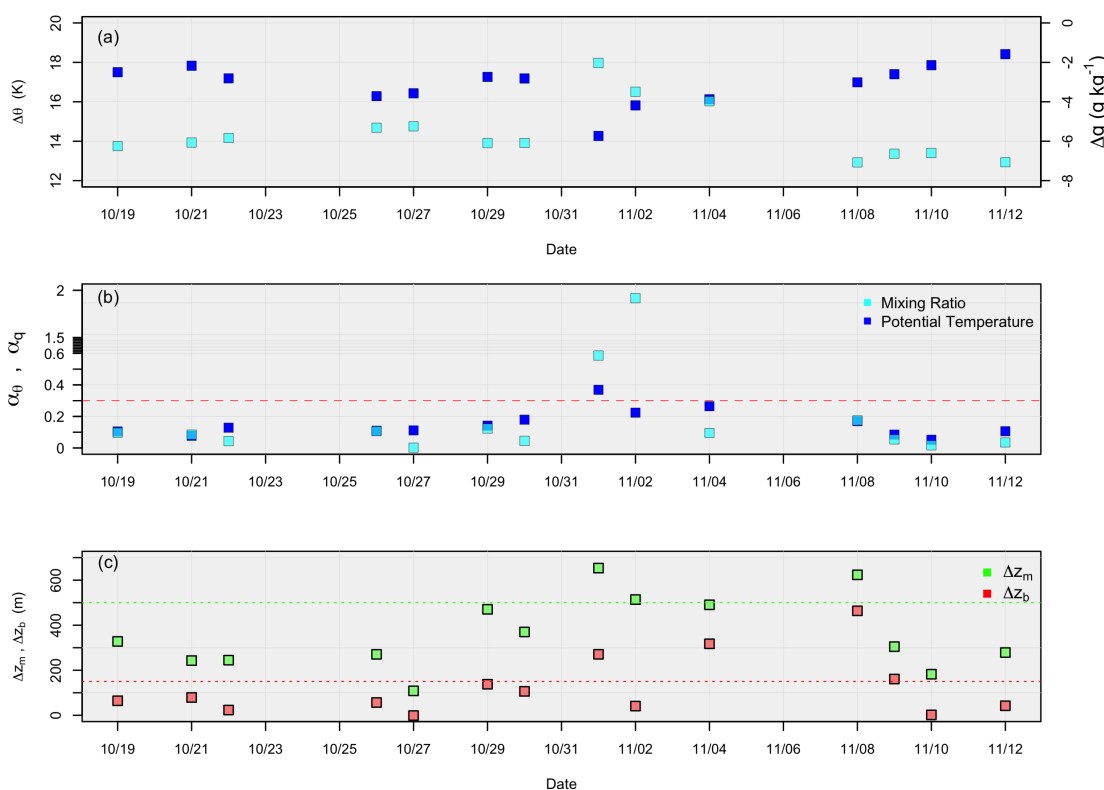

**Figure 7.** (a) $\theta$ (left y-axis, dark blue) and $q$ (right y-axis, light blue) differences across the inversion, for all 14 flights; (b) The decoupling parameters for mixing ratio (light blue) and potential temperature (dark blue), where the red dashed line represents the 0.30 value; (c) Mixed layer cloud thickness (green) and the difference between cloud base and the LCL (red), where the red dashed line represents the 500 value and the green dashed line represents the 150 value.

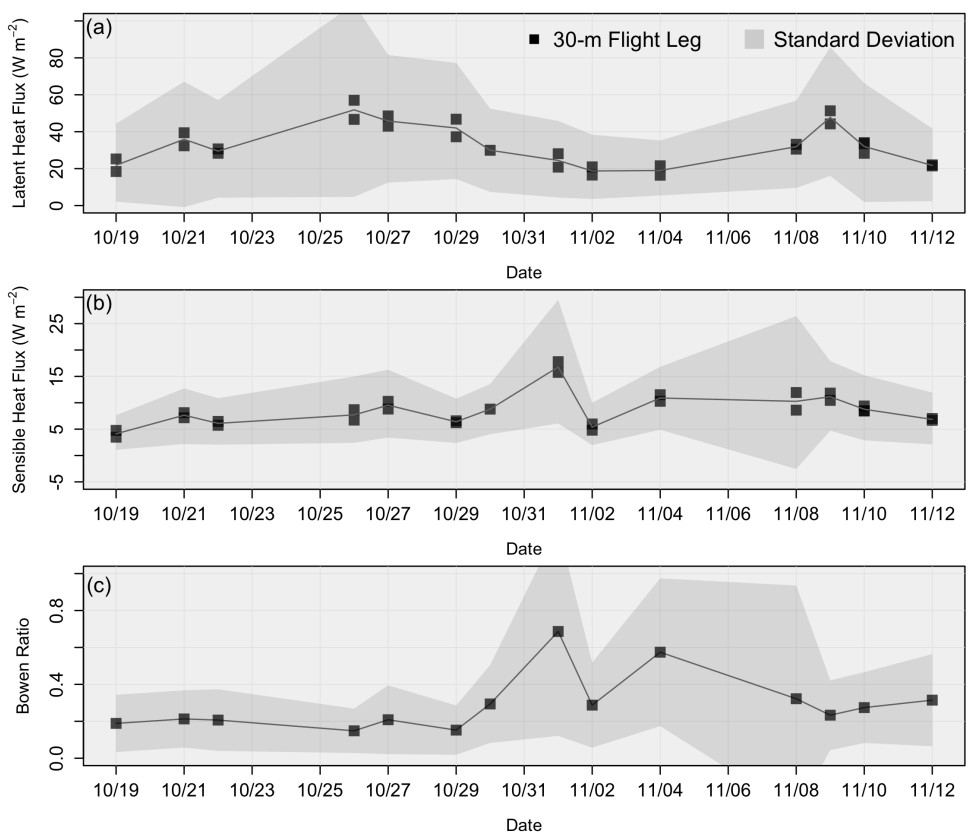

**Figure 8.** Values of (a) surface LHF ($\mathrm{Wm^{-2}}$); (b) surface SHF ($\mathrm{Wm^{-2}}$); and (c) the Bowen ratio, for each flight day. Note that each black square is a mean of a 30-m horizontal flight leg, while the gray envelope represents the standard deviation.

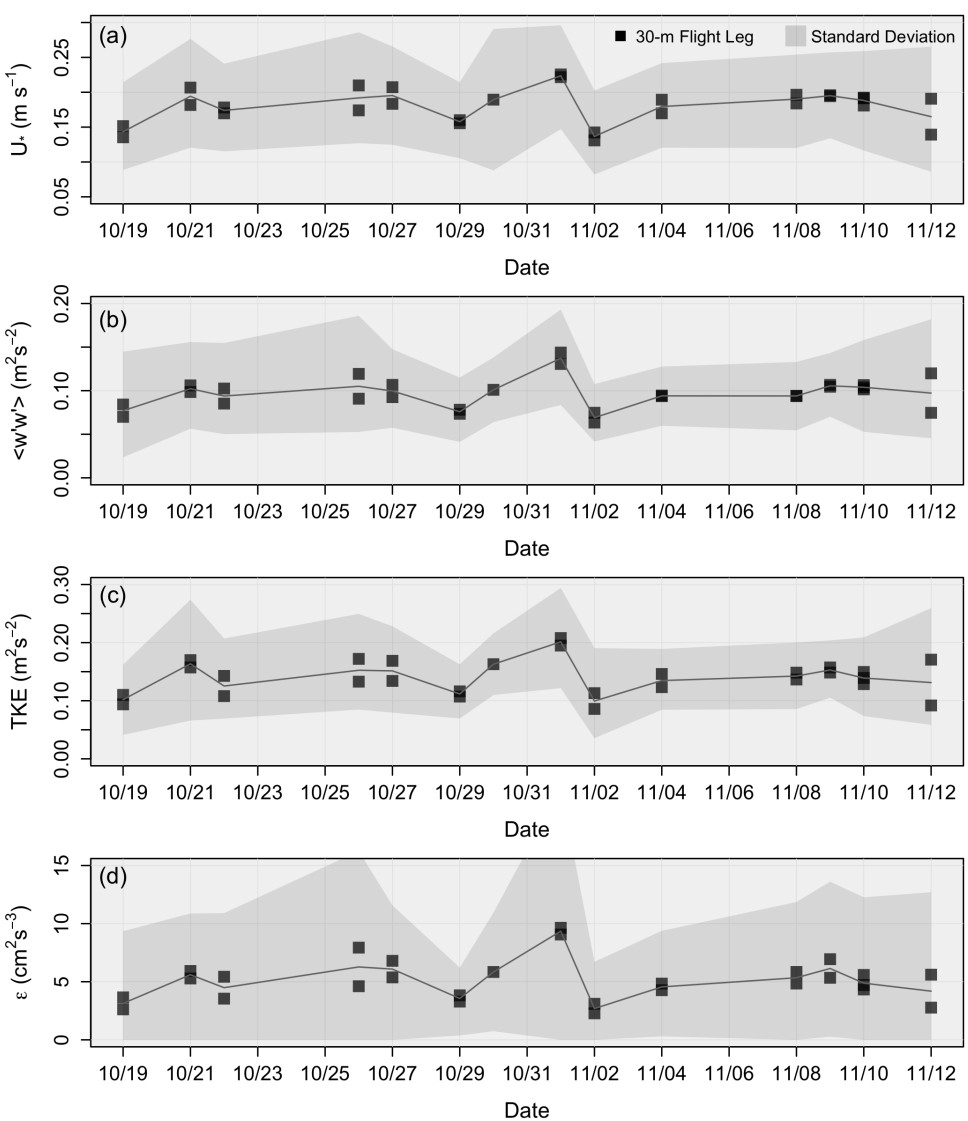

**Figure 9.** As in Figure 8, except for (a) Friction velocity ($m^2s^{-2}$); (b) Vertical velocity variance ($m^2s^{-2}$); (c) TKE ($m^2s^{-2}$); (d) $\epsilon$ ($cm^2s^{-3}$).

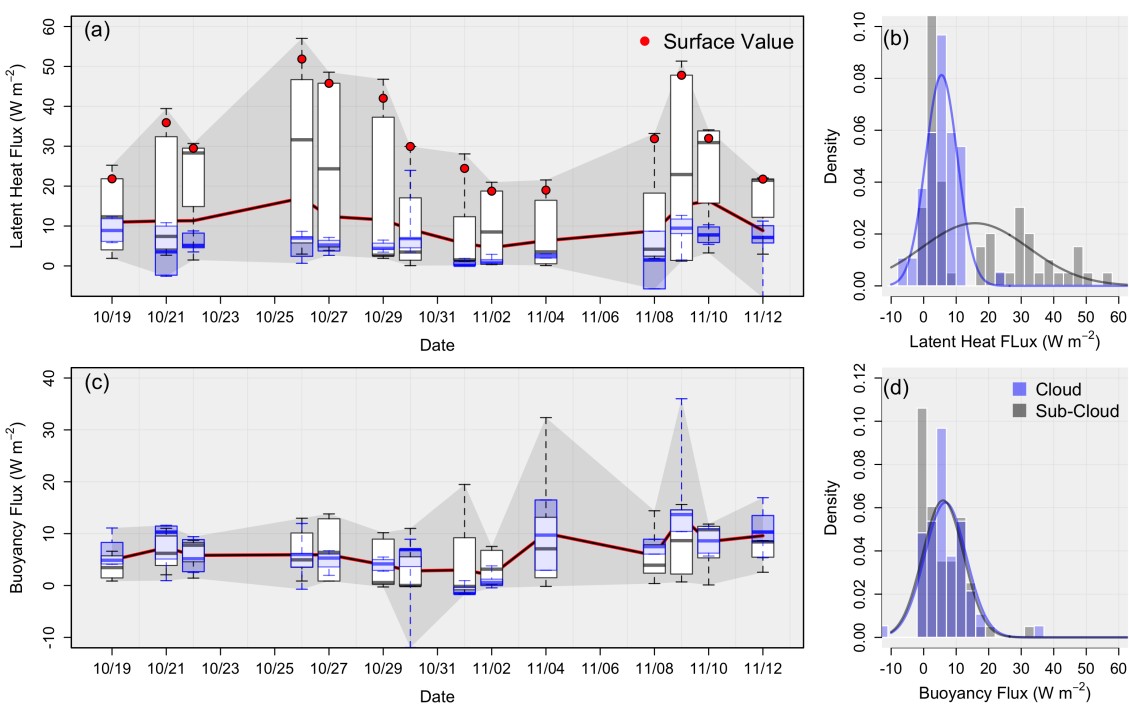

**Figure 10.** Boxplots of in-cloud (blue) and sub-cloud (white) data using mean values of horizontal flight legs for (a) the LHF (Wm$^{-2}$) and (c) the Buoyancy flux (Wm$^{-2}$). Note that the gray envelope represents the range of the data, while the red line represents each flight mean values. Panels (b) and (d) shows the distributions of the data populations (with normal distributions overlaid for reference) for the LHF and buoyancy flux, respectively.

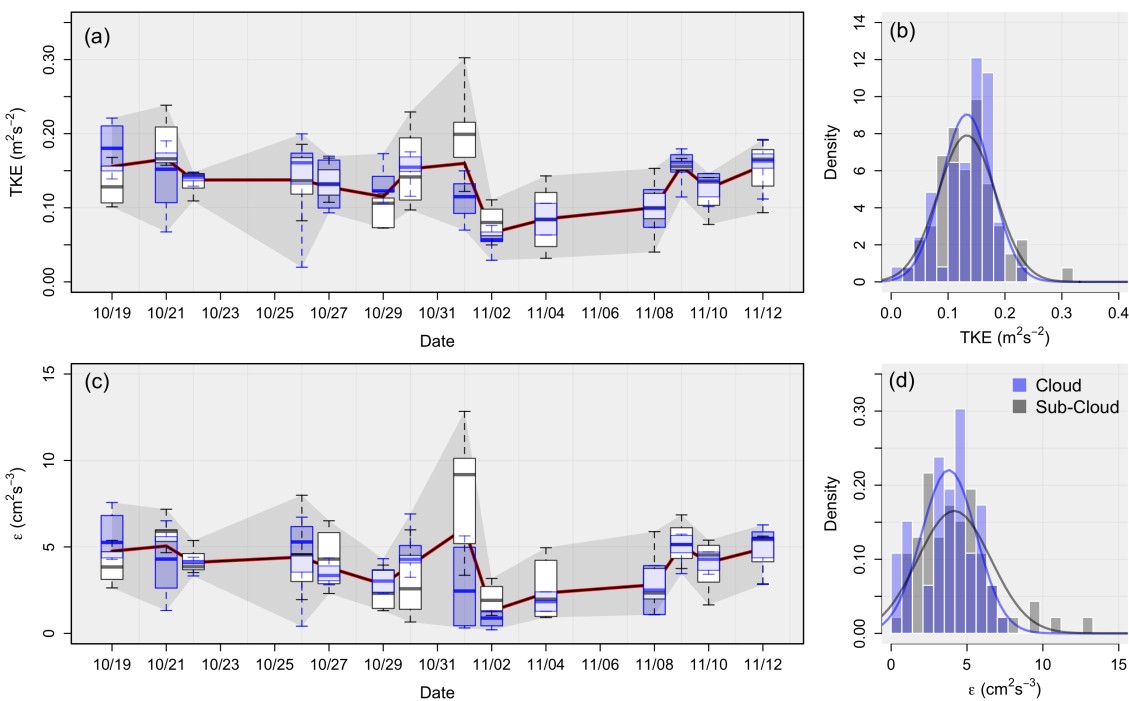

**Figure 11.** As in Figure 10, except for TKE (m$^2$s$^{-2}$) in (a) and (b) and $\epsilon$ (cm$^{-2}$s$^{-3}$) in (c) and (d)

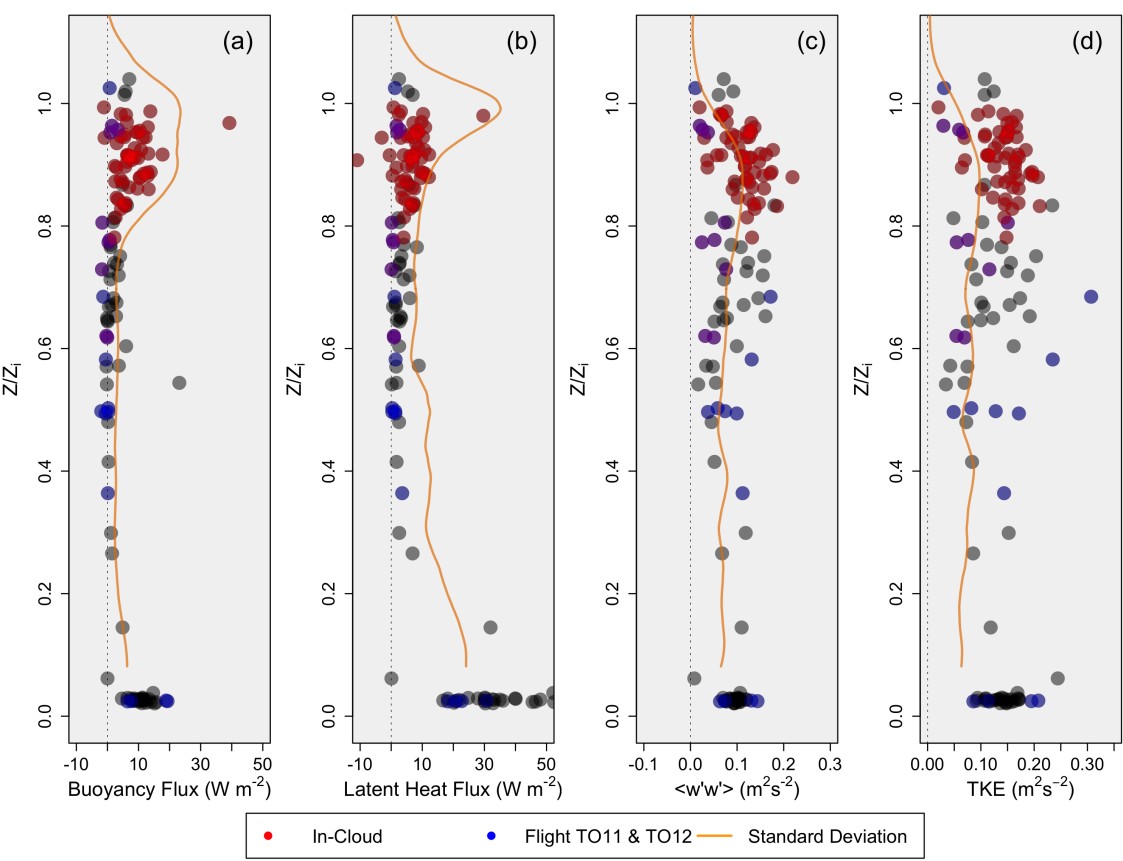

**Figure 12.** Mean values of horizontal flight legs plotted as a function of normalized boundary layer height. In-cloud data is red while data collected during Nov. $1^{st}$ and $2^{nd}$ is blue. The standard deviation is represented in orange from vertical sounding data. (a) Buoyancy flux $(Wm^{-2})$; (b) LHF $(Wm^{-2})$; (c) Vertical velocity variance $(m^2s^{-2})$; (d) TKE $(m^2s^{-2})$.

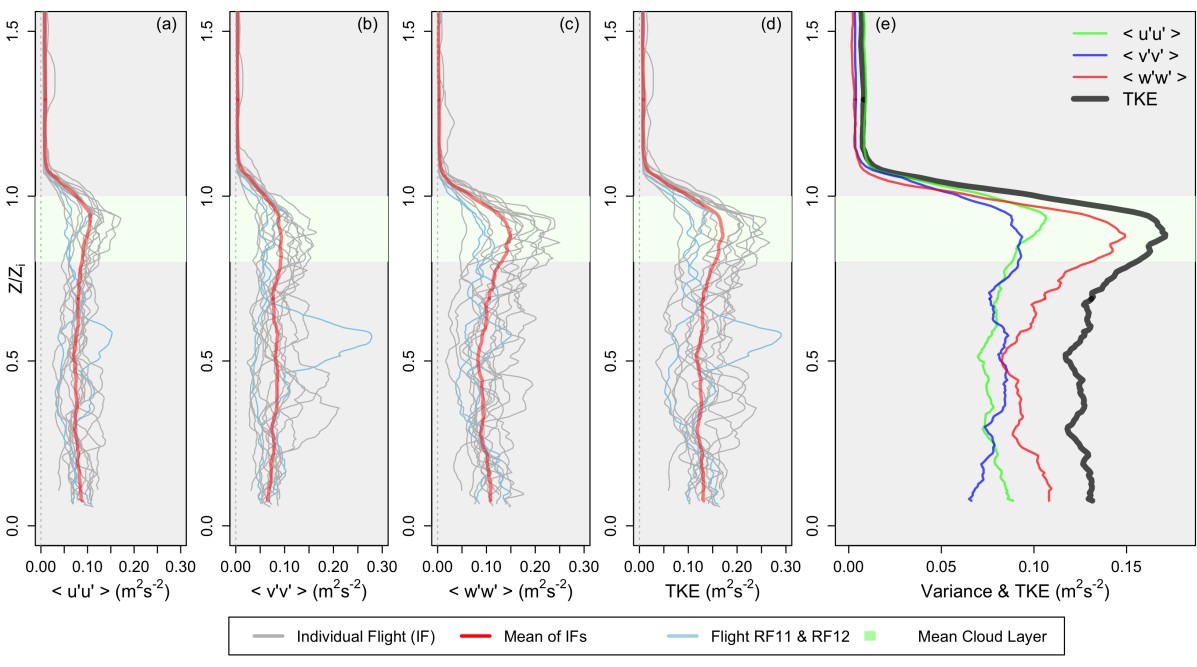

**Figure 13.** Vertical profiles (from data collected during flight profiles) of (a) u-variance ($m^2s^{-2}$); (b) v-variance ($m^2s^{-2}$); (c) w-variance ($m^2s^{-2}$); (d) TKE ($m^2s^{-2}$). Individual flights are displayed in gray, the mean value is displayed in red, with RF11 and RF12 shown in blue. Panel (e) shows the mean values from each of panels (a) through (d)

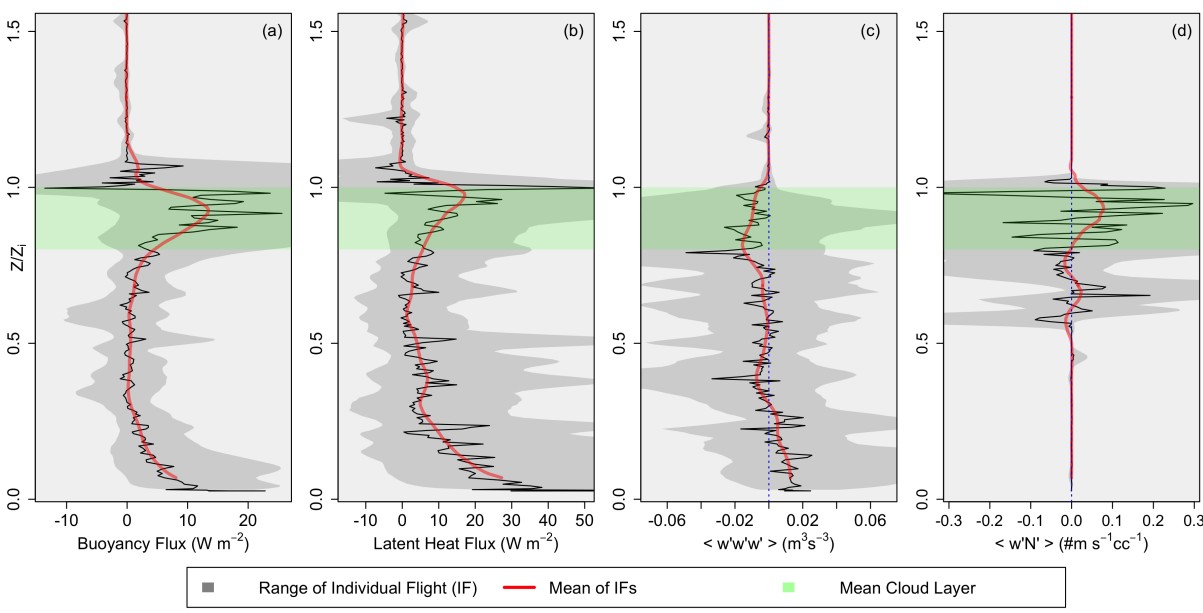

**Figure 14.** Vertical profiles (from data collected during flight profiles) of (a) Buoyancy flux (Wm$^{-2}$); (b) LHF (Wm$^{-2}$); (c) Vertical velocity skewness (m$^3$s$^{-3}$); (d) cloud droplet flux (ms$^{-1}$cc$^{-1}$). Note that the red line is the smoothed average of the raw data (black), while the gray envelope represents the range of values encountered.

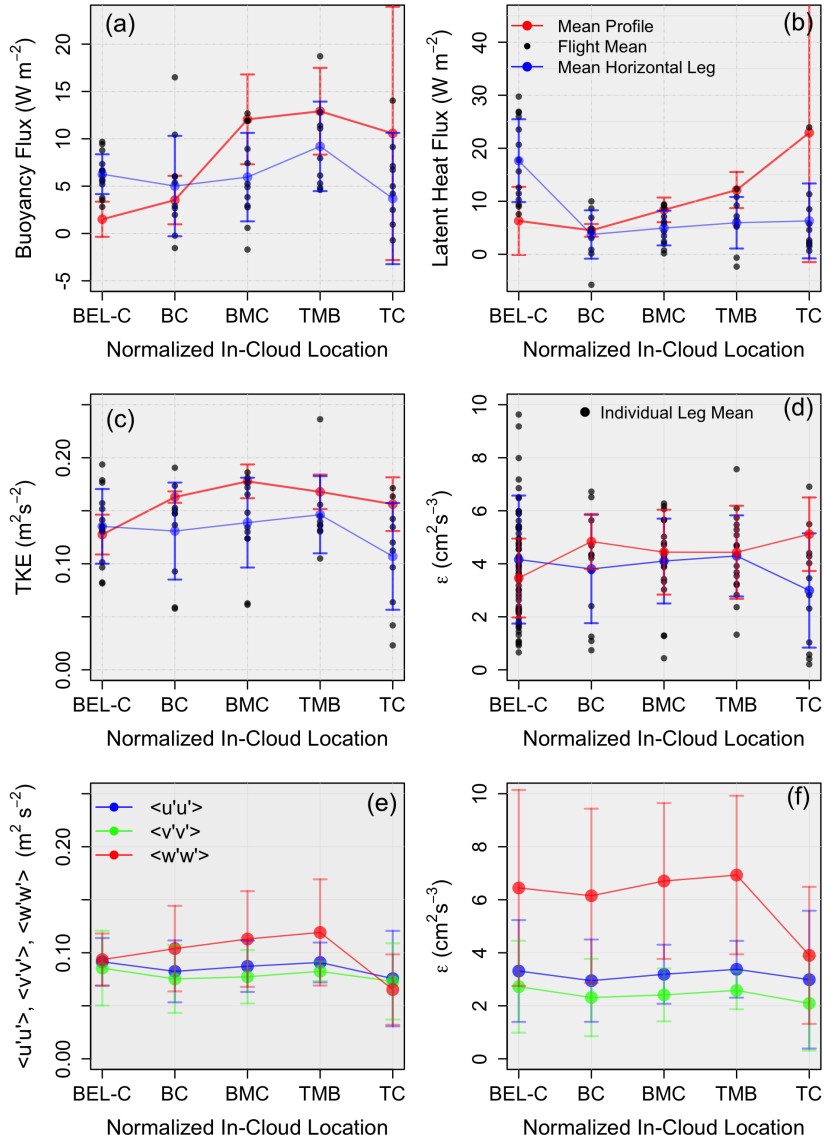

**Figure 15.** (a) Buoyancy flux (Wm$^{-2}$); (b) LHF (Wm$^{-2}$); (c) TKE (m$^2$s$^{-2}$); (d) $\epsilon$ (cm$^2$s$^{-3}$). Data is divided into layers, including below-cloud (BEL-C), bottom of cloud (BC), bottom-middle of cloud (BMC), top-middle cloud (TMC), and top of cloud (TC). Red represents mean values for each layer using data collected during flight vertical profiles while blue represents mean values for each layer using data collected during horizontal flight legs. Black dots represent mean values for each flight using horizontal flight leg data. Note that the black dots in Panel (e) represent individual leg mean values as opposed to mean flight values. Panels (e) and (f) represent $u$ (blue), $v$ (green), and $w$ (red) components of the TKE and $\epsilon$, respectively.

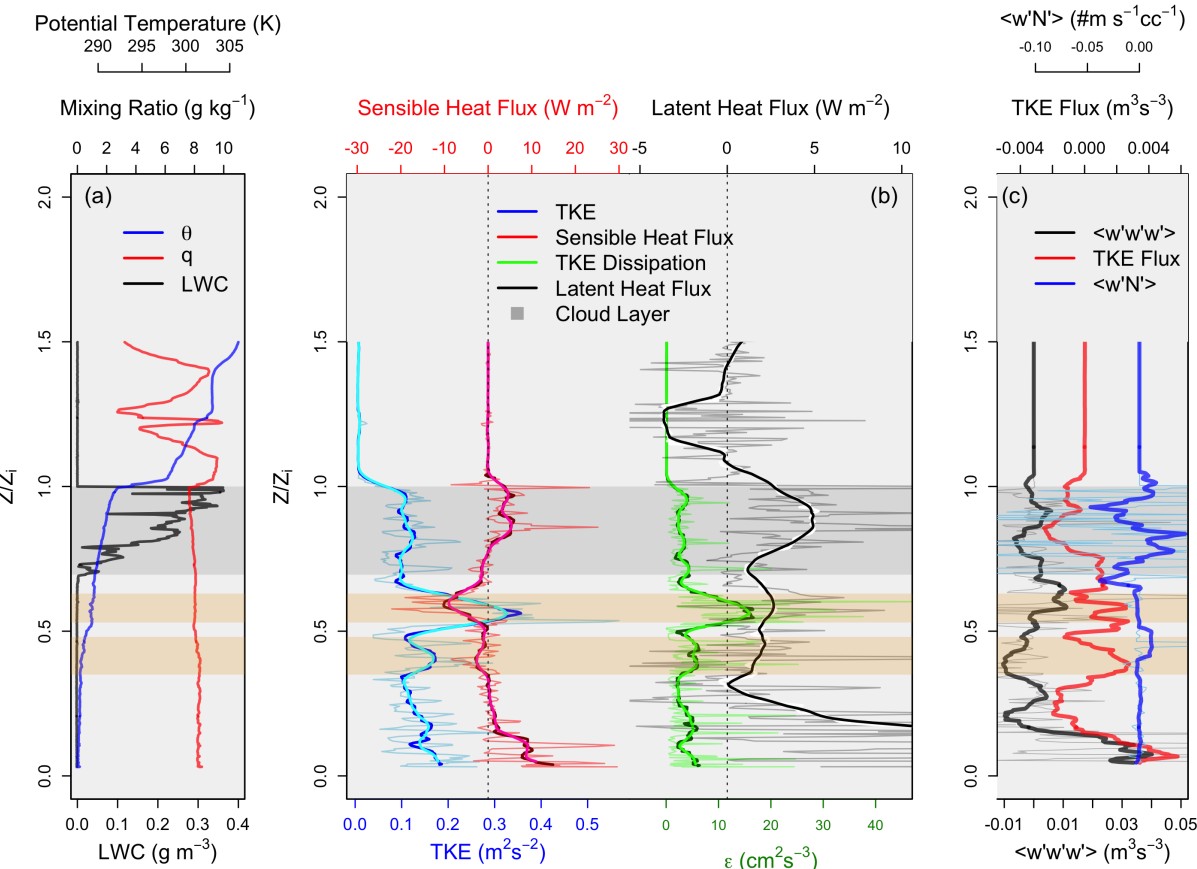

**Figure 16.** Vertical profiles as functions of normalized boundary layer height of (a) $\theta$ (K) in blue, $q$ (g kg$^{-1}$) in red, and LWC (gm$^{-3}$) in black; (b) TKE (m$^2$s$^{-2}$) in blue, SHF (Wm$^{-2}$) in red, $\epsilon$ (cm$^2$s$^{-3}$) in green, and the LHF (Wm$^{-2}$) in black. Note that the thin light colored lines represent raw values, while the dark thick lines represent smoothed averages; (c) $<w'w'w'>$ (m$^3$s$^{-3}$) in black, the TKE Flux (m$^3$s$^{-3}$) in red, and the droplet number concentration flux (m s$^{-1}$cc$^{-1}$) in blue. Note that the gray envelope represents the cloud layer, and the orange envelopes represent areas of interest (decoupling locations).

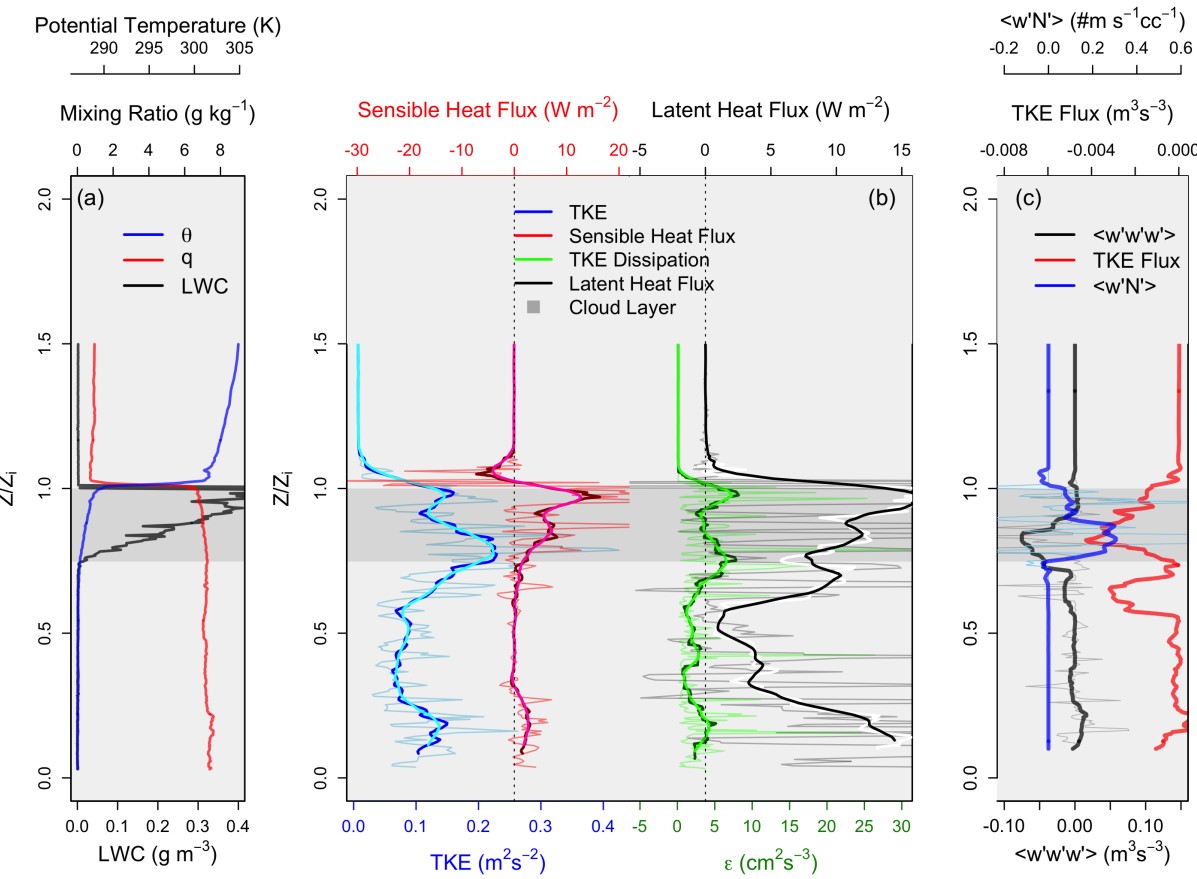

**Figure 17.** As in Figure 16, except for the well-mixed boundary layer case of RF03 (Oct $19^{th}$)

**Table 1.** Column (1): Research Flight (RF) identification; (2) The corresponding date; (3) Flight start and end times at Point Alpha. Note that local time: UTC – 4; (4) Boundary Layer conditions for each flight.

| Flight | Date | Time (UTC) | BL Conditions |
| --- | --- | --- | --- |
| RF 3 | Oct 19, 2008 | 12:05 - 14:40 | Well Mixed |
| RF 4 | Oct 21, 2008 | 12:10 - 14:50 | Well Mixed |
| RF 5 | Oct 22, 2008 | 12:00 - 14:40 | Well Mixed |
| RF 7 | Oct 26, 2008 | 12:00 - 15:00 | Well Mixed |
| RF 8 | Oct 27, 2008 | 15:55 - 19:00 | Well Mixed |
| RF 9 | Oct 29, 2008 | 11:50 - 15:00 | Well Mixed |
| RF 10 | Oct 30, 2008 | 11:50 - 15:00 | Well Mixed |
| RF 11 | Nov 01, 2008 | 12:05 - 15:05 | Wind Shear / Moisture Above |
| RF 12 | Nov 02, 2008 | 11:55 - 15:00 | Moisture Above |
| RF 13 | Nov 04, 2008 | 11:50 - 14:40 | Wind Shear |
| RF 15 | Nov 08, 2008 | 11:50 - 15:00 | Decoupled |
| RF 16 | Nov 09, 2008 | 11:50 - 15:05 | Well Mixed |
| RF 17 | Nov 10, 2008 | 14:45 - 18:00 | Well Mixed |
| RF 18 | Nov 12, 2008 | 11:50 - 15:15 | Well Mixed |

**Table 2.** Mean, standard deviation, and range of values for select variables over the 14 flights analyzed, with standard deviation values in parenthesis.

| | Mean | Range |
|---|---|---|
| $z_i$ (m) | 1173 (119) | 993 - 1450 |
| Cloud Base (m) | 936 (141) | 716 - 1291 |
| Cloud Thickness (m) | 237 (101) | 107 - 475 |
| Boundary Layer $\theta$ (K) | 289 (1.12) | 287 - 291 |
| Boundary Layer $q$ (g kg$^{-1}$) | 7.51 (0.48) | 6.83 - 8.32 |
| $\Delta\theta$ (K) | 16.8 (1.10) | 13.89 - 18.42 |
| $\Delta q$ (g kg$^{-1}$) | -5.53 (1.50) | -7.10 - 1.46 |
| Boundary Layer PCASP (cc$^{-1}$) | 410 (127) | 642 - 230 |
| CDNC (cc$^{-1}$) | 262 (110) | 80.5 - 423 |
| Drop Size ($\mu$m) | 12.33 (2.83) | 9.6 - 20.5 |
| Boundary Layer Wind Speed (m s$^{-1}$) | 4.42 (1.44) | 2.03 - 6.31 |
| Boundary Layer Wind Direction ($^\circ$) | 170 (46) | NA |
| Free Atmosphere Wind Speed (m s$^{-1}$) | 5.16 (1.89) | 2.83 - 8.34 |
| Free Atmosphere Wind Direction ($^\circ$) | 280 (115) | NA |
| $\alpha_\theta$ | 0.15 (0.08) | 0.05 - 0.37 |
| $\alpha_q$ | 0.071 (0.049) | 0.002 - 1.94 |
| $\Delta z_m$ (m) | 363 (164) | 108 - 653 |
| $\Delta z_b$ (m) | 125 (136) | 1.8 - 463 |

**Table 3.** Mean and range of values for select surface variables over the 14 flights analyzed, with standard deviation and the research flight number in parentheses for column mean and range, respectively.

| | Mean | Range |
|---|---|---|
| Latent heat flux (Wm$^{-2}$) | 32.6 (11.5) | 24.1 (RF 03) - 53.3 (RF 04) |
| Sensible heat flux (Wm$^{-2}$) | 8.6 (3.2) | 3.93 (RF 03) - 17.1 (RF 01) |
| Bowen ratio | 0.29 (0.15) | 0.15 (RF 04) - 0.68 (RF 11) |
| Friction velocity (ms$^{-1}$) | 0.17 (0.023) | 0.13 (RF 11) - 0.22 (RF12) |
| Vertical velocity variance (ms$^{-1}$) | 0.097 (0.017) | 0.073 (RF 11) - 0.114 (RF 12) |
| TKE (m$^2$s$^{-2}$) | 0.14 (0.27) | 0.051 (RF 11) - 0.20 (RF 12) |
| TKE dissipation rate (cm$^2$s$^{-3}$) | 5.14 (1.64) | 9.40 (RF 11) - 2.64 (RF 12) |

**Table 4.** Correlation coefficient values in the right-panel, with the variables in the left panel. Note that $\leftrightarrow$ divides the variables being compared. GPH is geopotential height (i.e., a proxy for pressure). $N_a$ and $N_D$ represent the aerosol number concentration and the cloud droplet number concentration, respectively.

| | Correlation |
|---|:---:|
| $z_i \leftrightarrow$ GPH | -0.37 |
| $\omega \leftrightarrow$ GPH | -0.89 |
| Wind speed $\leftrightarrow$ GPH | 0.14 |
| SHF $\leftrightarrow$ wind speed | 0.80 |
| LHF $\leftrightarrow$ wind speed | 0.36 |
| $z_i \leftrightarrow$ wind speed | 0.30 |
| $z_i \leftrightarrow$ SHF | 0.44 |
| $z_i \leftrightarrow$ LHF | 0.36 |
| LHF $\leftrightarrow$ cloud thickness | -0.50 |
| SHF $\leftrightarrow$ cloud thickness | -0.10 |
| in-cloud $\epsilon \leftrightarrow z_i$ | -0.34 |
| sub-cloud $\epsilon \leftrightarrow z_i$ | -0.13 |
| in-cloud TKE $\leftrightarrow z_i$ | -0.32 |
| sub-cloud TKE $\leftrightarrow z_i$ | -0.20 |
| $N_a \leftrightarrow$ TKE | 0.35 |
| $N_D \leftrightarrow$ TKE | 0.42 |
| drop size ($\mu$m) $\leftrightarrow$ TKE | -0.32 |
| $N_a \leftrightarrow \epsilon$ | 0.37 |
| $N_D \leftrightarrow \epsilon$ | 0.37 |
| drop size ($\mu$m) $\leftrightarrow \epsilon$ | -0.32 |

**Table 5.** Mean values for each layer discussed in Figure 15, where the top rows represent data calculated using flight vertical profiles, while the bottom rows represent data calculated using horizontal flight legs. See the text for exact partitioning of the cloud layer.

| Vertical Profile Data | $<w'\theta_v'>$ (Wm$^{-2}$) | $<w'q'>$ (Wm$^{-2}$) | TKE (m$^2$s$^{-2}$) | $\epsilon$ (cm$^2$s$^{-3}$) |
|---|---|---|---|---|
| Below Cloud | 1.95 | 7.83 | 0.129 | 3.43 |
| Cloud Base | 3.75 | 4.88 | 0.165 | 4.69 |
| Bottom Middle | 12.21 | 8.18 | 0.178 | 4.37 |
| Top Middle | 13.58 | 13.77 | 0.167 | 4.73 |
| Cloud Top | 10.57 | 21.05 | 0.154 | 5.03 |
| **Horizontal Leg Data** | | | | |
| Below Cloud | 6.13 | 17.67 | 0.136 | 4.16 |
| Cloud Base | 4.55 | 4.50 | 0.131 | 3.81 |
| Bottom Middle | 6.10 | 5.03 | 0.139 | 4.10 |
| Top Middle | 8.30 | 7.56 | 0.145 | 4.30 |
| Cloud Top | 4.10 | 6.02 | 0.108 | 3.01 |