# Peer review of "Turbulent and Boundary Layer Characteristics during VOCALS-REx"

_Atmospheric Chemistry and Physics, 2020_

## Referee Comment (RC1) · Anonymous Referee #1 · 29 May 2020

Review of Dodson and Small-Griswold: Turbulent and Boundary Layer Characteristics during VOCALS-REx

This manuscript analyzes aircraft data from the VOCALS field campaign in 2008. Its focus is on meteorological influences on boundary layer properties such as TKE. Although it presents results from most flight days, more attention is placed on a period influenced by a synoptic system. The contrast between this period and other days is useful.

The value of the paper is in its broad documentation of data from the campaign. I was surprised to see that other than the Zheng et al. study that looked at the CIRPAS Twin Otter data, no other Twin Otter studies are referenced, so perhaps these data have not been looked at before. That being said, this manuscript is not near a state in which I

could recommend publication. I was extremely frustrated by its lack of clarity, rambling unfocused explanations, and grammatical and spelling errors throughout. The number of figures is also disproportionate to the information content of the manuscript.

I will provide some specific examples below. I will only review this manuscript again if a very serious effort has been made to significantly improve the presentation and to focus the message(s). I will leave out the very long list of grammatical and spelling errors. It's really not fair to expect reviewers to clean up manuscripts.

1) Abstract needs significant tightening and clarification of message. What is the role of radiative cooling? The last two lines could be written much more simply ("lower pressure allows the BL and entrainment zone thickness to increase" plus an explanation in a separate sentence of why turbulence decreases). Mentioning both in one sentence left me scratching my head.

2) Introduction: Other than the message of "we're going to look at the turbulence data from this field campaign" I didn't get a sense of a focused science question. The authors should rewrite this with the benefits of hindsight to provide that focus.

3) Section 1.1, there is a general lack of clarity and synthesis. Please be specific about mixing ratio (of water vapor!) and give it a symbol at the outset so its intuitive. Why is the Sc to Cu transition relevant here? "Buoyancy flux is the primary generator of TKE in the STBL" but what drives buoyancy flux? The reader has to read through to pick out the pieces and figure it out. The description of Bowen ratio is straightforward and the text should be streamlined. Lines 104 through the end of this section ramble. On the other hand, you might explain why larger latent heat flux causes decoupling. Please simplify where necessary and expand where necessary to focus your key questions.

4) Section 2.1: Shortwave absorption doesn't only occur at cloud top. Line 142: Don't you mean "on any given day"?

Given the focus on velocities, surely the instrumentation should be described rather

than completely deferred to Zheng et al.?

5) Section 2.2: You talk about 300-point averaging windows before discussing the sampling time. This is upside-down. Why is linear regression required to get the mean?

6) "Thetav is commonly used as a proxy for density". Please give a concise theoretical reasoning for this. You mention a structure function method but provide no explanation of what it is. Please give a brief one. Simplify line 188. What are "interactions with the plane"?

7) Section 3.1: What is omega? Surely it should be defined and given a symbol?

8) Section 3.3: although you don't have flight data on consecutive days, you do have reanalysis that I expect would be helpful to address boundary layer height changes. (ECMWF?) Top of pg 10: why does enhanced moisture above the BL translate to higher aerosol? Here and elsewhere you would help the reader a great deal by using symbols like z' for normalized altitude, theta, q, etc – i.e., symbols that are in common use. You mention a secondary cloud layer (line 329). Is this a layer of penetrating cumulus? Or something else?

9) Section 4: Line 332, don't you mean horizontal layers? Line 347: This doesn't make sense. An increase in the Bowen ratio means an increase in SHF or decrease in LHF. Line 378: How can Fig. 11 display the same information as Fig. 10? Perhaps you mean it has the same format. There are similar instances. The use of geopotential height is distracting, and for no good reason. You could make your points much more clearly by talking about pressure. I had to read the text starting from Eq. (9) through to near the end of the section a half dozen times and I still don't know what you are trying to say. Correlations are mentioned and causation is implied. And when it is not, one is left wondering why there is a correlation, and what confounders might be driving the correlation. The summary section might have helped, but it is poorly written, sometimes repetitive, and circular. Why is geopotential height correlated with sensible heat flux. It may be simple, but at least provide a physical explanation. Stating "agreement with

Palm (1996)" doesn't help. The last 3 lines of this section do make sense, and the 'could be' might not be necessary.

10) Section 4.2: Line 441, the variance peak at z'=0.99 might simply be because of the strong q gradient. Lines 480-418, you make it sound like the updrafts and downdrafts are meeting in the middle, but they must be spatially displaced.

11) Section 4.3: This entire section should be tightened. I get contradictory messages on the role of precipitation. It can both stabilize the BL (cooling near surface) or desta-bilize (cooling higher up). I don't have a clear picture of the precipitation/evaporative cooling profile. Line 558, why bring in the skewness with a single sentence? How does it tie into the text above. What do you mean by "the boundary layer has been turned over"? Please be more precise.

12) Conclusions If pressure increased after the passage of the front, why did the BL height increase? The bullet points are helpful. The paper would benefit greatly if the Conclusions contained more synthesis like this – particularly if focused science ques-tions/hypotheses were addressed. I sincerely hope the authors will focus the revised manuscript around science questions. Lines 610-611: This isn't an interesting result. It's an artifact of the sampling. I don't know why it is in the Conclusions. The last lines are so far from the theme of this paper that I wonder why the authors mention these topics.
* * *

---

## Referee Comment (RC2) · Anonymous Referee #2 · 5 Jun 2020

Review of the article titled "Turbulent and Boundary Layer Characteristics during VOCALS-Rex" by Dodson and Griswold for publication in the Atmos. Chem. Phys.

The authors have used data collected by the Twin Otter aircraft during the VOCALS field campaign to characterize the boundary layer turbulence and microphysical properties. The article describes the synoptic, cloud, turbulence and boundary layer properties at the point-alpha, and then goes on to do correlation statistics between them. The article is overall well-written, and easy to read. However, the analysis falls short in many aspects. The principle issue stems from a lot of research already been done on the collected data. Please find below some suggestions for improving the manuscript. I recommend this article for publication after major revisions. The amount and nature of comments are rather severe, sorry about that, but they are meant for improving the

paper.

Major Comments: The referencing in this article looks little outdated. Most of the references in the introduction section are from the 80s, 90s, 00s, and the latest paper is Wood 2012. It will be good if the authors can do a thorough literature review and only refer papers from the last 5-10 years. I completely agree with the authors that the old papers are still valuable and are relevant. However, some of the conclusions/speculations reached by the authors have already been made by the subsequent article. It will be good if the authors can improve the referencing. There have been plethora of stratocumulus-turbulence interaction studies done in the past 5-10 years, using the cloud radars and large domain LES models. Line 94-99 document the turbulence structure of stratocumulus topped boundary layers, and it seems that the authors are not aware of recent findings.

In a similar vein, it is unclear to me why the authors have not considered other papers from the VOCALS campaign. Especially as they are all in the VOCALS special issue in ACP. The conclusions similar to this article have been reached by Jones et al., and Bretherton et al. papers in the special issue. It will be good if you can put your results in the context of other studies. Thanks.

Abstract line 10: the main conclusion of the article is "Findings show that the influence of a synoptic system on Nov 1st and 2nd brings in a moist layer above the boundary layer, leading to a deepening cloud layer and precipitation during passage, and a large increase in boundary layer height and cloud thinning after passage". This is contradictory to the notion that moisture above the boundary layer reduces the cloud top cooling, thereby inhibiting turbulence and thinning the clouds. Please see Eastman and Wood (2018 JAS) and other papers. Do you think that the deepening of the boundary layer might be due to decrease in subsidence or increase in the surface fluxes? In any case, correlation does not imply causation, so maybe you can rephrase this sentence. Thanks.

Section 2.2 documents the way turbulence statistics have been calculated. It will be good if you can also include some sort of error analysis in it. I suspect the differences in you see are not statistically significant. This is often the case, however you should at least document these. Your results still should be relevant. The w'N' and the skewness of vertical velocity are the prime suspects in my opinion. Please see papers by David Turner and Wulfmeyer on the calculation of higher order moments. Also, how good are the temperature and humidity measurements within the cloud layer. The sensors suffer from significant drop shattering and cooling. Can you please discuss if the measurements are sufficient for calculating buoyancy fluxes. Thanks.

You are confusing the inversion layer and the entrainment zone. These are two different things. The entrainment zone is within (plus-minus 25 m) of the cloud top, while the inversion layer can span 100s of meters at times. There is no known mechanism that can bring air from above the top of the inversion into the cloud layer. This needs to be changes throughout the document. Please see the papers by Juan-Pedro Mellado. Thanks.

One of the main conclusions is that "A maximum in TKE on Nov. 1st (both overall average and largest single value measured) is due to precipitation acting to destabilize the sub-cloud layer, while acting to stabilize the cloud layer." This contradicts your earlier statement in the introduction about evaporating drizzle stabilizing the sub-cloud layer. There have been LES modeling studies and some observational studies showing drizzle to stabilize the sub-cloud layer, directly contradicting your conclusions.

Minor Comments:

Line 236-237: This has been already stated in the introduction section, so please remove. Thanks.

Line 268: The modulus of a number does not read well. I think you mean the absolute change. Maybe you can just mention (absolute change > 0.1)? thanks

Equation 9 seems out of place. I am not sure if it conveys anything meaningful.

Figure 1: Convert Omega to Pa/day and put latitude and longitude in regular (-ve for southern hemisphere) units.

Figure 2: Panel (b) is surface air temperature?

Figure 3: please convert Omega to Pa/day. The figure also doesn't tell much, so maybe you can move it to supplemental material.

Figure 4: Instead or in addition to the wind roses, it will be good if you also show the profile of wind speed. Thanks.

---

## Author Comment (AC1) · 17 Jul 2020

COMMENT: The referencing in this article looks little outdated. Most of the references in the introduction section are from the 80s, 90s, and 00s, and the latest paper is Wood 2012. It will be good if the authors can do a thorough literature review and only refer papers from the last 5-10 years. I completely agree with the authors that the old papers are still valuable and relevant. However, some of the conclusions/speculations reached by the authors have already been made by the subsequent article. It will be good if the authors can improve the referencing. There has been a plethora of stratocumulus-turbulence interaction studies in the last 5-10 years, using the cloud radars and large domain LES models. Line 94-99 document the turbulence structure of stratocumulus topped boundary layers, and it seems that the authors are not aware of recent findings.

REPLY:

* Referencing throughout the article has been updated. Although you make it sound as if we should only refer papers from the last 5-10 years, we have kept most of the original references and added newer (post 2010) references. Previous reviewers of previous articles have been picky about referencing the original papers. However, we do understand the need for having more balance, which we think the current manuscript achieves.

* A total of 42 references have been added (we won't list them out here) throughout the article which are dated 2009 or after, providing a more balanced approach to the references.

COMMENT: In a similar vein, it is unclear to me why the authors have not considered other papers from the VOCALS campaign. Especially as they are all in the VOCALS special issue in ACP. The conclusions similar to this article have been reached by Jones et al., and Bretherton et al. papers in the special issue. It will be a good idea if you can put your results in the context of other studies. Thanks.

REPLY:

* Originally, the lack of other VOCALS papers stemmed from the fact that most of the papers which have been published used datasets other than the Twin Otter at point Alpha. Papers which do use Twin Otter data tend to focus on aerosol and cloud microphysical properties, and not turbulence.

* The Twin Otter data is not the primary focus of analysis for other papers that have been published from VOCALS-REx. For example, Jones et al. (2011) and Bretherton et al. (2010), although they analyze the boundary layer structure and decoupling, the data being used is from the NSF C130 and/or UK BVAe146. However, you are correct in saying that results here can therefore be related to those findings.

* We have added a section to discuss previous VOCALS papers on lines 56-68 of the

new manuscript. In particular, results have been related to Jones et al. (2011) and Bretherton et al. (2010), including adding new measures of boundary layer decoupling in Figure 7 that are presented in Jones et al. (2011), as to better relate findings here to their results.

* We also found several instances where we mention findings from Zheng et al. (2011), but fail to circle back around and compare our finding with theirs (A specific example of this is the statement on lines 407-408 of the new manuscript, and the follow up statement on lines 653-655).

COMMENT: Abstract line 10: the main conclusion of the article is "Findings show that the influence of a synoptic system on Nov 1st and 2nd brings in a moist layer above the boundary layer, leading to a deepening cloud layer and precipitation during passage.". This is contradictory to the notion that moisture above the boundary layer reduces the cloud top cooling, thereby inhibiting turbulence and thinning the clouds. Please see Eastman and Wood (2018 JAS) and other papers.

REPLY:

* The sentence in question here is no longer directly in the abstract. We do discuss the precipitation and synoptic system on lines 13-18 in the new manuscript, and it should be worded more properly. We have also added an extensive discussion on how the moisture above the boundary layer can affect cloud top cooling and other cloud processes. Please see lines 629-634 of the new manuscript.

COMMENT: Do you think that the deepening of the boundary layer might be due to decrease in subsidence or increase in the surface fluxes? In any case, correlation does not imply causation, so maybe you can rephrase this sentence. Thanks.

REPLY:

* The deepening of the boundary layer....do you mean in regards to after the synoptic system passage? In the sentence in question, we state that the cloud layer deepens

(becomes more thick). As for the boundary layer height, it remains relatively unchanged between Nov 1st and 2nd (decreases roughly 50-m), but the cloud thickness becomes 100-m thicker. This is due to reduced cloud top cooling limiting the deepening of the boundary layer, while entrainment that is occurring will result in a lower LCL due to the higher moisture content. Please see lines 657-660 of the new manuscript.

* If you were referring to the deepening boundary layer after synoptic system passage, we discuss that on lines 699-704 of the new manuscript.

* It should also be noted, in particular when we are discussing the correlation coefficients, that we do our best to word the phrases properly as to not imply causation. For example, on lines 7-12 in the new manuscript, we state that "As the latent heat flux (LHF) and sensible heat flux (SHF) increases, $z_i$ increases, along with the cloud thickness decreasing with increasing LHF." This makes more sense than stating "as $z_i$ increases, the LHF and SHF increases." We know that stronger surface fluxes will increase $z_i$, but the correlation coefficients only tell us that they are correlated, not which causes the other. Everything should be phrased properly throughout.

COMMENT: Section 2.2 documents the way turbulence statistics have been calculated. It will be good if you can also include some sort of error analysis in it. I suspect the differences in you see are not statistically significant. This is often the case, however you should at least document these. Your results still should be relevant. The w'N' and the skewness of vertical velocity are the prime suspects in my opinion. Please see papers by David Turner and Wulfmeyer on the calculation of higher order moments.

REPLY:

* We have added several paragraphs at the end of Section 2.2 (See lines 227-265 in the new manuscript) addressing these concerns. You are correct that they should be documented. Figures 16 and 17 have also had the raw calculations added to the profiles of w'N' and w'w'w', which clearly shows that the mean values that were being displayed are NOT statistically significant (as you assumed).

COMMENT: Also, how good are the temperature and humidity measurements within the cloud layer. The sensors suffer from significant drop shattering and cooling. Can you please discuss if the measurements are sufficient for calculating buoyancy fluxes. Thanks.

REPLY:

* Please see lines 162-167 in the new manuscript, which addresses the concerns laid out above. Although we have a limited capacity to the detail and length of explanation which can be given within the manuscript, we think the information added should address the concerns. Also, if you are curious, you can see the links provided for more information on the total set up of the Twin Otter, which has taken great care to make the most accurate measurements possible.

https://archive.eol.ucar.edu/projects/post/meetings/200902/documents/khelif_POST_SLC_F

https://www.researchgate.net/figure/UCI-Turbulence-instrumentation-on-the-CIRPAS-Twin-Otter-in-POST-and-VOCALS-REx-field_fig1_228968823

COMMENT: You are confusing the inversion layer and the entrainment zone. These are two different things. The entrainment zone is within (plus-minus 25 m) of the cloud top, while the inversion layer can span 100s of meters at a times. There is no known mechanism that can bring air from above the top of the inversion into the cloud layer. This needs to be changed throughout the document. Please see papers by Juan-Pedro Mellado. Thanks.

REPLY:

* You are correct. We (multiple times) exchanged the terms inversion layer and entrainment zone. This has been corrected throughout the manuscript and a more accurate explanation has been added. In particular, see the discussion added in the introduction on lines 89-96:

"The boundary layer top is characterized by several strong gradients, including the

cloud boundary (gradient in LWC), the entrainment zone (gradient in vorticity, where the entrainment zone separates regions of weak and strong mixing between laminar flow above and turbulent flow below), and the capping inversion (gradient in potential temperature). The cloud boundary typically lies in the entrainment zone (Albrecht et al. 1985, Malinowski et al. 2013), which in turn lies in the capping inversion, although these layers do not necessarily coincide (Mellado, 2017). Turbulent analysis of these layers in Jen La Plant et al. (2016) found that turbulence (both TKE and TKE dissipation) decreases moving from cloud top into the free atmosphere above, where mixing of the laminar and turbulent flows occurs within the entrainment layer."

* All subsequent discussions of entrainment have been modified within the manuscript. Although there are multiple examples of this throughout the manuscript, please see lines 485-486 within the new manuscript for a specific example. All original explanations which made it sound like air was being entrained from above the inversion layer has been corrected.

COMMENT: One of the main conclusions is that "A maximum in TKE on Nov. 1st (both overall average and largest single value measured) is due to precipitation acting to destabilize the sub-cloud layer, while acting to stabilize the cloud layer.". This contradicts your earlier statement in the introduction about evaporating drizzle stabilizing the sub-cloud layer. There have been LES modeling studies and some observational studies showing drizzle to stabilize the sub-cloud layer, directly contradicting your conclusions.

REPLY:

* An updated discussion relating to precipitation and its effects on boundary layer stability has been provided. Originally the explanation of precipitation within the boundary layer and how it may change the turbulent profiles was lacking. In particular, lines 635-644 in the new manuscript provide an updated discussion on how precipitation can influence the boundary layer. Feingold et al. 1996 is the original (as far as we know)

study to demonstrate how evaporation from precipitation acts to change boundary layer turbulence. If evaporation is occurring in select regions away from the surface (say just below cloud base), the sub-cloud layer will become unstable (i.e., light precipitation is occurring). If evaporation is occurring throughout the vertical sub-cloud layer, and in particular near the surface (i.e., heavy precipitation is occurring), the sub-cloud layer will become stable. The most recent paper that we could find to report findings of this nature is Ghate and Cadeddu (2019), who found that for a similar amount of radiative cooling at the cloud top, the average vertical velocity variance in the sub-cloud layer was about 16% lower during strongly precipitating hours than during weakly precipitating hours.

* The earlier statement in the introduction was referring to the explanation provided in Zheng et al. (2011). It is stated that "Zheng et al. (2011) suggest drizzle processes act to stabilize the boundary layer, leading to decoupling on Nov. 1st." I have circled back around to this statement on lines 653-655 of the new manuscript, stating that Zheng et al. is correct in stating that drizzle acts to decouple the boundary layer, but wrong in suggesting that it acts to stabilize the boundary layer as well.

COMMENT: Line 236-237: This has been already stated in the introduction section, so please remove. Thanks.

REPLY:

* We have removed the statement in question from the new manuscript.

COMMENT: Line 268: The modulus of a number does not read well. I think you mean the absolute change. Maybe you can just mention (absolute change > 0.1)? Thanks.

REPLY:

* You are correct in that we mean the absolute change. We have taken your advice and made the necessary corrections. Please see lines 341-343 in the new manuscript.

COMMENT: Equation 9 seems out of place. I am not sure if it conveys anything meaningful.

REPLY:

*This equation has been removed, although what the equation conveys has been kept. Please see lines 491-495 in the new manuscript. We were just trying to relate that the boundary layer height changes based on entrainment and large scale subsidence. This can easily be described, as opposed to showing the equation however.

COMMENT: Figure 1: Covert Omega to Pa/day and put latitude and longitude in regular (-ve for southern hemisphere) units.

REPLY:

* The units have been converted to hPa/day, which is much more relatable than the original Pa/second. We are also unsure what you mean by –ve for the latitude units. However, we have changed the latitude and longitude labeling to match what has been published in previous VOCALS-REx publications. (see Zhang et al. 2011, Toniazzo et al. 2011, Rahn and Garreaud 2010). If you would prefer a different unit or way of labeling, we would be more than happy to change it.

COMMENT: Figure 2: Panel (b) is surface air temperature?

REPLY:

* Yes, it is surface air temperature. This has been added to both the figure description and figure label. See Figure 3 in the new manuscript.

COMMENT: Figure 3: Please convert Omega to Pa/day. The figure also doesn't tell much, so maybe you can move it to supplemental material.

REPLY:

* This figure has been removed, especially since we already have a large number of figures presented.

COMMENT: Figure 4: Instead or in addition to the wind roses, it will be good if you also show the profile of wind speed. Thanks.

REPLY:

* We have kept to wind roses, but have also added a vertical profile for wind speed. Please see Figure 4 Panel (e) in the new manuscript.

---

## Author Comment (AC2) · 17 Jul 2020

COMMENT: I was surprised to see that other than the Zheng et al. study that looked at the CIRPAS Twin Otter data, no other Twin Otter studies are referenced, so perhaps these data have not been looked at before. I was extremely frustrated by its lack of clarity, rambling unfocused explanations, and grammatical and spelling errors throughout. The number of figures is also disproportionate to the information content of the manuscript.

REPLY:

* We hope that the modifications to the manuscript have addressed your concerns. In particular, we have made most of the recommended changes in regards to your

specific comments below, and have hopefully clarified some of the explanations given throughout the manuscript.

* The number of figures remains at 17 (one removed, one added through the revision process). Although yes, this is disproportionate, we also find it reasonable. The nature of this paper remains in characterizing boundary layer turbulence. Characterizing, or to describe something based on its main qualities, involves going over all those qualities in a broad sense. If the paper was focused on a single (or multiple) scientific questions to be answered, the paper would no longer be a characterization. Given the title of the paper, we assume that the reader will be aware of this fact, and understand what it is they are about to read.

* The Twin Otter data is not the primary focus of analysis for other papers that have been published from VOCALS-REx. For example, Jones et al. (2011) and Bretherton et al. (2010), although they analyze the boundary layer structure and decoupling, the data being used is from the NSF C130 and UK BVAe146. Results here can therefore be related to those findings. A discussion has been added to the introduction (see lines 56-68 of the new manuscript) relating previous VOCALS publications to the datasets used.

COMMENT: 1) Abstract needs significant tightening and clarification of message. What is the role of radiative cooling? The last two lines could be written much more simply ("lower pressure allows the BL and entrainment zone thickness to increase" plus an explanation in a separate sentence of why turbulence decreases). Mentioning both in one sentence left me scratching my head.

REPLY:

* Hopefully you will find the structure more appropriate. We start the abstract by stating what it is we are going to do, the data we are using, where it was collected, and then provide various results which were found. It should be more focused.

* The role of radiative cooling seems to be out of place in the abstract, but is given more detail (in particular with regards to how radiative cooling is effected by the enhanced moist layer present on Nov. 1st and 2nd on lines 629-630 and/or 157-159 of the new manuscript.

* The last sentence from the old manuscript has been removed, but a similar statement has been made on lines 7-9 of the new manuscript. Pressure has been used in replacement of geopotential height, and the phrase has been divided into multiple sentences. Note we have replaced geopotential height with pressure when applicable throughout the manuscript.

COMMENT: 2) Introduction: Other than the message of "we're going to look at the turbulence data from this field campaign" I didn't get a sense of a focused science question. The authors should rewrite this with the benefits of hindsight to provide that focus.

REPLY:

* We would like to point out, that at no point in the original manuscript did we state "we're going to look at the turbulence data from this field campaign", we do outline our goals in two parts however (please see lines 51-54 in the new manuscript). While yes, these goals are broad, the point of the paper is to look at the in situ data collected to provide a general characterization of the turbulence within the boundary layer. This is a very reasonable objective, as no other paper from VOCALS has analyzed the turbulent data from the Twin-Otter aircraft to this extent. Given that the purpose of the Twin-Otter was to measure turbulent and microphysical properties, a paper which characterizes said turbulence seems more than reasonable (we also find it odd that one has not been published up to this point). We outline this in lines 65-68 of the new manuscript.

* The Twin Otter data is not the primary focus of analysis for other papers that have been published from VOCALS-REx. For example, Jones et al. (2011) and Bretherton et al. (2010), although they analyze the boundary layer structure and decoupling, the

data being used is from the NSF C130 and UK BVAe146. Results here can therefore be related to those findings.

COMMENT: 3) Section 1.1, there is a general lack of clarity and synthesis. Please be specific about mixing ratio (of water vapor!) and give it a symbol at the outset so its intuitive. Why is the Sc to Cu transition relevant here? "Buoyancy flux is the primary generator of TKE in the STBL" but what drives buoyancy flux? The reader has to read through to pick out the pieces and figure it out. The description of the Bowen ratio is straightforward and the text should be streamlined. Lines 104 through the end of this section ramble. On the other hand, you might explain why a larger latent het flux causes decoupling. Please simplify where necessary and expand where necessary to focus your key questions.

REPLY:

* The mixing ratio has been given a symbol in Section 2.1, where after that point it becomes much more common throughout the text. See line 161-162 of the new manuscript. The latent heat flux and sensible heat flux have also been simplified by using LHF and SHF, respectively.

* The original sentence on the Sc to Cu transition has been removed. Although not relevant, we were simply just pointing out that the process of entrainment and boundary layer deepening plays a key role in the Sc to Cu transition.

* Lines 100-101 in the new manuscript provides a very brief statement on what drives the buoyancy flux. Given that buoyancy arises from differences in density, which depends on temperature and moisture content, having to explain what the buoyancy flux depends on seems rudimentary. The statement in the original manuscript (and in the new manuscript) of "According to Shaw (2003), one of the main sources of TKE in clouds is evaporative cooling and condensational heating, implying the buoyancy flux is the primary generator of TKE in the STBL" on lines 107-110 of the new manuscript and lines 85-88 of the original manuscript also provides the fact that the buoyancy flux

is dependent on evaporative cooling and condensational heating.

\* The explanation of the Bowen ratio has been simplified. Please see lines 118-119 of the new manuscript.

\* The original statements from lines 104 through the end of the section have mostly been removed, even though they describe why the latent heat flux caused decoupling (perhaps you misread this portion). The discussion has been presented in a different format to hopefully make it more straightforward. Please see lines 120-123 of the new manuscript.

COMMENT: 4) Section 2.1: Shortwave absorption doesn't only occur at cloud top. Line 142: Don't you mean "on any given day"?

REPLY:

\* You are correct, solar absorption does not occur only at cloud top (although this is where it is primarily confined). We have clarified this statement in lines 157-160 of the new manuscript. \* "on any given day" and "from day to day" mean the same thing. However, we have reworded this to "on any given day" on line 173 of the new manuscript.

COMMENT: Given the focus on velocities, surely the instrumentation should be described rather than completely differed to Zhang et. al.?

REPLY:

\* A more detailed explanation of the instrumentation, in particular that of the velocity measurements and moisture measurements, has been provided on lines 162-167 of the new manuscript.

COMMENT: 5) Section 2.2: You talk about 300-point averaging windows before discussing the sampling time. This is upside-down. Why is linear regression required to get the mean?

REPLY:

* The sampling time is initially discussed in the previous section (2.1) on line 134 of the original manuscript. However, we have corrected this and mentioned that the data used is 40-Hz before discussing the averaging technique on line 184 in the new manuscript.

* Linear regression is not required to get the mean, but it is one of multiple options. Other options include applying just a single mean value, or applying low pass filtering. Please see the figure 1 for examples of the three methods, all which are based on 320-pt. averaging.

COMMENT: 6) "Thetav is commonly used as a proxy for density". Please give a concise theoretical reasoning for this. You mention structure function method but provide no explanation of what it is. Please give a brief one. Simplify line 188. What are "interactions with the plane"?

REPLY:

* Virtual potential temperature is given by: theta_v = theta(1+0.61q-ql), where theta is the actual potential temperature and q is the mixing ratio of water vapor, and ql is the mixing ratio of liquid water in the air. Because water vapor is less dense than dry air, humid air has a warmer theta_v than dry air, while liquid water drops, if falling at terminal velocity, make the air heavier and therefore is associated with colder theta_v . Therefore, theta_v can be used for buoyancy. To be honest, this seems rudimentary to be included in a journal, but has been added nonetheless. Please see lines 200-203 in the new manuscript.

* We are quite confused on "you mention structure function method but provide no explanation of what it is". Please see Equations (5) and (6), which provide the structure function in mathematical form followed by an explanation. Please see lines 113-114 in the new manuscript. We have added a very brief explanation of what a structure function does (i.e., it is just a statistic to analyze common variation in a time series).

* Line 188 in the original manuscript has been simplified. See lines 221-223 in the new manuscript.

* Interactions with the plane (at higher frequencies) include aircraft vibrations, etc. Please see Figure 1 Panel (a) in the new manuscript, where the 0.3-5 Hz frequency range is shown as a light gray envelope. 0.3-5Hz covers the inertial subrange of the data, with a spike in energy located at roughly 10Hz, again, attributed to interactions with plane vibrations, etc. This spike in energy is also observed in Jen-La Plante et. al. (2016), where their explanation is "interactions with the plane".

COMMENT: 7) Section 3.1: What is omega? Surely it should be defined and given a symbol?

REPLY:

* Omega is the vertical velocity in pressure coordinates (so positive omega is negative vertical velocity), having units of pressure per time. Since much of operational meteorology uses pressure surfaces, omega is a more common quantity to see, especially when quantifying larger temporal scale vertical motions. We have added a brief description of this on lines 275-276 of the new manuscript.

COMMENT: 8) Section 3.3: Although you don't have flight data on consecutive days, you do have reanalysis that I expect would be helpful to address boundary layer height changes. (ECMWF?)

REPLY:

* This is a great point, and an excellent addition to the manuscript. Figure 5 in the new manuscript has added ECMWF-BLH data that was derived from the extrapolation of relative humidity (RH) data, where the BLH was determined in the vertical layer that had the largest gradient of RH. There is (relatively) good agreement between the in-situ and ECMWF data, and the ECMWF provides a look at what the BLH did during days where flights did not occur.

COMMENT: Top of page 10: why does enhanced moisture above the BL translate to higher aerosol? Here and elsewhere you would help the reader a great deal by using symbols like z' for normalized altitude, theta, q, etc. – i.e., symbols that are in common use.

REPLY:

* Increased moisture can lead to aerosol swelling for aerosols that are hygroscopic. This means that aerosols that are smaller than the size range being measured by the PCASP (range 0.1 – 2 um) under dry conditions, may increase in size enough under more moist conditions to be measured. This has been discussed on lines 363-365 of the new manuscript. Also, symbols have been added for variables such as zi (inversion height or BLH), q (mixing ratio) and (potential temperature) throughout the new manuscript.

COMMENT: You mention a secondary cloud layer (line 329). Is this a layer of penetrating cumulus? Or something else?

REPLY:

* It is believed to be a layer of cumulus, but not penetrating the Sc deck. The profile of LWC in Figure 2 can give you a better idea of the structure of the profile, where the main Sc deck is in red, and the secondary cloud layer (cumulus layer) is in blue. See lines 412-414 in the new manuscript.

COMMENT: 9) Section 4: Line 332, don't you mean horizontal layers?

REPLY:

* We did mean vertical layers, in reference to analyzing the boundary layer through distinct vertical bins or layers (i.e., between z/zi =0 to 0.25, or the bottom $\frac{1}{4}$ of the boundary layer, etc.). However, we think the paragraph reads better by just removing the sentence in question.

COMMENT: Line 347: This doesn't make sense. An increase in the Bowen ratio means an increase in SHF or decrease in LHF.

REPLY:

* This was a typo, and has been changed to sensible heat flux in the new manuscript (Figure 8 did display the correct information, the text just mixed up the sensible heat flux and latent heat flux). Please see line 438 in the new manuscript.

COMMENT: Line 378: How can Fig. 11 display the same information as Fig 10? Perhaps you mean it has the same format. There are similar instances.

REPLY:

* Yes, we mean that it has the same format as that of Fig. 11. This has been corrected throughout the manuscript. Please see line 468 in the new manuscript for an example.

COMMENT: The use of geopotential height is distracting, and for no good reason. You could make your points much more clearly by talking about pressure. I had to read the text starting from Eq. (9) through to near the end of the section a half dozen times and I still don't know what you are trying to say. Correlations are mentioned and causation is implied. And when it is not, one is left wondering why there is a correlation, and what confounders might be driving the correlation. The summary section might have helped, but it is poorly written, and sometimes repetitive, and circular. Why is geopotential height correlated with sensible heat flux? It may be simple, but at least provide a physical explanation. Stating "agreement with Palm (1996)" doesn't help. The last 3 lines of this section do make sense, and the 'could be' might not be necessary.

REPLY:

* Where applicable, geopotential height has been replaced with pressure. Please see lines 490-495 in the new manuscript for an example (although there are multiple instances where GPH has been replaced with pressure throughout the new manuscript).

* We believe we tried to convey to much information within this section (in regards to discussing the correlations), and as a result it seemed confusing and congested, with no clear start and finish (i.e., circular and repetitive, as you stated). This entire section has been re-written and simplified. Correlations which are not directly discussed in the text have been removed from Table 4. Each paragraph is arranged to discuss a correlation, to go along with a physical explanation for why said correlation exists.

* For example, the second to last paragraph in this section (lines 508-511 in the new manuscript), discuss the correlation between turbulence and pressure and boundary layer height. Correlation values are given, and a physical explanation for this correlation is provided. Again, the last paragraph in this section (lines 512-516 in the new manuscript), discuss the correlation between Na, Nd, and drop size with turbulence. Correlation values are given, and a physical explanation for this correlation is provided. The summary section has been removed (we feel it is no longer needed with how this section has been re-written), and the last three lines of the section from the original manuscript have been moved to lines 509-510 of the new manuscript, and the 'could be' has been removed.

COMMENT: 10) Section 4.2: Line 441, the variance peak at z'=0.99 might simply be because of the strong q gradient. Lines 480-418, you make it sound like the updrafts and downdrafts are meeting in the middle, but they must be spatially displaced.

REPLY:

* You are absolutely correct. The variance due to the strong q gradient is common in most boundary layer vertical profiles of q, and should have been mentioned. This has been added at two points within the new manuscript. Please see line 526 and line 575 in the new manuscript.

* The updrafts and downdrafts are not meeting in the middle. Here, we are simply implying that the peak in w'theta_v' at an in-cloud normalized height of 0.59 (near cloud middle) is due to the w' and theta_v' both being large and positive (i.e., warm

moist updrafts, a positive flux)) and due to the w' and theta_v' both being large and negative (i.e., cool dry downdrafts, still a positive flux). This height value of 0.59 just so happens to be where this is a maximum. The flux is still positive throughout the cloud layer however, meaning that warm moist updrafts and/or cool dry downdrafts are present throughout the cloud layer depth, they are just enhanced (or at a maximum) just above cloud middle. If the flux was negative, warm moist downdrafts and/or cool dry updrafts would be dominant.

COMMENT: 11) Section 4.3: This entire section should be tightened. I get contradictory messages on the role of precipitation. It can both stabilize the BL (cooling near the surface) or destabilize (cooling higher up). I don't have a clear picture of the precipitation/evaporative cooling profile. Line 558, why bring in the skewness with a single sentence? How does it tie into the text above? What do you mean by "the boundary layer has been turned over"? please be more precise.

REPLY:

* An updated discussion relating to precipitation and its effects on boundary layer stability has been provided. Originally the explanation of precipitation within the boundary layer and how it may change the turbulent profiles was lacking. In particular, lines 635-644 in the new manuscript provide an updated discussion on how precipitation can influence the boundary layer. Feingold et al. 1996 is the original (as far as we know) study to demonstrate how evaporation from precipitation acts to change boundary layer turbulence. If evaporation is occurring in select regions away from the surface (say just below cloud base), the sub-cloud layer will become unstable (i.e., light precipitation is occurring). If evaporation is occurring throughout the vertical sub-cloud layer, and in particular near the surface (i.e., heavy precipitation is occurring), the sub-cloud layer will become stable. The most recent paper that we could find to report findings of this nature is Ghate and Cadeddu (2019), who found that for a similar amount of radiative cooling at the cloud top, the average vertical velocity variance in the sub-cloud layer was about 16% lower during strongly precipitating hours than during weakly precipitating hours. Hopefully the updated discussion provides a clearer picture of what is occurring. Our results (based on profiles of LHF and SHF) demonstrate evaporation occurring away from cloud base, near $z/z_i = 0.40$ and $0.60$ (orange envelopes in Figure 16), leading to the increased turbulent values measured in the sub-cloud.

* We have changed the sentence mentioning the skewness to include the phrase "providing more evidence that" on line 665 of the new manuscript. This is done as to provide more meaning for the skewness in relation to providing further evidence that the boundary layer is decoupled. It is mentioned in previous sections that the skewness is negative for well-mixed boundary layers. We are simply just trying to connect back to this concept. If needed, we can remove this sentence completely and the reader can just refer to the figure.

* The sentence originally containing the phrase "the boundary layer has been turned over" has been removed from the new manuscript. However, we were simply just referring to the fact that turbulent mixing stabilizes the boundary layer (the mixing reduces the instability, or the boundary layer has been overturned meaning the warm air near the surface and cold air in evaporative regions were mixed together).

COMMENT: 12) Conclusions if pressure increased after the passage of the front, why did the BL height increase? The bullets are helpful. The paper would benefit greatly if the conclusions contained more synthesis like this – particularly if focused science questions/hypothesis were addressed. I sincerely hope the authors will focus the revised manuscript around science questions. Lines 610-611: this isn't an interesting result. It's an artifact of the sampling. I don't know why it is in the conclusions. The last lines are so far from the theme of this paper that I wonder why the authors mention these topics.

REPLY:

* A small discussion has been added on lines 699-704 of the new manuscript to address why the BL height increased while pressure (i.e., subsidence) was increasing.

\* We have discussed our reasoning in regards to focusing on specific questions/hypothesis in our previous reply towards the beginning of this document. This paper sets out to characterize boundary layer turbulence (there are many papers that simply characterize results from a field campaign, without focusing on specific questions or hypothesis). In regards to the VOCALS dataset, in particular that collected at point Alpha in the twin otter (with an objective to measure turbulence among other things), very little work has been published in regards to the turbulent structure. Most published work revolves around other aircraft and ship based measurements that were made at other sampling regions during VOCALS. For example, Jones et al. (2011) and Bretherton et al. (2010), both which focus on boundary layer structure, decoupling, and precipitation, use data collected from the NSF C-130. Having this characterization of the turbulence in our paper goes a long way toward not only relating to results found in other regions during the campaign, but leads to a better understanding of turbulence on a day to day basis, and what variables can influence it.

\* Although not an interesting result, we do believe it is worth mentioning that how one sets out to measure turbulence will ultimately influence the results. Figure 15 is devoted to looking at this through differences when using vertical profiles or horizontal flight legs. Although this is an artifact of the sampling, and the reader can infer that you get vastly different results based on the measurement and averaging methods used, it is still central to the results that are presented and worth mentioning. \* The last three lines pertaining to future work have been removed.

[Figure]

**Fig. 1.**

[Figure]

**Profile 11**

Altitude

LWC

Lower Cloud Layer
Upper Cloud Layer

**Fig. 2.**

[Figure]

---

## Author Comment (AC3) · 17 Jul 2020

Reply to Comments:

Note that original reviewer comments are in bold below, while the author reply is represented by regular type font with bullet points. The tracked changes manuscript is attached after the reply to comments, with removed text in red and added text in blue.

REPLY TO REVIEWER #1

**I was surprised to see that other than the Zheng et al. study that looked at the CIRPAS Twin Otter data, no other Twin Otter studies are referenced, so perhaps these data have not been looked at before. I was extremely frustrated by its lack of clarity, rambling unfocused explanations, and grammatical and spelling errors throughout. The number of figures is also disproportionate to the information content of the manuscript.**

- We hope that the modifications to the manuscript have addressed your concerns. In particular, we have made most of the recommended changes in regards to your specific comments below, and have hopefully clarified some of the explanations given throughout the manuscript.
- The number of figures remains at 17 (one removed, one added through the revision process). Although yes, this is disproportionate, we also find it reasonable. The nature of this paper remains in characterizing boundary layer turbulence. Characterizing, or to describe something based on its main qualities, involves going over all those qualities in a broad sense. If the paper was focused on a single (or multiple) scientific questions to be answered, the paper would no longer be a characterization. Given the title of the paper, we assume that the reader will be aware of this fact, and understand what it is they are about to read.
- The Twin Otter data is not the primary focus of analysis for other papers that have been published from VOCALS-REx. For example, Jones et al. (2011) and Bretherton et al. (2010), although they analyze the boundary layer structure and decoupling, the data being used is from the NSF C130 and UK BVAe146. Results here can therefore be related to those findings. A discussion has been added to the introduction (see lines 56-68 of the new manuscript) relating previous VOCALS publications to the datasets used.

**1) Abstract needs significant tightening and clarification of message. What is the role of radiative cooling? The last two lines could be written much more simply ("lower pressure allows the BL and entrainment zone thickness to increase" plus an explanation in a separate sentence of why turbulence decreases). Mentioning both in one sentence left me scratching my head.**

- Hopefully you will find the structure more appropriate. We start the abstract by stating what it is we are going to do, the data we are using, where it was collected, and then provide various results which were found. It should be more focused.
- The role of radiative cooling seems to be out of place in the abstract, but is given more detail (in particular with regards to how radiative cooling is effected by the enhanced

moist layer present on Nov. 1st and 2nd on lines 629-630 and/or 157-159 of the new manuscript.

- The last sentence from the old manuscript has been removed, but a similar statement has been made on lines 7-9 of the new manuscript. Pressure has been used in replacement of geopotential height, and the phrase has been divided into multiple sentences. Note we have replaced geopotential height with pressure when applicable throughout the manuscript.

**2) Introduction: Other than the message of "we're going to look at the turbulence data from this field campaign" I didn't get a sense of a focused science question. The authors should rewrite this with the benefits of hindsight to provide that focus.**

- We would like to point out, that at no point in the original manuscript did we state "we're going to look at the turbulence data from this field campaign", we do outline our goals in two parts however (please see lines 51-54 in the new manuscript). While yes, these goals are broad, the point of the paper is to look at the in situ data collected to provide a general characterization of the turbulence within the boundary layer. This is a very reasonable objective, as no other paper from VOCALS has analyzed the turbulent data from the Twin-Otter aircraft to this extent. Given that the purpose of the Twin-Otter was to measure turbulent and microphysical properties, a paper which characterizes said turbulence seems more than reasonable (we also find it odd that one has not been published up to this point). We outline this in lines 65-68 of the new manuscript.
- The Twin Otter data is not the primary focus of analysis for other papers that have been published from VOCALS-REx. For example, Jones et al. (2011) and Bretherton et al. (2010), although they analyze the boundary layer structure and decoupling, the data being used is from the NSF C130 and UK BVAe146. Results here can therefore be related to those findings.

**3) Section 1.1, there is a general lack of clarity and synthesis. Please be specific about mixing ratio (of water vapor!) and give it a symbol at the outset so its intuitive. Why is the Sc to Cu transition relevant here? "Buoyancy flux is the primary generator of TKE in the STBL" but what drives buoyancy flux? The reader has to read through to pick out the pieces and figure it out. The description of the Bowen ratio is straightforward and the text should be streamlined. Lines 104 through the end of this section ramble. On the other hand, you might explain why a larger latent het flux causes decoupling. Please simplify where necessary and expand where necessary to focus your key questions.**

- The mixing ratio has been given a symbol in Section 2.1, where after that point it becomes much more common throughout the text. See line 161-162 of the new manuscript. The latent heat flux and sensible heat flux have also been simplified by using LHF and SHF, respectively.

- The original sentence on the Sc to Cu transition has been removed. Although not relevant, we were simply just pointing out that the process of entrainment and boundary layer deepening plays a key role in the Sc to Cu transition.
- Lines 100-101 in the new manuscript provides a very brief statement on what drives the buoyancy flux. Given that buoyancy arises from differences in density, which depends on temperature and moisture content, having to explain what the buoyancy flux depends on seems rudimentary. The statement in the original manuscript (and in the new manuscript) of "According to Shaw (2003), one of the main sources of TKE in clouds is evaporative cooling and condensational heating, implying the buoyancy flux is the primary generator of TKE in the STBL" on lines 107-110 of the new manuscript and lines 85-88 of the original manuscript also provides the fact that the buoyancy flux is dependent on evaporative cooling and condensational heating.
- The explanation of the Bowen ratio has been simplified. Please see lines 118-119 of the new manuscript.
- The original statements from lines 104 through the end of the section have mostly been removed, even though they describe why the latent heat flux caused decoupling (perhaps you misread this portion). The discussion has been presented in a different format to hopefully make it more straightforward. Please see lines 120-123 of the new manuscript.

**4) Section 2.1: Shortwave absorption doesn't only occur at cloud top. Line 142: Don't you mean "on any given day"?**

- You are correct, solar absorption does not occur only at cloud top (although this is where it is primarily confined). We have clarified this statement in lines 157-160 of the new manuscript.
- "on any given day" and "from day to day" mean the same thing. However, we have reworded this to "on any given day" on line 173 of the new manuscript.

**Given the focus on velocities, surely the instrumentation should be described rather than completely differed to Zhang et. al.?**

- A more detailed explanation of the instrumentation, in particular that of the velocity measurements and moisture measurements, has been provided on lines 162-167 of the new manuscript.

**5) Section 2.2: You talk about 300-point averaging windows before discussing the sampling time. This is upside-down. Why is linear regression required to get the mean?**

- The sampling time is initially discussed in the previous section (2.1) on line 134 of the original manuscript. However, we have corrected this and mentioned that the data used is 40-Hz before discussing the averaging technique on line 184 in the new manuscript.

- Linear regression is not required to get the mean, but it is one of multiple options. Other options include applying just a single mean value, or applying low pass filtering. Please see the figure below for examples of the three methods, all which are based on 320-pt. averaging.

[Figure]

**6) "Thetav is commonly used as a proxy for density". Please give a concise theoretical reasoning for this. You mention structure function method but provide no explanation of what it is. Please give a brief one. Simplify line 188. What are "interactions with the plane"?**

- Virtual potential temperature is given by: $\theta_v = \theta(1 + 0.61q - q_l)$, where $\theta$ is the actual potential temperature and q is the mixng ratio of water vapor, and $q_l$ is the mixing ratio of liquid water in the air. Because water vapor is less dense than dry air, humid air has a warmer $\theta_v$ than dry air, while liquid water drops, if falling at terminal velocity, make the air heavier and therefore is associated with colder $\theta_v$. Therefore, $\theta_v$ can be used for buoyancy. To be honest, this seems rudimentary to be included in a journal, but has been added nonetheless. Please see lines 200-203 in the new manuscript.
- We are quite confused on "you mention structure function method but provide no explanation of what it is". Please see Equations (5) and (6), which provide the structure function in mathematical form followed by an explanation. Please see lines 113-114 in the new manuscript. We have added a very brief explanation of what a structure function does (i.e., it is just a statistic to analyze common variation in a time series).
- Line 188 in the original manuscript has been simplified. See lines 221-223 in the new manuscript.

- Interations with the plane (at higher frequencies) include aircraft vibrations, etc. Please see Figure 1 Panel (a) in the new manuscript, where the 0.3-5 Hz frequency range is shown as a light gray envelope. 0.3-5Hz covers the inertial subrange of the data, with a spike in energy located at roughly 10Hz, again, attributed to interactions with plane vibrations, etc. This spike in energy is also observed in Jen-La Plante et. al. (2016), where their explanation is "interactions with the plane".

**7) Section 3.1: What is omega? Surely it should be defined and given a symbol?**

- Omega is the vertical velocity in pressure coordinates (so positive omega is negative vertical velocity), having units of pressure per time. Since much of operational meteorology uses pressure surfaces, omega is a more common quantity to see, especially when quantifying larger temporal scale vertical motions. We have added a brief description of this on lines 275-276 of the new manuscript.

**8) Section 3.3: Although you don't have flight data on consecutive days, you do have reanalysis that I expect would be helpful to address boundary layer height changes. (ECMWF?)**

- This is a great point, and an excellent addition to the manuscript. Figure 5 in the new manuscript has added ECMWF-BLH data that was derived from the extrapolation of relative humidity (RH) data, where the BLH was determined in the vertical layer that had the largest gradient of RH. There is (relatively) good agreement between the in-situ and ECMWF data, and the ECMWF provides a look at what the BLH did during days where flights did not occur.

**Top of page 10: why does enhanced moisture above the BL translate to higher aerosol? Here and elsewhere you would help the reader a great deal by using symbols like z' for normalized altitude, theta, q, etc. – i.e., symbols that are in common use.**

- Increased moisture can lead to aerosol swelling for aerosols that are hygroscopic. This means that aerosols that are smaller than the size range being measured by the PCASP (range 0.1 – 2 um) under dry conditions, may increase in size enough under more moist conditions to be measured. This has been discussed on lines 363-365 of the new manuscript. Also, symbols have been added for variables such as $z_i$ (inversion height or BLH), q (mixing ratio) and $\theta$ (potential temperature) throughout the new manuscript.

**You mention a secondary cloud layer (line 329). Is this a layer of penetrating cumulus? Or something else?**

- It is believed to be a layer of cumulus, but not penetrating the Sc deck. The profile of LWC below can give you a better idea of the structure of the profile, where the main Sc

deck is in red, and the secondary cloud layer (cumulus layer) is in blue. See lines 412-414 in the new manuscript.

**Profile 11**

**9) Section 4: Line 332, don't you mean horizontal layers?**

- We did mean vertical layers, in reference to analyzing the boundary layer through distinct vertical bins or layers (i.e., between $z/z_i = 0$ to 0.25, or the bottom ¼ of the boundary layer, etc.). However, we think the paragraph reads better by just removing the sentence in question.

**Line 347: This doesn't make sense. An increase in the Bowen ratio means an increase in SHF or decrease in LHF.**

- This was a typo, and has been changed to sensible heat flux in the new manuscript (Figure 8 did display the correct information, the text just mixed up the sensible heat flux and latent heat flux). Please see line 438 in the new manuscript.

**Line 378: How can Fig. 11 display the same information as Fig 10? Perhaps you mean it has the same format. There are similar instances.**

- Yes, we mean that it has the same format as that of Fig. 11. This has been corrected throughout the manuscript. Please see line 468 in the new manuscript for an example.

**The use of geopotential height is distracting, and for no good reason. You could make your points much more clearly by talking about pressure. I had to read the text starting from Eq. (9) through to near the end of the section a half dozen times and I still don't know what you are trying to say. Correlations are mentioned and causation is implied. And when it is not, one is left wondering why there is a correlation, and what confounders might be driving the correlation. The summary section might have helped, but it is poorly written, and sometimes repetitive, and circular. Why is geopotential height correlated with sensible heat flux? It may be simple, but at least provide a physical explanation. Stating "agreement with Palm (1996)" doesn't help. The last 3 lines of this section do make sense, and the 'could be' might not be necessary.**

- Where applicable, geopotential height has been replaced with pressure. Please see lines 490-495 in the new manuscript for an example (although there are multiple instances where GPH has been replaced with pressure throughout the new manuscript).
- We believe we tried to convey to much information within this section (in regards to discussing the correlations), and as a result it seemed confusing and congested, with no clear start and finish (i.e., circular and repetitive, as you stated). This entire section has been re-written and simplified. Correlations which are not directly discussed in the text have been removed from Table 4. Each paragraph is arranged to discuss a correlation, to go along with a physical explanation for why said correlation exists.
- For example, the second to last paragraph in this section (lines 508-511 in the new manuscript), discuss the correlation between turbulence and pressure and boundary layer height. Correlation values are given, and a physical explanation for this correlation is provided. Again, the last paragraph in this section (lines 512-516 in the new manuscript), discuss the correlation between $N_a$ $N_D$ and drop size with turbulence. Correlation values are given, and a physical explanation for this correlation is provided.
- The summary section has been removed (we feel it is no longer needed with how this section has been re-written), and the last three lines of the section from the original manuscript have been moved to lines 509-510 of the new manuscript, and the 'could be' has been removed.

**10) Section 4.2: Line 441, the variance peak at z'=0.99 might simply be because of the strong q gradient. Lines 480-418, you make it sound like the updrafts and downdrafts are meeting in the middle, but they must be spatially displaced.**

- You are absolutely correct. The variance due to the strong q gradient is common in most boundary layer vertical profiles of q, and should have been mentioned. This has been added at two points within the new manuscript. Please see line 526 and line 575 in the new manuscript.
- The updrafts and downdrafts are not meeting in the middle. Here, we are simply implying that the peak in w'$\theta_v$' at an in-cloud normalized height of 0.59 (near cloud middle) is due to the w' and $\theta_v$' both being large and positive (i.e., warm moist updrafts, a positive flux)) and due to the w' and $\theta_v$' both being large and negative (i.e., cool dry downdrafts, still a positive flux). This height value of 0.59 just so happens to be where this is a maximum. The flux is still positive throughout the cloud layer however, meaning that warm moist updrafts and/or cool dry downdrafts are present throughout the cloud layer depth, they are just enhanced (or at a maximum) just above cloud middle.  If the flux was negative, warm moist downdrafts and/or cool dry updrafts would be dominant.

**11) Section 4.3: This entire section should be tightened. I get contradictory messages on the role of precipitation. It can both stabilize the BL (cooling near the surface) or destabilize (cooling higher up). I don't have a clear picture of the precipitation/evaporative cooling profile. Line 558, why bring in the skewness with a single sentence? How does it tie into the text above? What do you mean by "the boundary layer has been turned over"? please be more precise.**

- An updated discussion relating to precipitation and its effects on boundary layer stability has been provided. Originally the explanation of precipitation within the boundary layer and how it may change the turbulent profiles was lacking. In particular, lines 635-644 in the new manuscript provide an updated discussion on how precipitation can influence the boundary layer. Feingold et al. 1996 is the original (as far as we know) study to demonstrate how evaporation from precipitation acts to change boundary layer turbulence. If evaporation is occurring in select regions away from the surface (say just below cloud base), the sub-cloud layer will become unstable (i.e., light precipitation is occurring). If evaporation is occurring throughout the vertical sub-cloud layer, and in particular near the surface (i.e., heavy precipitation is occurring), the sub-cloud layer will become stable. The most recent paper that we could find to report findings of this nature is Ghate and Cadeddu (2019), who found that for a similar amount of radiative cooling at the cloud top, the average vertical velocity variance in the sub-cloud layer was about 16% lower during strongly precipitating hours than during weakly precipitating hours. Hopefully the updated discussion provides a clearer picture of what is occurring. Our results (based on profiles of LHF and SHF) demonstrate evaporation occurring away from cloud base, near $z/z_i$ = 0.40 and 0.60 (orange envelopes in Figure 16), leading to the increased turbulent values measured in the sub-cloud.
- We have changed the sentence mentioning the skewness to include the phrase "providing more evidence that" on line 665 of the new manuscript. This is done as to provide more meaning for the skewness in relation to providing further evidence that the boundary layer is decoupled. It is mentioned in previous sections that the skewness is negative for well-mixed boundary layers. We are simply just trying to connect back to

this concept. If needed, we can remove this sentence completely and the reader can just refer to the figure.

- The sentence originally containing the phrase "the boundary layer has been turned over" has been removed from the new manuscript. However, we were simply just referring to the fact that turbulent mixing stabilizes the boundary layer (the mixing reduces the instability, or the boundary layer has been overturned meaning the warm air near the surface and cold air in evaporative regions were mixed together).

**12) Conclusions if pressure increased after the passage of the front, why did the BL height increase? The bullets are helpful. The paper would benefit greatly if the conclusions contained more synthesis like this – particularly if focused science questions/hypothesis were addressed. I sincerely hope the authors will focus the revised manuscript around science questions. Lines 610-611: this isn't an interesting result. It's an artifact of the sampling. I don't know why it is in the conclusions. The last lines are so far from the theme of this paper that I wonder why the authors mention these topics.**

- A small discussion has been added on lines 699-704 of the new manuscript to address why the BL height increased while pressure (i.e., subsidence) was increasing.
- We have discussed our reasoning in regards to focusing on specific questions/hypothesis in our previous reply towards the beginning of this document. This paper sets out to characterize boundary layer turbulence (there are many papers that simply characterize results from a field campaign, without focusing on specific questions or hypothesis). In regards to the VOCALS dataset, in particular that collected at point Alpha in the twin otter (with an objective to measure turbulence among other things), very little work has been published in regards to the turbulent structure. Most published work revolves around other aircraft and ship based measurements that were made at other sampling regions during VOCALS. For example, Jones et al. (2011) and Bretherton et al. (2010), both which focus on boundary layer structure, decoupling, and precipitation, use data collected from the NSF C-130. Having this characterization of the turbulence in our paper goes a long way toward not only relating to results found in other regions during the campaign, but leads to a better understanding of turbulence on a day to day basis, and what variables can influence it.
- Although not an interesting result, we do believe it is worth mentioning that how one sets out to measure turbulence will ultimately influence the results. Figure 15 is devoted to looking at this through differences when using vertical profiles or horizontal flight legs. Although this is an artifact of the sampling, and the reader can infer that you get vastly different results based on the measurement and averaging methods used, it is still central to the results that are presented and worth mentioning.
- The last three lines pertaining to future work have been removed.

REPLY TO REVIEWER #2

**The referencing in this article looks little outdated. Most of the references in the introduction section are from the 80s, 90s, and 00s, and the latest paper is Wood 2012. It will be good if the authors can do a thorough literature review and only refer papers from the last 5-10 years. I completely agree with the authors that the old papers are still valuable and relevant. However, some of the conclusions/speculations reached by the authors have already been made by the subsequent article. It will be good if the authors can improve the referencing. There has been a plethora of stratocumulus-turbulence interaction studies in the last 5-10 years, using the cloud radars and large domain LES models. Line 94-99 document the turbulence structure of stratocumulus topped boundary layers, and it seems that the authors are not aware of recent findings.**

- Referencing throughout the article has been updated. Although you make it sound as if we should only refer papers from the last 5-10 years, we have kept most of the original references and added newer (post 2010) references. Previous reviewers of previous articles have been picky about referencing the original papers. However, we do understand the need for having more balance, which we think the current manuscript achieves.
- A total of 42 references have been added (we won't list them out here) throughout the article which are dated 2009 or after, providing a more balanced approach to the references.

**In a similar vein, it is unclear to me why the authors have not considered other papers from the VOCALS campaign. Especially as they are all in the VOCALS special issue in ACP. The conclusions similar to this article have been reached by Jones et al., and Bretherton et al. papers in the special issue. It will be a good idea if you can put your results in the context of other studies. Thanks.**

- Originally, the lack of other VOCALS papers stemmed from the fact that most of the papers which have been published used datasets other than the Twin Otter at point Alpha. Papers which do use Twin Otter data tend to focus on aerosol and cloud microphysical properties, and not turbulence.
- The Twin Otter data is not the primary focus of analysis for other papers that have been published from VOCALS-REx. For example, Jones et al. (2011) and Bretherton et al. (2010), although they analyze the boundary layer structure and decoupling, the data being used is from the NSF C130 and/or UK BVAe146. However, you are correct in saying that results here can therefore be related to those findings.
- We have added a section to discuss previous VOCALS papers on lines 56-68 of the new manuscript. In particular, results have been related to Jones et al. (2011) and Bretherton et al. (2010), including adding new measures of boundary layer decoupling in Figure 7 that are presented in Jones et al. (2011), as to better relate findings here to their results.
- We also found several instances where we mention findings from Zheng et al. (2011), but fail to circle back around and compare our finding with theirs (A specific example of

this is the statement on lines 407-408 of the new manuscript, and the follow up statement on lines 653-655).

**Abstract line 10: the main conclusion of the article is "Findings show that the influence of a synoptic system on Nov 1ˢᵗ and 2ⁿᵈ brings in a moist layer above the boundary layer, leading to a deepening cloud layer and precipitation during passage.". This is contradictory to the notion that moisture above the boundary layer reduces the cloud top cooling, thereby inhibiting turbulence and thinning the clouds. Please see Eastman and Wood (2018 JAS) and other papers.**

- The sentence in question here is no longer directly in the abstract. We do discuss the precipitation and synoptic system on lines 13-18 in the new manuscript, and it should be worded more properly. We have also added an extensive discussion on how the moisture above the boundary layer can affect cloud top cooling and other cloud processes. Please see lines 629-634 of the new manuscript.

**Do you think that the deepening of the boundary layer might be due to decrease in subsidence or increase in the surface fluxes? In any case, correlation does not imply causation, so maybe you can rephrase this sentence. Thanks.**

- The deepening of the boundary layer….do you mean in regards to after the synoptic system passage? In the sentence in question, we state that the cloud layer deepens (becomes more thick). As for the boundary layer height, it remains relatively unchanged between Nov 1ˢᵗ and 2ⁿᵈ (decreases roughly 50-m), but the cloud thickness becomes 100-m thicker. This is due to reduced cloud top cooling limiting the deepening of the boundary layer, while entrainment that is occurring will result in a lower LCL due to the higher moisture content. Please see lines 657-660 of the new manuscript.
- If you were referring to the deepening boundary layer after synoptic system passage, we discuss that on lines 699-704 of the new manuscript.
- It should also be noted, in particular when we are discussing the correlation coefficients, that we do our best to word the phrases properly as to not imply causation. For example, on lines 7-12 in the new manuscript, we state that "As the latent heat flux (LHF) and sensible heat flux (SHF) increases, $z_i$ increases, along with the cloud thickness decreasing with increasing LHF."  This makes more sense than stating "as $z_i$ increases, the LHF and SHF increases." We know that stronger surface fluxes will increase $z_i$, but the correlation coefficients only tell us that they are correlated, not which causes the other. Everything should be phrased properly throughout.

**Section 2.2 documents the way turbulence statistics have been calculated. It will be good if you can also include some sort of error analysis in it. I suspect the differences in you see are not statistically significant. This is often the case, however you should at least document these. Your results still should be relevant. The w'N' and the skewness of vertical velocity are**

**the prime suspects in my opinion. Please see papers by David Turner and Wulfmeyer on the calculation of higher order moments.**

- We have added several paragraphs at the end of Section 2.2 (See lines 227-265 in the new manuscript) addressing these concerns. You are correct that they should be documented. Figures 16 and 17 have also had the raw calculations added to the profiles of w'N' and w'w'w', which clearly shows that the mean values that were being displayed are NOT statistically significant (as you assumed).

**Also, how good are the temperature and humidity measurements within the cloud layer. The sensors suffer from significant drop shattering and cooling. Can you please discuss if the measurements are sufficient for calculating buoyancy fluxes. Thanks.**

- Please see lines 162-167 in the new manuscript, which addresses the concerns laid out above. Although we have a limited capacity to the detail and length of explanation which can be given within the manuscript, we think the information added should address the concerns. Also, if you are curious, you can see the links provided for more information on the total set up of the Twin Otter, which has taken great care to make the most accurate measurements possible.

https://archive.eol.ucar.edu/projects/post/meetings/200902/documents/khelif_POST_SLC_Feb_2008_sm.pdf

https://www.researchgate.net/figure/UCI-Turbulence-instrumentation-on-the-CIRPAS-Twin-Otter-in-POST-and-VOCALS-REx-field_fig1_228968823

**You are confusing the inversion layer and the entrainment zone. These are two different things. The entrainment zone is within (plus-minus 25 m) of the cloud top, while the inversion layer can span 100s of meters at a times. There is no known mechanism that can bring air from above the top of the inversion into the cloud layer. This needs to be changed throughout the document. Please see papers by Juan-Pedro Mellado. Thanks.**

- You are correct. We (multiple times) exchanged the terms inversion layer and entrainment zone. This has been corrected throughout the manuscript and a more accurate explanation has been added. In particular, see the discussion added in the introduction on lines 89-96:

"The boundary layer top is characterized by several strong gradients, including the cloud boundary (gradient in LWC), the entrainment zone (gradient in vorticity, where the entrainment zone separates regions of weak and strong mixing between laminar flow above and turbulent flow below), and the capping inversion (gradient in potential temperature). The cloud boundary typically lies in the entrainment zone (Albrecht et al. 1985, Malinowski et al. 2013), which in turn lies in the capping inversion, although these layers do not necessarily coincide (Mellado, 2017). Turbulent analysis of these layers in Jen La Plant et al. (2016) found that turbulence (both TKE and TKE dissipation) decreases moving from cloud top into the free atmosphere above, where mixing of the laminar and turbulent flows occurs within the entrainment layer."

- All subsequent discussions of entrainment have been modified within the manuscript. Although there are multiple examples of this throughout the manuscript, please see lines 485-486 within the new manuscript for a specific example. All original explanations which made it sound like air was being entrained from above the inversion layer has been corrected.

**One of the main conclusions is that "A maximum in TKE on Nov. 1$^{st}$ (both overall average and largest single value measured) is due to precipitation acting to destabilize the sub-cloud layer, while acting to stabilize the cloud layer.". This contradicts your earlier statement in the introduction about evaporating drizzle stabilizing the sub-cloud layer. There have been LES modeling studies and some observational studies showing drizzle to stabilize the sub-cloud layer, directly contradicting your conclusions.**

- An updated discussion relating to precipitation and its effects on boundary layer stability has been provided. Originally the explanation of precipitation within the boundary layer and how it may change the turbulent profiles was lacking. In particular, lines 635-644 in the new manuscript provide an updated discussion on how precipitation can influence the boundary layer. Feingold et al. 1996 is the original (as far as we know) study to demonstrate how evaporation from precipitation acts to change boundary layer turbulence. If evaporation is occurring in select regions away from the surface (say just below cloud base), the sub-cloud layer will become unstable (i.e., light precipitation is occurring). If evaporation is occurring throughout the vertical sub-cloud layer, and in particular near the surface (i.e., heavy precipitation is occurring), the sub-cloud layer will become stable. The most recent paper that we could find to report findings of this nature is Ghate and Cadeddu (2019), who found that for a similar amount of radiative cooling at the cloud top, the average vertical velocity variance in the sub-cloud layer was about 16% lower during strongly precipitating hours than during weakly precipitating hours.
- The earlier statement in the introduction was referring to the explanation provided in Zheng et al. (2011). It is stated that "Zheng et al. (2011) suggest drizzle processes act to stabilize the boundary layer, leading to decoupling on Nov. 1$^{st}$." I have circled back around to this statement on lines 653-655 of the new manuscript, stating that Zheng et al. is correct in stating that drizzle acts to decouple the boundary layer, but wrong in suggesting that it acts to stabilize the boundary layer as well.

**Minor Comments:**

**Line 236-237: This has been already stated in the introduction section, so please remove. Thanks.**

- We have removed the statement in question from the new manuscript.

**Line 268: The modulus of a number does not read well. I think you mean the absolute change. Maybe you can just mention (absolute change > 0.1)? Thanks.**

- You are correct in that we mean the absolute change. We have taken your advice and made the necessary corrections. Please see lines 341-343 in the new manuscript.

**Equation 9 seems out of place. I am not sure if it conveys anything meaningful.**

- This equation has been removed, although what the equation conveys has been kept. Please see lines 491-495 in the new manuscript. We were just trying to relate that the boundary layer height changes based on entrainment and large scale subsidence. This can easily be described, as opposed to showing the equation however.

**Figure 1: Covert Omega to Pa/day and put latitude and longitude in regular (-ve for southern hemisphere) units.**

- The units have been converted to hPa/day, which is much more relatable than the original Pa/second. We are also unsure what you mean by –ve for the latitude units. However, we have changed the latitude and longitude labeling to match what has been published in previous VOCALS-REx publications. (see Zhang et al. 2011, Toniazzo et al. 2011, Rahn and Garreaud 2010). If you would prefer a different unit or way of labeling, we would be more than happy to change it.

**Figure 2: Panel (b) is surface air temperature?**

- Yes, it is surface air temperature. This has been added to both the figure description and figure label. See Figure 3 in the new manuscript.

**Figure 3: Please convert Omega to Pa/day. The figure also doesn't tell much, so maybe you can move it to supplemental material.**

- This figure has been removed, especially since we already have a large number of figures presented.

**Figure 4: Instead or in addition to the wind roses, it will be good if you also show the profile of wind speed. Thanks.**

- We have kept to wind roses, but have also added a vertical profile for wind speed. Please see Figure 4 Panel (e) in the new manuscript.

[revised manuscript text omitted]

---

## Referee Report (RR1)

Second Review of Dodson and Small-Griswold: Turbulent and Boundary Layer Characteristics during VOCALS-REx

Although the manuscript has been improved, my general impressions remain much the same, which can be summed up as: "A large number of pages and figures, not commensurate with the information content." I still see value in the manuscript's broad documentation of data from the campaign.

I had hoped for a clean, sharp, synthesis in the second round but unfortunately that was not the case. In an attempt to be complete, the authors lose sight of the essence of their work. Heavy **Synthesis** is required.

In addition, grammatical errors persist throughout the paper. The reader encounters sloppiness already in the abstract. (VOMOS?, LFH, "Data was…"). This is inexcusable from native English speakers. I note some grammatical errors below but it's an incomplete list.

A comment from my first review:

*4) Section 2.1: Shortwave absorption doesn't only occur at cloud top.*

• You are correct, solar absorption does not occur only at cloud top (although this is where it is primarily confined). We have clarified this statement in lines 157-160 of the new manuscript.

The revised text says "shortwave absorption, which is largest near cloud top due to the scattering of solar radiation limiting absorption lower in the cloud layer (Hignett,1991).

But Hignett 1991 states on Pg 1474: "In contrast to the longwave cooling, the heating which results from absorption of shortwave solar radiation is distributed more deeply in the cloud layer". Please do some reading on this topic and revise to better reflect the knowledge in the field.

Below I give examples of problematic text and places that need improved analysis. More than this would require me to sit down and rewrite paragraphs and I'm not going to do that.

**Abstract:**

What is "evaporation away from the surface"? Is it evaporation from the surface, or evaporation some distance from the surface? And if the latter, where? You can calculate this from the qc flux profiles.

What does "completely offset from one another" mean? Do you mean spatially offset? I waded through the paper and looked at the figures and still don't get it. It's a key result!

The inability of the LW cooling to mix through deeper boundary layers is not a new result! Begin your sentence with: "As shown previously …" and then refer to the papers later (as you do).

Lines 10/11 require work. Sentence structure, grammar.

**Introduction:**

Long and unnecessarily wordy
"subgride"?
Line 71: grammar (maintain consistent tense)
Line 76: Here and elsewhere: "height throughout the depth of the boundary layer"
Line 82: what do you mean by "Convection in the STBL is limited?" By definition, Sc are convective. They may nt penetrate very high because of the capping inversion. Please reword.

Line 91 and throughout: "dryer" is a machine that dries your clothes. "drier".
Line 91/92. These parentheses within parentheses are very distracting.
Line 124: Lewellen et al. 1996
Line 128: vertical velocity skewness is a strong function of diurnal cycle as shown in VOCALS.

**Data and Methods**

Line 153: Although probes failed on some of the days, the key cloud probes are ones that measure LWC. Surely there were other probes measuring bulk LWC (Hot wire, JW, Gerber PVM?) The aerosol data aren't even used in this paper, except as a possible indicator of reduced drizzle. And w'N' analysis is so inconclusive that I see no reason to keep it.
Line 178: "e.g., using Reynolds composition" -- an example of how a little effort can make the sentence more readable.

Line 204: "assumption that the flow is isotropic".
Why was isotropy never directly calculated (see below)?
Line 221: structure function (n=2)
Line 254: "the error bars would not plot"?? this sentence needs rewriting.
Line 263: What does "this" refer to? An equation? If so say so and refer to the Eq. number.

**Synoptic**
Long and low in information content
Line 286: remove "as was found in
Various places: "minimums" ???
Line 334: metrics instead of dictators.
Liune 335, grammar
Line 342: what does "per measurement" mean?
Line 357 grammar
Line 365: aircraft typically measure aerosols in their dry state. Please check this was not the case here.

Line 376: helped to produce
Line 386 "that"
Line 298: What is mixed layer cloud thickness in this context? Since you're comparing LCL and cloud base, I think you mean mixed layer thickness from surface to cloud base.
Line 405 grammar
Line 413: rewrite as "By these metrics 28% of…"
Line 420: "relatively consistent" in terms of what? Depth?

**Results**

Line 435: grammar
Line 436/437: (Table 3) will suffice.
Line 458 "are" not "will be".
Line 460: "Based on" not "From just analyzing.."
Line 465 grammar
Line 480: driven by
Change TO11, TO 12 to RF11, RF12 (or vice versa) for consistency
Line 491: Can be → is
Line 497: Given that $z_i$ decreases…

Table 4 gives correlations that sometimes don't fit specific cases. The later focus on Nov 1 should be brought in. How well do Nov 1 and 2 follow the broad correlations?

Line 514: "Physically this makes sense" does not make sense. There are many studies showing aerosol causing increases and decreases in LWC.
Line 516: What are 'enhanced latent heating effects'? Do you mean 'enhanced latent heating'? And if so where?

Fig. 13: Please add a line on your plots for the anisotropy ratio = $2w'w'/(u'u'+v'v')$. It goes a long way in showing the source of TKE and its relationship to top-down/bottom-up driven turbulence. Add discussion as necessary.

Line 581: "downdrafts are smallest". Do you mean smallest areal coverage?

I'm not convinced that this paper has shown anything interesting about $w'N'$. The plots are incredibly noisy and no clear results are shown. So why clutter the paper with results like this. Please go back through the paper with a similar view on other plots and remove if they don't yield a clear result. You will be doing yourselves a real service in the long run.

A moot point perhaps but on your figures for $w'N'$, cc-1 is not a standard unit.

Figure captions are meant to be cryptic. Please remove all "Note that.." from figure captions.

Figure 16 caption doesn't even say its Nov 1.

Line 619: Theta is normalized 0 to 1 (min=0; max = 1).
But since theta keeps increasing above the BL, this would typically place the unit value at the highest point of your profile. So how did you normalize theta? The plot suggests you did this correctly.

Line 625: Where is the evaporation occurring? It makes a difference. Why not calculate the divergence of the qc flux.

Pgph starting line 629: there's another factor and that is less dilution of cloud water in response to entrainment.

Line 632 grammar

Line 638 it's more about the cooling profile than the precip rate, as you discuss later. Please shorten/tighten descriptions.

Line 646: What are 'latent heating effects'? Don't you just mean 'latent heating'?

Line 652 and abstract/conclusions, "away from the surface" is highly imprecise language, and since this is a key result please make your arguments crystal clear.
The profiles show no sub-cloud drizzle that would destabilize the sub-cloud layer. Does LWC include drizzle/rain?

As mentioned above, please tie this discussion of Nov 1 back to the broader correlations and point out differences if they exist, or simply note consistency. I couldn't wade through the text one more time to figure it all out.

Line 652: "activation" not "activation through condensation". Please stick to accepted terminology.

Line 664: I don't believe you have evidence of aerosol activation in updrafts at cloud base. Your measurements are far too noisy to draw this conclusion. Updraft velocities are just not strong enough to raise supersaturation at cloud top.
Why is this not simply evaporation in downdrafts? And as I said earlier, since you can't really say much about aerosols, why dilute your message with distracting, uncertain results?

Line 655, poorly mixed BLs can be locally well coupled by local, penetrative updrafts.

Line 671: Can you explain why the consistency in wind direction across the inversion should translate to mixing across the inversion? It's possible that mixing occurs but this isn't a way to address it. The theta jump is about 4K.

Fig. 13: Why are there 2 identical blue lines for RF12 and 13 when the caption and legend lump them together?

General comment on SHF and LHF: these are generally defined at the surface so please clarify. Eqns 2, 3 give the definitions. You should define them as $\overline{w'q'}$, $\overline{w'\theta'}$!

**Conclusions**

Conclusions need to be tightened/clarified.
Line 703: entrainment acts to increase $z_i$. $Z_i$ is a number that can increase or decrease but not deepen.
Line 710: please add appropriate reference
Line 711: grammar
Line 712/713: this is a convoluted sentence. Deepening BLs entrain drier air and therefore increase surface LHF

Line 715: Again, please tell us what you mean by evaporation away from the surface.

Line 713: again explain: "completely offset one another"

Line 738: grammar

---

## Author Response (AR2)

**Reply to Second Review**

NOTE: Reviewer comments are in black below, while the reply to comments are followed in blue.

Second Review of Dodson and Small-Griswold: Turbulent and Boundary Layer Characteristics during VOCALS-REx

Although the manuscript has been improved, my general impressions remain much the same, which can be summed up as: "A large number of pages and figures, not commensurate with the information content." I still see value in the manuscript's broad documentation of data from the campaign.

I had hoped for a clean, sharp, synthesis in the second round but unfortunately that was not the case. In an attempt to be complete, the authors lose sight of the essence of their work. Heavy **Synthesis** is required.

Hopefully you will find that we have made an extensive effort (both from the first round of revisions and this second round) to reduce and simplify the message being presented as much as is realistically possible without losing the "broad documentation" you see value in. Heavy synthesis and broad documentation do not go hand in hand, heavy synthesis removes a lot of the broad documentation, or a lack of heavy synthesis would keep a lot of the broad documentation. We have tried to find a balance between the two. Again, the word "Characteristics" is in the title, this suggests right from the start this paper sets out to create a broad documentation of work.

Through the synthesis:

- Original Figure 2, 12, and 15 have been removed completely, along with Table 5.
- Panels (a), (b) and (d) have been removed in Original Figure 3.
- Original Figures 8 and 9 have been combined into a single Figure, along with removing Panel (c) from the original Figure 8 and Panels (a) and (b) from the original Figure 9. Panels (c) and (d) (i.e., TKE and TKE dissipation) from original Figure 9 have also been combined into a single panel.
- All w'N' data has been removed from the manuscript discussion and associated figures.

Note that roughly 2.5 pages of text has been removed.

We will address more specific concerns/comments below. Please note that all references provided in these comments can be found in the References section of the manuscript.

In addition, grammatical errors persist throughout the paper. The reader encounters sloppiness already in the abstract. (VOMOS?, LFH, "Data was..."). This is inexcusable from native English speakers. I note some grammatical errors below but it's an incomplete list.

We apologize for the typos. We have corrected VOMOS to VAMOS, after checking that it stands for Variability of the American Monsoon Systems. This is the official name of the campaign (VAMOS Ocean-Cloud-Atmosphere-Land Study Regional Experiment (VOCALS-REx)) as named in Wood et. al. (2011). We can find multiple examples (Zheng et al. 2011, Rahn and Garreaud 2010, Bretherton et al. 2010) where they do not explicitly define VAMOS in the abstract.

Secondly, we find the comment "This is inexcusable from native English speakers" to be offensive, inappropriate, and unprofessional coming from a reviewer of a reputable journal. We also note that the second reviewer said that "The article is overall well-written, and easy to read". Regardless, we will make as much changes as realistically possible to improve how the manuscript reads without completely changing our specific writing "voice".

A comment from my first review:

*4) Section 2.1: Shortwave absorption doesn't only occur at cloud top.*

• You are correct, solar absorption does not occur only at cloud top (although this is where it is primarily confined). We have clarified this statement in lines 157-160 of the new manuscript.

The revised text says "shortwave absorption, which is largest near cloud top due to the scattering of solar radiation limiting absorption lower in the cloud layer (Hignett,1991).

But Hignett 1991 states on Pg 1474: "In contrast to the longwave cooling, the heating which results from absorption of shortwave solar radiation is distributed more deeply in the cloud layer". Please do some reading on this topic and revise to better reflect the knowledge in the field.

The point of this comment, is that shortwave radiation acts to stabilize the cloud layer (where longwave cooling is the only source at night, leading to enhanced convection). The entire sentence states "…as turbulence typically displays diurnal patterns, with the strongest turbulent mixing occurring during the night when longwave radiational cooling dominates **due to the absence of the stabilizing effect of shortwave absorption**, which is largest near cloud top due to the scattering of solar radiation limiting absorption lower in the cloud layer (Hignett, 1991)."

The Hignett citation has been moved to just after the bold text above. Hignett on pg. 1474 states "The net result of the combined entrainment, shortwave heating and longwave cooling may be that the cloud layer is warmed relative to the subcloud layer. This implies the formation of a stable layer, which serves to limit the vertical extent of mixing from the surface and reduces transport between the upper and lower parts of the boundary layer".

The sentence in questions states "…of shortwave absorption, **which is largest near cloud top**…". Attached below are two figures from Stephens (1978), which show shortwave heating profiles. These figures show that shortwave radiational heating is largest at cloud top, with a majority of the heating occurring in the upper 20% of the cloud. The statement made in bold is NOT factually incorrect, it is largest near cloud top. To be more precise, it is at a maximum (or is largest) at cloud top. The stratocumulus review from Wood (2012) states on pg. 2392 "solar heating in a cloud layer is largest at the cloud top…." and goes on to cite Stephens (1978). The

Stephens (1978) citation has been added in the location where the original (Hignett 1991) citation was located. Please see lines 147-150 in the new manuscript.

We understand that shortwave absorption depends on a multitude of variables, such as solar zenith angle, cloud optical depth (which is dependent on droplet size). Along with this, roughly 50% of the solar radiation that is absorbed is done so by water vapor, which is a function of cloud thickness and temperature. Seeing as the reviewer already finds the manuscript long and wordy, we do not see the need to go into a detailed discussion of shortwave radiation absorption, especially since this comment is made to simply state that it acts to stabilize (or at the very least limit convection) the boundary layer. Given the references and text provided above, this discussion should be sufficient.

[Figure]

FIG. 5. The shortwave heating profiles in eight cloud models described in Table 3. $Z_N$ represents the total geometric thickness of the cloud. The surface albedo is 0.3 and normal solar incidence is assumed.

[Figure]

FIG. 6. The shortwave heating rates in a Sc I cloud layer for different solar elevation angles. Surface albedo is 0.3 and the cloud is positioned between the 1 and 1.5 km levels in the McClatchey et al. (1972) tropical atmosphere.

Below I give examples of problematic text and places that need improved analysis. More than this would require me to sit down and rewrite paragraphs and I'm not going to do that.

We will address all comments (especially in relation to improved analysis or scientific inconsistencies) and reword where we find it improves the clarity of the science being presented.

**Abstract:**

What is "evaporation away from the surface"? Is it evaporation from the surface, or evaporation some distance from the surface? And if the latter, where? You can calculate this from the qc flux profiles. Slight

We have clarified this statement. From Figure 12 in the new manuscript, the orange envelopes bring attention to the layers in which most of the evaporation is occurring. From this, evaporation is occurring between a $z/z_i$ of 0.4-0.6 (i.e., away from the surface but below cloud base). See lines 14-15 in the new manuscript. This will also be discussed in further detail in specific comments below.

What does "completely offset from one another" mean? Do you mean spatially offset? I waded through the paper and looked at the figures and still don't get it. It's a key result!

Looking at Figure 9 in the new manuscript, Panel (a) shows box plots for in-cloud and sub-cloud TKE measurements. The box represents the 25-75 percent quantiles of the data, with the range of the whiskers representing the range of the data measured. Focusing on Nov. 1st, we can see that the boxplots are completely offset from one another with very minimal overlap (i.e., the range of the turbulence measured is completely different for in-cloud and sub-cloud with minimum overlap). This suggest two completely different turbulent environments for in-cloud and sub-cloud.) Simply, the "completely offset from one another" means that the turbulent values for in-cloud and sub-cloud are completely different.

This is explicitly stated in the manuscript, where lines 473-474 in the original manuscript state "…several cases (including Nov 1st and Nov. 2nd) where the entire turbulent distribution of the sub-cloud data is shifted to larger values than those of in-cloud data, with minimum overlap". This sentence explicitly states what we mean. We have clarified this in the abstract however. Please see lines 18-19 in the new manuscript.

The inability of the LW cooling to mix through deeper boundary layers is not a new result! Begin your sentence with: "As shown previously ..." and then refer to the papers later (as you do).

This has been added. Please see line 8 in the new manuscript.

Lines 10/11 require work. Sentence structure, grammar.

"increases" has been changed to "increase". A grammatical check did not find anything wrong, and this appears to be the only possible issue related to the grammar we could find. Grammatical errors are not intentional, if you have specific grammar issues that are not picked up by several standard grammar checkers please let us know what you are thinking. Other than the error corrected above it is unclear what issue you have. Please see line 11 of the new manuscript.

**Introduction:**

Long and unnecessarily wordy "subgride"?

Sentences and unnecessary words (in our opinion at least) have been removed. Roughly a paragraph of text has been removed in the new manuscript from Section 1.1.

"subgride" has been changed to subgrid-scales. Please see line 37 in the new manuscript.

Line 71: grammar (maintain consistent tense)

Grammar has been corrected to maintain consistent tense. Please see lines 71-73 in the new manuscript.

Line 76: Here and elsewhere: "height throughout the depth of the boundary layer"

"Height throughout the depth of the boundary layer" has been added where necessary. Please see line 77 in the new manuscript.

Line 82: what do you mean by "Convection in the STBL is limited?" By definition, Sc are convective. They may not penetrate very high because of the capping inversion. Please reword.

We were simply stating that convection is limited due to the inversion (which the next two sentence went on to clarify. This phrase has been removed completely in the new manuscript.

Line 91 and throughout: "dryer" is a machine that dries your clothes. "drier".

All three instances where "dryer" is used has been replaced with "drier" Please see lines 91, 515, and 545 of the new manuscript.

Line 91/92. These parentheses within parentheses are very distracting.

This has been changed. Please see lines 89-92 in the new manuscript.

Line 124: Lewellen et al. 1996

We assume you want us to add the Lewellen et al. (1996) reference. Bretherton and Wyant (1997) state: "A similar conclusion was reached by Lewellen et al. (1996) from numerical simulations of shallow stratocumulus layers subject to prescribed surface fluxes. In this paper, we suggest that surface latent heat fluxes are the most important determinant of decoupling for subtropical CTBLs". Therefore, we will add the Lewellen reference and keep the Bretherton reference. Please see lines 115-116 in the new manuscript.

Line 128: vertical velocity skewness is a strong function of diurnal cycle as shown in VOCALS.

For the sake of simplicity, we will leave out the statement that vertical velocity skewness(w'w'w') is a strong function of diurnal cycle (vertical velocity variance is also a strong function of diurnal cycle, but we do not explicitly state this either). It makes the discussion of w'w'w' even more wordy, and since all but two flights have a 7:00 AM start, this is not necessarily relevant to the results. We have already removed some unnecessary statements in regards to the discussion of w'w'w' to simplify it. We state what the expected w'w'w' is for a well-mixed boundary layer (and consequently a decoupled boundary layer) because this is relevant to the results. Also the vertical profile of w'w'w' for a precipitating boundary layer is relevant to the results, which is why it is briefly discussed.

**Data and Methods**

Line 153: Although probes failed on some of the days, the key cloud probes are ones that measure LWC. Surely there were other probes measuring bulk LWC (Hot wire, JW, Gerber PVM?) The aerosol data aren't even used in this paper, except as a possible indicator of reduced drizzle. And w'N' analysis is so inconclusive that I see no reason to keep it.

We have removed the w'N' data (where the PDI or Phase Doppler Interferometer is needed for the N' data since this instrument provides a time series of droplet arrival times). Therefore, we have added four more flights for a total of 18 flights (the PDI is what failed on the four flights that were not initially included). Note that there is no cabin data available for the Nov. 5th flight. This is also conveyed in Zheng et. al. (2011). w'N' has been removed from the manuscript completely.

A side note that is no longer relevant to the manuscript: Since the PDI provides an accurate count of drops since the PDI provides specific arrival times of each droplet with no instrumentation dead time, we believe the w'N' data is more conclusive than one would believe (as opposed to having only 1-hz CDNC data or some other similar dataset). The lack of discussion on the PDI data may have led to this confusion.

Line 178: "e.g., using Reynolds composition" -- an example of how a little effort can make the sentence more readable.

The use of "e.g.," has been implemented. See line 168 in the new manuscript.

Line 204: "assumption that the flow is isotropic". Why was isotropy never directly calculated (see below)?

The isotropy has been added to Figure 10. See the below discussion on the reviewer comment on the anisotropy ratio

Line 221: structure function (n=2)

The necessary correction has been made. Please see line 210 in the new manuscript.

Line 254: "the error bars would not plot"?? this sentence needs rewriting.

The necessary corrections have been made. Please see lines 242-243 in the revised manuscript.

Line 263: What does "this" refer to? An equation? If so say so and refer to the Eq. number.

Line 263: "this" has been replaced with "Equation 7". Please see line 252 in the revised manuscript.

**Synoptic**

Long and low in information content

Roughly a full-page worth of text has been removed from this section. Also, original sections 3.1 and 3.2 have been combined into a single section. Most of the information removed was from the original section 3.1. We did this to keep the focus on the synoptic variability and boundary layer characteristics, which play a direct role in the turbulence results in various ways.

Line 286: remove "as was found in"

This phrase has been removed. Please see line 267 in the new manuscript

Various places: "minimums" ???

According to Merriam-Webster Dictionary, the plural form of minimum can be either "minimums" or "minima". It is plural since we are talking about the minimum for both the 500 and 700-hPa geopotential heights. This has since been rephrased however. See lines 276-278 in the new manuscript.

Line 334: metrics instead of dictators. Line 335, grammar

Metrics has been inserted. Please see line 308 of the new manuscript. The sentence which contained the grammar issues from line 335 has been removed from the new manuscript.

Line 342: what does "per measurement" mean?

 "per measurement", refers to each 1-hz data point available. If the gradient does not exceed the limit provided in the text between each 1-hz data point, then we are outside the range of the inversion. We have added "1-hz" into the new manuscript. Please see lines 313-314 of the new manuscript.

Line 357 grammar

The phrase in question has been removed from the new manuscript.

Line 365: aircraft typically measure aerosols in their dry state. Please check this was not the case here.

The passive cavity aerosol spectrometer probe (PCASP) is an airborne, optical spectrometer that uses the intensity of scattered laser light to measure an aerosol size distribution from $0.1{-}3.0$ $\mu$m. The PCASP doesn't specifically dry particles before measuring them unless there is a heater added to the inlet though the sheath air and is dried which can result in the drying of some, but not necessarily all particles We do note that they do not recommend using the PCASP in cloud due to shattering effects. Also, from the blue profiles in Figure 5 of the new manuscript, the $N_a$ above cloud clearly mirrors the associated profiles of mixing ratio above cloud.

NOTE that the original statement this reviewer comment is based on is no longer in the new manuscript.

https://www.eol.ucar.edu/instruments/passive-cavity-aerosol-spectrometer-probe

[Figure]

Figure 3: PCASP-100X Airflow Diagram

Line 376: helped to produce

"Helped to produce" has replaced "in producing" Please see line 335 in the new manuscript.

Line 386 "that"

The phrase in question has been completely reworded. Please see lines 345-346 of the new manuscript.

Line 298: What is mixed layer cloud thickness in this context? Since you're comparing LCL and cloud base, I think you mean mixed layer thickness from surface to cloud base.

As is stated in the manuscript, mixed layer cloud thickness is "the difference between $z_i$ and the LCL". Per the Jones et al. (2011) (citation provided at the end of the sentence in question) on page 7143: "Decoupling is strongly correlated to the "mixed layer cloud thickness", defined as the difference between the capping inversion height and the LCL".

Line 405 grammar

"are" has been replaced with "is" for the sentence in question (we assume this is what you wanted). Please see line 363 in the new manuscript.

Line 413: rewrite as "By these metrics 28% of..."

This has been rewritten as suggested. Please see line 372 of the new manuscript.

Line 420: "relatively consistent" in terms of what? Depth?

In terms of cloud layer thickness. "thickness" has been added to the sentence to clarify. Please see line 378 of the new manuscript.

**Results**

Line 435: grammar

This sentence has been slightly rewritten, although we are not sure if the grammar you were referring to has been corrected. Please see lines 392-393 of the new manuscript.

Line 436/437: (Table 3) will suffice.

This has been corrected. Please see line 394 of the new manuscript.

Line 458 "are" not "will be".

"will be" has been changed to "are". Please see lines 408-409 of the new manuscript.

Line 460: "Based on" not "From just analyzing.."

The suggested change has been made. Please see line 412 of the new manuscript.

Line 465 grammar

"Coefficient correlation" has been changed to "correlation Coefficient". This is the only thing that we found wrong with this line. Please see line 418 of the new manuscript.

Line 480: driven by

"from" has been replaced with "by", please see line 432 of the new manuscript.

Change TO11, TO 12 to RF11, RF12 (or vice versa) for consistency

All occurrences of TO have been replaced with RF, both in the main text, figure axis labels and legends, and figure captions.

Line 491: Change "Can be" to "is".

The suggested change has been made. Please see line 449 of the new manuscript.

Line 497: Given that zi decreases...

"acts to" has been removed from "given that $z_i$ act to decrease…". Please see line 455 of the new manuscript.

Table 4 gives correlations that sometimes don't fit specific cases. The later focus on Nov 1 should be brought in. How well do Nov 1 and 2 follow the broad correlations?

Nov. $1^{st}$ and $2^{nd}$ follow the correlations (all correlations were calculated including Nov. $1^{st}$ and $2^{nd}$). It would not make sense to remove cases that one would not expect to fit a specific correlation. We would consider this manipulating the data to fit a specific narrative. With that said though, we did highlight (see table caption for Table 3) specific cases relating to the decoupling parameter for mixing ratio ($\alpha_q$) where Nov. $1^{st}$ and $2^{nd}$ were removed. This is because $\alpha_q$ had abnormally large values due to the increase in q above the boundary layer.

For an example, though, correlations with and without (respectively) for specific variables are given:

BLH – GPH = -0.49, -0.46
Omega – GPH = -0.71, -0.58
Wind – GPH = 0.37, 0.42
Wind – LHF = 0.56, 0.61
Wind – SHF = 0.62, 0.60

We won't go through all the variables listed in Table 4, but as you can see, in general there isn't a large change in the correlations. Since Section 4.3 focuses on the Nov. $1^{st}$ case, we can tie in the broad correlations in that section. See below reply to comments for more detail.

Line 514: "Physically this makes sense" does not make sense. There are many studies showing aerosol causing increases and decreases in LWC.

We are looking at the correlations between Na, Nd and drop size. It is general knowledge that enhanced aerosol load leads to larger cloud droplet number concentrations and smaller droplet sizes. We did not calculate the direct correlations between any of these variables and the LWC. However, we have removed "physically this makes sense". We do go on to explain the observed correlations. Please see line 467 in the new manuscript.

Line 516: What are 'enhanced latent heating effects'? Do you mean 'enhanced latent heating'? And if so where?

Yes, we mean the enhanced latent heating (we say effects since latent heating can be a release of energy (condensation) or absorption of energy (evaporation)). We have clarified this statement, and added "in-cloud" to clarify the location. Please see line 469 in the new manuscript.

Fig. 13: Please add a line on your plots for the anisotropy ratio = 2w'w'/(u'u'+v'v'). It goes a long way in showing the source of TKE and its relationship to top-down/bottom-up driven turbulence. Add discussion as necessary.

The mean anisotropy ratio has been added to Panel (e) of Fig. 13. A brief discussion has been added to the text. Please see lines 487-490 of the new manuscript.

Line 581: "downdrafts are smallest". Do you mean smallest areal coverage?

Smallest in spatial coverage. Given we define w'w'w' in the introduction "A positive (negative) vertical velocity skewness indicates that strong, narrow updrafts (downdrafts) are surrounded by larger areas of weaker downdrafts (updrafts)". We shouldn't have to repeat this definition. Especially given the fact that the sentence in question goes on to state "…the downdrafts are smallest, yet strongest at cloud base while updrafts are spatially larger, yet weaker". Regardless, we have added that "…downdrafts are **spatially** smallest, yet strongest at cloud base…". Please see lines 521-522 of the new manuscript.

I'm not convinced that this paper has shown anything interesting about w'N'. The plots are incredibly noisy and no clear results are shown. So why clutter the paper with results like this. Please go back through the paper with a similar view on other plots and remove if they don't yield a clear result. You will be doing yourselves a real service in the long run.

All w'N' data has been removed, along with the included discussion in the text. Our comments at the beginning provide a summary of figures and figure panels that have been removed or changed.

A moot point perhaps but on your figures for w'N', cc-1 is not a standard unit.

We assume you would prefer $cm^{-3}$? Actually, this is a mistake on our part anyways, as N' is the number of drops per second (not a concentration such as Nd).

Figure captions are meant to be cryptic. Please remove all "Note that.." from figure captions.

All "Note that" have been removed from the figure captions.

Figure 16 caption doesn't even say its Nov 1.

Thank you for pointing this out. Nov. $1^{st}$ has been added. Please see the figure caption for Figure 12.

Line 619: Theta is normalized 0 to 1 (min=0; max = 1).  But since theta keeps increasing above the BL, this would typically place the unit value at the highest point of your profile. So how did you normalize theta? The plot suggests you did this correctly.

Theta in original Figure 16 is NOT normalized from 0 to 1…please look at the figure axis. Normalizing theta from 0 to 1 was an example to bring attention to the fact that a large amount of entrained air had resulting in increasing theta throughout the boundary layer down to $z/z_i = 0.4$. This has been removed to simplify the text and not confuse the reader.

Line 625: Where is the evaporation occurring? It makes a difference. Why not calculate the divergence of the qc flux?

We do not know what you mean by the qc flux (do you mean the divergence in the cloud droplet mixing ratio flux?)…. regardless we do not believe this is necessary. The profile of Fθ shows a clear sink for theta (i.e., cooling) while there is a slight enhancement of Fq (i.e., an enhanced source of vapor). The text explains that this is an indicator that evaporation is occurring. And that it primarily is occurring in the orange envelopes. We do believe that the new Panel (c) with Nd

and drop size proves that precipitation is occurring, and shows that it is primarily confined above $z/z_i = 0.40$. Please see lines 575-584 in the new manuscript for the new discussion on this.

Pgph starting line 629: there's another factor and that is less dilution of cloud water in response to entrainment.

The second factor states "Entrainment of more moist air near cloud top, reducing evaporational cooling…". This is only the case if there is less dilution of cloud water. If there was more dilution of cloud water, there would be no reduction in evaporative cooling. The reduced evaporation from the entrainment of more moist air is in relation to the fact that if the air was not more moist, there would not be reduced evaporation. We have reworded this to read "Entrainment of more moist air near cloud top, reducing evaporational cooling that would otherwise occur through the entrainment of drier air". Hopefully this is sufficient to account for this comment. Please see line 544-546 in the new manuscript.

Line 632 grammar

We have changed "top and slows" to "top, slowing" hopefully this is what you were referring to. Please see line 546 in the new manuscript.

Line 638 it's more about the cooling profile than the precip rate, as you discuss later. Please shorten/tighten descriptions.

These few sentences have been reworded. Please see lines 551-553 in the new manuscript.

Line 646: What are 'latent heating effects'? Don't you just mean 'latent heating'?

Yes, we mean "latent heating", this has been changed. Please see lines 559-560 in the new manuscript.

Also, note that "latent heating effects" is common terminology. Wood (2012), a paper with over 600 citations, uses the phrase repeatedly. One example on page 2394: "Large buoyancy fluxes are caused mainly by radiative cooling and are additionally enhanced by latent heating effects".

Line 652 and abstract/conclusions, "away from the surface" is highly imprecise language, and since this is a key result please make your arguments crystal clear. The profiles show no sub-cloud drizzle that would destabilize the sub-cloud layer. Does LWC include drizzle/rain?

Panel (c) in Figure 12 now makes it clear that (1) yes there is sub-cloud drizzle occurring. The text also clarifies where this evaporation is occurring (primarily near $z/z_i \sim 0.40$ and 0.60, the two orange envelopes from Figure 12). As per the Phase Doppler Interferometer (PDI), and the discussion on lines 575-583 of the new manuscript, there are so few droplets in the sub-cloud layer we would assume this is why the LWC profile does not indicate any liquid water. The PDI is an extremely precise instrument. The droplet arrival time can accurately be measured to < 3.5 µs, resulting in accurately mapping droplets down to 2.1E-4 m (assuming average aircraft speed).

Also, note that in the w'N' profile from the original manuscript, this would have indicated that yes, there was sub-cloud drizzle occurring (as w'N' would be zero when there is no N present, as

was the case for original Figure 17).

As mentioned above, please tie this discussion of Nov 1 back to the broader correlations and point out differences if they exist, or simply note consistency. I couldn't wade through the text one more time to figure it all out.

We will simply point out consistency. Please see lines 603-605 of the new manuscript.

Line 652: "activation" not "activation through condensation". Please stick to accepted terminology.

All discussions relating to w'N' have been removed.

Line 664: I don't believe you have evidence of aerosol activation in updrafts at cloud base. Your measurements are far too noisy to draw this conclusion. Updraft velocities are just not strong enough to raise supersaturation at cloud top. Why is this not simply evaporation in downdrafts? And as I said earlier, since you can't really say much about aerosols, why dilute your message with distracting, uncertain results?

All discussions relating to w'N' have been removed.

Line 655, poorly mixed BLs can be locally well coupled by local, penetrative updrafts.

This is true, but we do not see any evidence of this for the Nov. 1$^{st}$ case. Therefor we do not see the need to include such a statement.

Line 671: Can you explain why the consistency in wind direction across the inversion should translate to mixing across the inversion? It's possible that mixing occurs but this isn't a way to address it. The theta jump is about 4K.

This discussion has been removed to simplify the text.

Fig. 13: Why are there 2 identical blue lines for RF12 and 13 when the caption and legend lump them together?

There are two lines, for the two flights (RF11 and RF12)…and they are not identical. Each line represents the profile for each flight (as was the case in original Figure 6 and 12). Perhaps you are looking at a different figure than Fig. 13? We went through all the figures and cannot identify what you are saying in this comment.

We did change the blue color however. Instead of two identical light blue profile lines, we have one dark blue and one light blue profile line in Figures 3, 5, and 10.

General comment on SHF and LHF: these are generally defined at the surface so please clarify. Eqns 2, 3 give the definitions. You should define them as w'q', w'theta'!

This has been implemented throughout the text (i.e., $F_\theta$, $F_q$, and $F_{\theta v}$ per the equation definitions) unless we are specifically talking about surface values, in which case we use LHF and SHF.

**Conclusions**

Conclusions need to be tightened/clarified.

Clarification has been made where needed (i.e., specific instances outlined below).

Line 703: entrainment acts to increase zi. Zi is a number that can increase or decrease but not deepen.

This has been corrected (throughout the manuscript). Please see lines 622 of the new manuscript.

Line 710: please add appropriate reference

This statements was also made in Section 4.1. The reference (Bretherton and Wyant 1997) has been inserted at that location. Please see lines 96-98 of the new manuscript. Adding the reference here at the end seems out of place.

Line 711: grammar

"As the LHF and SHF increases (decreases), zi increases (decreases)" has been changed to " As the LHF and SHF increase (decrease), zi increases (decreases)". Please see lines 628-631 of the new manuscript.

Line 712/713: this is a convoluted sentence. Deepening BLs entrain drier air and therefore increase surface LHF

This is incorrect. A deepening boundary layer through the entrainment of drier air does not lead to an increase in surface LHF. The increase in LHF is what leads to the deepening boundary layer. Please see the discussion on lines 457-464 of the new manuscript (back in Section 4.1). This sentence has been restructured to make the point more clear. Please see lines 628-631 of the new manuscript.

Line 715: Again, please tell us what you mean by evaporation away from the surface.

This has been clarified throughout. Please see lines 633-634 of the new manuscript.

 Line 713: again explain: "completely offset one another"

This has been clarified throughout the manuscript.

Line 738: grammar

We were unable to determine a grammatical error on or around line 738.

**List of Relevant Changes**

(1)  Four flights were added to the analysis, increasing the total from 14 to 18 flights.

(2) Section 3.1 and Section 3.2 were combined into a single section: Section 3.1 Synoptic Variability at Point Alpha.

(3) Figure 2, Figure 12, Figure 15, and Table 5 have been removed completely from the original manuscript.

(4) Panels (a), (b), and (d) have been removed from Figure 3 in the original manuscript.

(5) Figures 8 and 9 in the original manuscript have been combined into a single Figure (Figure 7 in the new manuscript). Panel (c) from the original Figure 8 and Panels (a) and (b) from the original Figure 9 have been removed. Panels (c) and (d) (i.e., TKE and TKE dissipation) from original Figure 9 have been combined into a single panel.

(6) All w'N' data have been removed from the manuscript, including Panel (d) in Figure 14 and from Panel (c) in Figures 16 and 17.

[revised manuscript text omitted]

---

## Author Response (AR3)

**Reply to Editor:**

NOTE: Editor comments are in black below, while the reply to comments are followed in blue.

Comments to the Author:
Dear Authors

Thank you for providing your detailed comments and revisions in response to the re-review of this manuscript. Most of the outstanding issues have now been addressed. However, given the extensive guidance on spelling and grammatical errors provided by the reviewer, I was surprised to find multiple language issues in the new version of the manuscript. I am highlighting a few of these below but would expect the authors to perform a careful check of language and grammar before resubmission.

Specifically:

1) Spelling: L481 "similr",

This has been corrected. Please see lines 409 in the revised manuscript.

2) Grammar: L480 ", however, "

This has been corrected. Please see lines 408 in the revised manuscript.

3) Spelling: L589 "Vertical turulence"

This has been corrected. Please see lines 487 in the revised manuscript.

On top of these corrections, several other slight changes have been made within the manuscript, including other spelling corrections and slight addition/subtraction to some sentences to help with word flow. The most noteworthy change has been providing the symbol $Z_*$ for "in-cloud normalized height". Please see the tracked changes version of the manuscript for all changes.